# ETC: TRAINING-FREE DIFFUSION MODELS ACCELERATION WITH ERROR-AWARE TREND CONSISTENCY

## ABSTRACT

Diffusion models have achieved remarkable generative quality but remain bottlenecked by costly iterative sampling. Recent training-free methods accelerate diffusion process by reusing model outputs. However, these methods ignore denoising trends and lack error control for model-specific tolerance, leading to trajectory deviations under multi-step reuse and exacerbating inconsistencies in the generated results. To address these issues, we introduce Error-aware Trend Consistency (ETC), a framework that (1) introduces a consistent trend predictor that leverages the smooth continuity of diffusion trajectories, projecting historical denoising patterns into stable future directions and progressively distributing them across multiple approximation steps to achieve acceleration without deviating; (2) proposes a model-specific error tolerance search mechanism that derives corrective thresholds by identifying transition points from volatile semantic planning to stable quality refinement. Experiments show that ETC achieves a 2.65× acceleration over FLUX with negligible (-0.074 SSIM score) degradation of consistency.[1]

## 1 INTRODUCTION

Diffusion models (Sohl-Dickstein et al., 2015; Song & Ermon, 2019; Ho et al., 2020) have demonstrated remarkable generative capabilities across diverse domains including images, videos and audio. However, their superior generative performance typically requires larger model architectures and multi-step denoising processes, resulting in substantial computational overhead and inference latency. Training-based methods (Salimans & Ho, 2022; Luo et al., 2023) accelerate diffusion by learning a few-step model from the denoising process of a multi-step model. However, this paradigm requires extensive training and there often remains a discrepancy between the predictive distributions of the original and the few-step model (Stanton et al., 2021), weakening generalization capabilities.

In contrast, training-free methods (Chen et al.; Ye et al., 2024) accelerate diffusion without model retraining or performance degradation by leveraging feature similarity between adjacent timesteps, broadly categorized into inner-level and step-wise feature reuse mechanisms. Inner-level (Ma et al., 2024a; Liu et al., 2025b) methods accelerate inference by reusing internal layer features within each denoising iteration but require architecture-specific designs. In contrast, step-wise (Ye et al., 2024; Liu et al., 2025a) methods evaluate model output robustness to determine multi-step reuse and reduce the total number of model inferences, achieving better generalized acceleration compared to inner-level alternatives. However, step-wise methods overlook denoising trends between adjacent timesteps, leading to inconsistent generation results. While recent approaches (Chen et al.; Liu et al., 2025a) attempt to address this by employing residuals between model outputs for trend prediction, short-term output fluctuations often deviate from the long-term denoising trajectory, thereby exacerbating trend inconsistency caused by the multi-step use of fluctuating residuals as shown in Figure 1a. Consequently, achieving **multi-step approximations with consistent trajectories** remains an unresolved challenge. Moreover, existing methods rely on manually-defined fixed thresholds for approximation error assessment, neglecting model-specific error tolerances shown in Figure 1b. Consequently, fixed threshold strategy may fails to address scenarios where accumulated errors cause irreversible trajectory deviations beyond the model's corrective capacity, resulting in **inadequate error control** and amplified generation inconsistency. To summarize, step-wise feature

---

[1]More samples and source code are available at `https://etcdiff.github.io/`

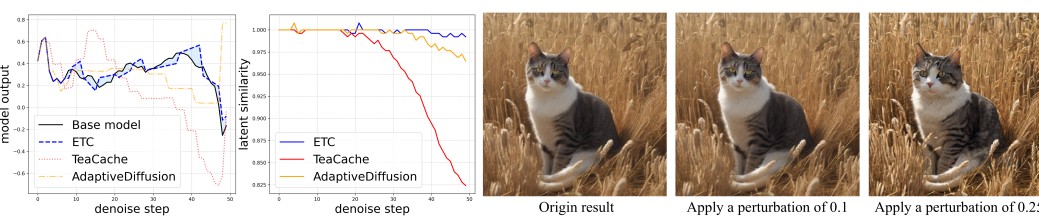

(a) Comparison of denoising trajectories      (b) Effect of different levels of denoising errors

Figure 1: Visualization of trajectory deviation and denoise error tolerance. Subfigure (a) shows that existing methods fail to follow the original denoising trajectory and reduce latent similarity. Subfigure (b) shows the model maintains consistent results to a certain degree of denoising errors.

reuse research demonstrates significant untapped acceleration potential but is challenged with trajectory deviations under multi-step approximation due to the lack of accurate capture of denoising trends, and inadequate error control due to insufficient exploration of model error tolerance limits.

To address the above problems, we propose ETC, a framework that achieves step-wise diffusion acceleration through trajectory consistency and model-specific error control. For the multi-step trajectory consistency problem, we aim to derive future directional flows from the global stabilization properties inherent in the denoising process. Specifically, we compute weighted projections of cross-step changes in model outputs, attenuating fluctuations caused by approximation errors while amplifying directional components of long-term dynamics. To further enhance acceleration, we design an adaptive expansion strategy that dynamically extends or contracts the approximation window based on whether deviations between projected trends and periodic model inferences remain within the model's corrective capacity. Moreover, we employ a progressive distribution paradigm that proportionally allocates estimated direction flows across each approximation step, thereby achieving aggressive acceleration while preventing divergence from the denoising trajectory. As for the inadequate error control problem, we treat different models' error tolerance as emergent properties within denoising dynamics. Observing that the semantic planning phase in denoising process exhibit high dynamic variance (Liu et al., 2025b), we quantify the perceptual influence of deviation perturbations on generation quality and derive critical transition points from volatile semantic planning to smooth quality refinement phases that reflect model limits for error correction.

In summary, the main contributions of out work are as follows: (1) We introduce a consistent trend predictor that projects stabilizing denoising trends into future flows and progressively distributes them across approximation steps to ensure trajectory consistency. (2) We propose a model-specific error tolerance search method that quantifies the impact of accumulated errors and locates the transition point between volatile semantic planning and stable quality refinement to reflect the error tolerance threshold. (3) Experiments show that ETC outperforms other state-of-the-art baselines in terms of generation consistency and speed across images, videos and audio synthesis.

## 2 RELATED WORK

**Diffusion model** Diffusion models have achieved great success in generative tasks. Early diffusion models (Ho et al., 2020; Song et al., 2020a) implemented iterative denoising processes directly on the original data modality, which imposed computational overhead due to high-dimensional operations. To address computational efficiency constraints, latent diffusion models (Rombach et al., 2022; Blattmann et al., 2023) compress the original modality into lower-dimensional representations before applying the diffusion process. Initially developed with the U-Net (Ronneberger et al., 2015), latent diffusion models have demonstrated impressive performance. However, U-Net-based design encountered scalability constraints that limited larger model training and practical deployment. Diffusion Transformers (DiT) (Peebles & Xie, 2023) address this limitation by leveraging transformer (Vaswani et al., 2017) for enhanced scalability and achieve state of-the-art performance across diverse domains (Labs, 2024; Wan et al., 2025; Hung et al., 2024). Despite the high quality achieved by diffusion models, the multi-step denoising design slows down the inference process.

**Diffusion model acceleration** Diffusion model acceleration can be categorized into training-based and training-free approaches. Training-based methods aim to learn a few-step model from the multi-step denoising process. Progressive distillation (Salimans & Ho, 2022) progressively matches

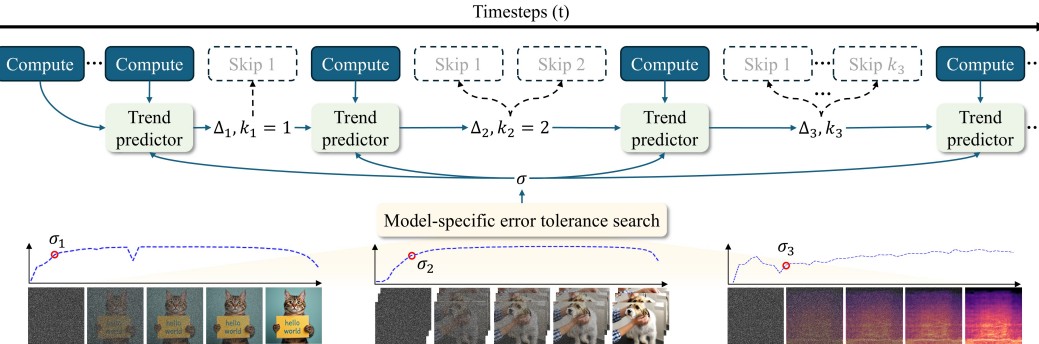

Figure 2: An overview of ETC. ETC leverages all historical model outputs to estimate future trends and dynamically adjusts approximation frequency according to each model's error tolerance limit.

noise predictions between teacher and student models. Latent Consistency Model (LCM) (Luo et al., 2023) achieves single-step sampling by imposing self-consistency constraints. However, these methods require time-consuming training and often suffer performance degradation due to the gap between the predictive distributions of the few-step and original models (Stanton et al., 2021).

In contrast, recent training-free approaches leverage feature reuse between adjacent time steps for acceleration and can be divided into inner-level and step-wise feature reuse. Inner-level reuse accelerates diffusion by reusing features within different layers of the model. DeepCache (Ma et al., 2024b) caches high-level features in the U-Net while dynamically updating only shallow features in subsequent steps. T-Gate (Liu et al., 2025b) caches DiT cross-attention outputs after convergence and keeps them fixed for remaining steps. However, these methods are model-specifically designed and lack generalizability. Step-wise methods exploit cross-temporal output similarity for general acceleration. AdaptiveDiffusion (Ye et al., 2024) detects redundancy across denoising steps using a third-order differential estimator and reuses historical model outputs. However, it neglects the denoising trend and leads to gradual trajectory deviations. TeaCache (Liu et al., 2025a) minimizes trajectory deviation by using residuals between adjacent steps as approximate trends, but it may amplify error accumulation due to short-term denoising fluctuations. SADA (Jiang et al., 2025) enhances trend estimation by combining three-step historical results, yet its acceleration potential is limited by a maximum four-step jump. Although step-wise methods show acceleration potential, maintaining trend consistency in multi-step approximations remains a challenge. Furthermore, existing step-wise methods rely on manually predefined thresholds to determine the reusability, resulting in inconsistent performance across different models due to the lack of model-specific thresholds.

In this paper, we focus on maintaining the consistency of denoised trajectories during step-wise acceleration. We combine all historical trend patterns to obtain more stable future direction estimates and identify model-specific error tolerances, ensuring the approximation process stays aligned with the original trajectory without significant deviation.

## 3 METHODOLOGY

In this section, we provide a detailed description of the architectural components of our proposed framework, ETC. As shown in Figure 2, the overall framework consists of two main modules: 1) Consistent trend predictor for predicting future denoising trends and approximate frequencies; 2) Model-specific error tolerance mechanism for identifying error thresholds across different models. The following subsections detail the design of each module.

### 3.1 PRELIMINARY

**Diffusion denoising process** Diffusion models consist of a forward process that progressively corrupts data with Gaussian noise and a reverse process that reconstructs the clean data through iterative denoising. During inference, only the reverse process is executed, starting from the Gaussian noise $x_T \sim \mathcal{N}(0, I)$ and progressively refines $x_T$ into the target output $x_0$ conditioned on input signal $c$ (e.g., text prompts), where $T$ is the predefined number of denoising steps. The refinement follows a

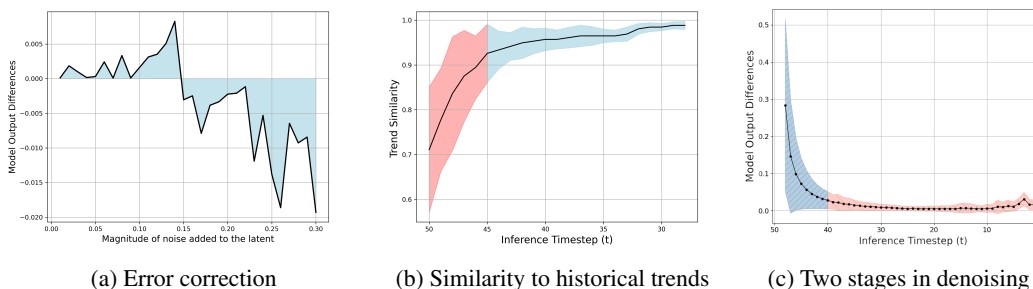

(a) Error correction      (b) Similarity to historical trends      (c) Two stages in denoising

Figure 3: The patterns of the denoising process observed during inference with FLUX on MSCOCO-2017 validation set. Subfigure (a) shows how the model output varies under different latent error. Subfigure (b) illustrates the similarity of current trend to each historical trends. Subfigure (c) depicts the stability of trend changes across different denoising stages.

generic update rule defined at each timestep $t$ :

$$x_{t-1} = f(t-1) \cdot x_t - g(t-1) \cdot \epsilon_\theta(x_t, t, c), \quad t = 1, \ldots, T, \tag{1}$$

where $\epsilon_\theta(x_t, t, c)$ is a noise prediction network trained to estimate the noise component in $x_t$ by taking $x_t$, timestep $t$ and an additional condition $c$ as input, while $f(t)$ and $g(t)$ are coefficients determined by the sampler (a.k.a, scheduler) (Ho et al., 2020; Song et al., 2020a). Both the noise prediction network $\epsilon_\theta$ and sampler coefficients $f(t)$, $g(t)$ directly impact generation quality. Using the same sampler ensures the consistency of $f(t)$ and $g(t)$, so the generation quality is primarily affected by the differences of the noise prediction model $\epsilon_\theta$. Detailed discussion is provided in Appendix A.2.

**Fluctuation and stabilization phases of model outputs** Previous works (Ho et al., 2020; Wu et al., 2024; Lee et al., 2025) have observed that diffusion models demonstrate substantial low-frequency changes in the early denoising steps, followed by more stable high-frequency refinements in later steps. Furthermore, Liu et al. (2025b) divided the denoising process into semantic planning phase and fidelity refinement phase by observing that cross-attention maps fluctuate in the early stage and become stable afterward. In alignment with these findings, we observe a similar pattern shown in Figure 3c by analyzing the difference in model outputs between two consecutive steps. Throughout most of the denoising process, model outputs between adjacent timesteps display stable change patterns. Therefore, we make the following assumption.

*Assumption 1* The model output $\epsilon_\theta(x_t, t, c)$ is jointly Lipschitz continuous in both $x_t$ and $t$, exhibiting structured and predictable variations across time steps.

The proof of Lipschitz continuity and a more detailed discussion of model output trends are provided in the Appendix A.2 and A.4. In this paper, we aim to estimate model outputs based on historical trends while ensuring cumulative errors do not cause significant deviations in the generated results.

### 3.2 CONSISTENT TREND PREDICTOR

**Historical denoising pattern projections** Based on Assumption 1, we can approximate the future denoised trajectory using the recent trend. However, as illustrated in Figure 3a, when the approximated trend introduces errors in the latent space, the model output increases compared to the original result in order to correct the cumulative error. Continuing to use this trend for future trajectory approximations leads to fluctuations and further exacerbates trajectory deviation. Therefore, our goal is to approximate trajectories by combining multiple historical trends, thereby minimizing the fluctuations introduced by the error-corrected model output.

Let $d_t^{t+1} = \epsilon_\theta(x_t, t, c) - \epsilon_\theta(x_{t+1}, t+1, c)$ represent the model outputs difference between time steps $t$ and $t + 1$. As shown in Figure 3b, recent historical trends more accurately reflect future changes, while even trends from earlier time steps maintain a similarity above 0.7, indicating they still capture some directional information. Therefore, we use a weighted sum of all historical trends, assigning higher weights to more recent trends. However, storing all historical trends incurs considerable computational overhead. To address this, we propose a recursive historical trend weighting method,

formulated as follows:

$$\Delta_{t-2} = (1-\alpha)\Delta_{t-1} + \alpha d_{t-1}^t, 0 \leq t < T, \Delta_{T-2} = d_{T-1}^T, \tag{2}$$

where $\Delta_{t-2}$ represents the estimated trend starting from time step $t-2$, $\alpha$ is the trend adjustment coefficient used to reduce volatility caused by error correction while preserving consistency with the historical denoised trajectory. Only the model outputs and the final approximation value from each round of multi-step approximation are used to calculate the future estimated trends. Furthermore, as shown in Figure 3b, the correlation between the trend in the initial denoising stage and the future trend shows substantial fluctuations (the red area represents the variance). Therefore, we first allow the model to perform n denoising steps to obtain a more stable trend estimate before initiating the approximation. Once the estimated trend is obtained, we can calculate the approximation output $\epsilon_{\theta}'$ using the following formula:

$$\epsilon_{\theta}'(x_{t-2}, t-2, c) = \epsilon_{\theta}(x_{t-1}, t-1, c) + \Delta_{t-2} \tag{3}$$

**Dynamic approximation window expansion strategy**    To achieve faster acceleration, our goal is to maximize the approximation frequency while maintaining trajectory consistency. We add a fixed value from the latent to represent the approximation error. As shown in Figure 3a, within the error correction range, the model can produce larger outputs to compensate for the reduced denoising caused by the error. However, if the error exceeds this range, the model's output begins to diverge from the original result. Based on this observation, we propose an approximation window expansion strategy. If the cumulative error of the previous iteration is below the threshold, the approximation step for the next iteration increases; otherwise, it decreases. The formula is as follows:

$$k_{t-2} = \begin{cases} k_{t-1} + 1, & |d_{t-1}^t - \Delta_{t-1}| < \psi, \\ k_{t-1} - 1, & |d_{t-1}^t - \Delta_{t-1}| \geq \psi, \end{cases} \tag{4}$$

where $k$ is the approximation step and $\psi$ is the threshold. To maintain consistency in multi-step approximations, we distribute the estimated trend averagely across each step. This design ensures that the final approximation direction aligns with the estimated trend, preventing the accumulation of estimation errors that could lead to trajectory deviation. The formula for calculating the approximation output at each step is as follows:

$$\epsilon_{\theta}'(x_{t-1-s, t-1-s, c}) = \epsilon_{\theta}(x_t - 1, t-1, c) + \frac{s}{k_{t-2}}\Delta_{t-2}, 1 \leq s \leq k_{t-2}. \tag{5}$$

**Error estimation**    The detailed analysis of estimation errors is provided in Appendix A.1. Assuming $\alpha = 0.5$, the formulation for the upper bound of cumulative error is as follows:

$$error_r \leq \| \sum_{m=0}^{k_r-1} ((1 + \frac{k_r - m}{2k_r})\sigma_{r-1} + \frac{k_r - m}{2k_r} \cdot d_{T-n-\sum_{l=1}^{r-1}(k_l+1)}^{T-n-\sum_{l=1}^{r-2}(k_l+1)} - d_{T-n-\sum_{j=1}^{m}(k_j+1)-k_r+m}^{T-n-\sum_{j=1}^{m-1}(k_j+1)})\|. \tag{6}$$

Let $t = T - n - \sum_{l=1}^{r-2}(k_l + 1)$. As shown in Figure 4, the approximation error depends on the alignment between the approximated trend (a weighted combination of $d$ and $\sigma$) and the future trend. Assuming $\sigma = 0$, the error depends solely on how much the future trend deviates from that of the previous round, ensuring that the denoised trajectory remains at least consistent with the prior step. Consequently, as long as the cumulative error in the early rounds is limited and $\sigma$ remains

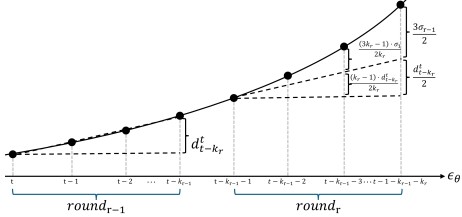

Figure 4: Error accumulation at $\alpha = 0.5$.

small, the estimated trend remains controllable and the denoising trajectory maintains consistency.

### 3.3 MODEL-SPECIFIC ERROR TOLERANCE SEARCH MECHANISM

**Necessity of error control**    Assuming that a fixed error $\beta$ is introduced at every timestep, the accumulated error at the final timestep can be formulated as:

$$error = \sum_{m=0}^{T-2}\left(\left(\prod_{j=0}^{T-2-m}f(j)\right) \cdot g(T-1-m) \cdot \beta\right) + g(0) \cdot \beta. \tag{7}$$

The proof can be found in Appendix A.5. As shown in Equation 7, any error introduced during the denoising process will progressively accumulate and be amplified. Therefore, it is necessary to ensure the accumulated error within a single approximation round does not lead to trajectory deviation. Otherwise, the deviation will be further amplified over multiple rounds, degrading the final generation quality.

**Error tolerance search** When model outputs exhibit semantic planning volatility, trajectory deviation may occurs and subsequent steps amplify errors. Therefore, we regard the semantic planning phase as fluctuations that occur outside the model's error tolerance range. By identifying the transition point from this fluctuating phase to the stable quality enhancement phase, we can approximate the model's error tolerance limit. Specifically, we introduce the model output difference $d$ from each denoising step as as an accumulated error added to the final latent and compute the similarity between the perturbed result and the original generated result:

$$s_i = Sim(D(x_0 + d_i), D(x_0)), \quad i = 1, ..., T, \tag{8}$$

where $x_0$ is the denoised output at the final timestep, $D$ is the decoder and $Sim$ is the similarity metrics. Using trend inflection point analysis tools (e.g., the ruptures package) to analyze the similarity trend, we can determine the model output difference at the transition point between the two phases, which corresponds to our estimated model error tolerance limit. The detailed process of error tolerance search and inflection point analysis is provided in Appendix A.7 and A.11.

# 4 EXPERIMENTS

## 4.1 EXPERIMENTAL SETTINGS

**Base models and compared methods** To demonstrate the general effectiveness of our method, we apply our technique to various diffusion models for image, video, and audio generation, including SDXL (Podell et al., 2023) and Flux (Labs, 2024) for image, Open-Sora 1.2 (Zheng et al., 2024) and Wan 2.1 (Wan et al., 2025) for video, and TangoFlux (Hung et al., 2024) for audio. We compare our method with recent state-of-the-art acceleration approaches, including AdaptiveDiffusion (Ye et al., 2024), SADA (Jiang et al., 2025), TeaCache (Liu et al., 2025a) and MagCache (Ma et al., 2025). To better assess the efficiency-quality trade-off, we utilize the fast configurations of TeaCache and MagCache for fair comparison.

**Evaluation metrics and datasets** We evaluate acceleration methods across two key dimensions: computational efficiency and generation quality. For efficiency assessment, we report Floating Point Operations (FLOPs), inference latency and speedup ratios. For quality evaluation, we employ modality-specific metrics to comprehensively assess the fidelity between accelerated and original outputs. For image and video generation, we measure visual similarity using LPIPS (Zhang et al., 2018), PSNR, and SSIM. Additionally, we evaluate text-image alignment using CLIP score (Radford et al., 2021) for images and employ VBench (Huang et al., 2024) for multi-aspect video quality assessment. For audio generation, we measure acoustic fidelity using FAD (Kilgour et al., 2018), MCD (Kubichek, 1993), and KL-divergence of classification probabilities (Copet et al., 2023; Koutini et al., 2021), while assessing text-audio alignment with CLAP score (Wu et al., 2023). To ensure fair comparison, all experiments are conducted using standardized prompt datasets: MSCOCO-2017 (Lin et al., 2014) validation set for image generation, VBench prompts for video generation and AudioCaps (Kim et al., 2019) test set for audio generation.

**Implementation details** All experiments are conducted on a same GPU using PyTorch, with FlashAttention (Dao et al., 2022) enabled across all configurations. To efficiently determine the step-skipping threshold, we sample a small set of prompts for each modality. For image and audio models, we partition MSCOCO-2017 and AudioCaps training prompts into 10 length-based bins and sample one prompt per bin. For video models, we sample one prompt from each VBench evaluation dimension. Threshold results are provided in Table 1. The hyperparameter $n$ is set to 4 for OpenSora and 6 for the remaining models, while $\alpha$ is all set to 0.5. SDXL, FLUX, OpenSora and TangoFlux use the Euler solver, while Wan uses the UniPC solver (Zhao et al., 2023).

## 4.2 MAIN RESULTS

**Quantitative comparison** Table 1 presents quantitative results evaluating both computational efficiency and generation quality. ETC achieves superior acceleration performance across diverse tasks

Table 1: Quantitative evaluation results. Best performance in **bold**, and the second best underlined. Our method achieves superior efficiency-quality trade-off across diverse tasks and architectures.

| Method | Visual Quality | | | | Efficiency | | |
|---|---|---|---|---|---|---|---|
| **Text to Image** | **LPIPS ↓** | **SSIM ↑** | **PSNR ↑** | **CLIP ↑** | **FLOPs (P) ↓** | **Speedup ↑** | **Latency (s) ↓** |
| SDXL ($T = 50$) | - | - | - | 27.253 | 0.67 | 1× | 9.63 |
| AdaptiveDiffusion | 0.172 | 0.816 | 25.062 | 27.048 | 0.31 | 1.98× | 4.87 |
| SADA | **0.096** | **0.882** | **28.881** | 27.055 | 0.35 | 1.81× | 5.07 |
| Ours ($\sigma = 0.0171$) | 0.105 | 0.876 | 28.017 | **27.103** | **0.26** | **2.12×** | **4.53** |
| FLUX ($T = 50$) | - | - | - | 26.319 | 3.64 | 1× | 28.85 |
| TeaCache-fast | 0.281 | 0.753 | 18.845 | 25.924 | 1.45 | 2.45× | 11.78 |
| SADA | **0.062** | 0.923 | **29.342** | 25.979 | 1.74 | 2.07× | 13.94 |
| Ours ($\sigma = 0.1269$) | 0.068 | **0.926** | 29.176 | **25.983** | **1.31** | **2.65×** | **10.86** |
| **Text to Video** | **LPIPS ↓** | **SSIM ↑** | **PSNR ↑** | **VBench ↑** | **FLOPs (P) ↓** | **Speedup ↑** | **Latency (s) ↓** |
| Open-Sora 1.2 ($T = 30$) | - | - | - | 77.18% | 3.15 | 1× | 45.29 |
| TeaCache-fast | 0.217 | 0.775 | 20.601 | 76.10% | 1.51 | 2.06× | 21.95 |
| MagCache-fast | 0.164 | 0.823 | 22.583 | 75.84% | 1.41 | 2.11× | 21.45 |
| Ours ($\sigma = 0.2101$) | **0.131** | **0.848** | **24.123** | **76.44%** | **1.32** | **2.15×** | **21.07** |
| Wan 2.1 ($T = 50$) | - | - | - | 81.01% | 76.71 | 1× | 970.49 |
| TeaCache-fast | 0.317 | 0.587 | 17.329 | 80.06% | 38.63 | 1.91× | 508.31 |
| MagCache-fast | 0.135 | 0.787 | 22.867 | 80.63% | 32.49 | 2.38× | 407.64 |
| Ours ($\sigma = 0.1181$) | **0.103** | **0.806** | **24.078** | **80.71%** | **28.63** | **2.49×** | **389.89** |
| **Text to Audio** | **FAD ↓** | **MCD ↓** | **KL ↓** | **CLAP ↑** | **FLOPs (T) ↓** | **Speedup ↑** | **Latency (s) ↓** |
| TangoFlux ($T = 50$) | - | - | - | 13.286 | 46.86 | 1× | 5.49 |
| TeaCache-fast | 0.101 | 3.598 | 0.187 | 13.225 | 16.58 | 2.27× | 2.42 |
| AdaptiveDiffusion | 0.042 | 3.145 | 0.181 | **13.309** | 27.91 | 1.48× | 3.72 |
| Ours ($\sigma = 0.0675$) | **0.026** | **1.877** | **0.157** | 13.251 | **14.83** | **2.43×** | **2.26** |

and architectures while maintaining high visual quality. In evaluating the Wan 2.1 text-to-video baseline, our method achieves a 2.5× speedup while maintaining the highest SSIM of 0.806, outperforming MagCache which attains only 2.38× acceleration with 5% lower generation consistency. With the TangoFlux text-to-audio baseline, our approach delivers the highest speedup of 2.43× with an MCD score 40% lower than AdaptiveDiff, which achieves only 1.48× acceleration due to its design prioritizing generation quality. For text-to-image baselines, our method achieves a 2.65× speedup on Flux with an SSIM of 0.926. Although our SSIM on SDXL is 0.6% lower than SADA, we achieve 22% faster inference, demonstrating a superior efficiency-quality trade-off.

**Visualization** Figure 5 compares the results generated by ETC against those by the original model and other baselines. For image generation, TeaCache and AdaptiveDiff exhibit noticeable structural distortions, such as changes in car front designs and chair leg deformations. Although SADA preserves the overall structure, it suffers from detail loss and text generation errors, such as the extra piece of litter near the chair and the altered license plate from "NX4 5KJ" to "AX4 5K". For video generation, TeaCache shows significant content deviation, with differences in dog appearance and petting gestures. MagCache maintains structural alignment but loses fine details such as raccoon fur texture and the distinctive yellow whiskers around the dog's mouth. For audio generation, while the task complexity is relatively lower and most methods preserve similarity to original results, TeaCache still introduces artifacts such as noise distortions highlighted in the red box. In contrast, our method preserves both global consistency and fine-grained detail through various tasks, demonstrating superior generation fidelity.

## 4.3 ABLATION STUDIES

**Statistical robustness** To evaluate the stability of generation performance across diverse text conditions, we analyze the distribution of SSIM and latency metrics using box plots for all three generation tasks. As shown in Figure 6, TeaCache and MagCache exhibit stable acceleration performance with narrow latency distributions across different prompts. However, for Wan 2.1, TeaCache's SSIM spans a wide range from 0.1 to 0.9, while MagCache shows slight improvement but still ranges from 0.5 to 0.9. In contrast, our method shows moderate latency variance but achieves significantly tighter SSIM distributions of 0.7 to 0.9. While SADA and AdaptiveDiff show similar adaptive behavior with moderate latency variation, both methods suffer from quality instability with numerous outliers compared to our method. These results demonstrate that ETC effectively balances acceleration and stability, achieving more consistent generation quality across diverse conditions.

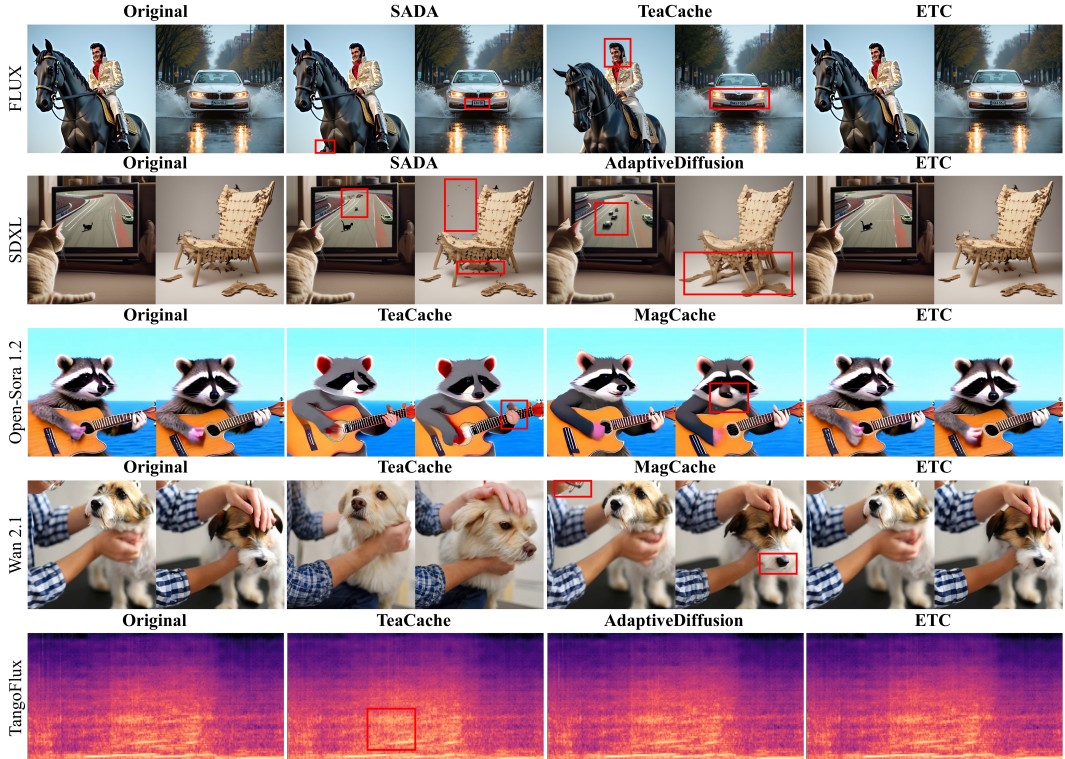

Figure 5: Comparison of visual quality with the competing method. Other methods exhibit issues such as text failure and missing details, whereas ETC achieves the best generation consistency.

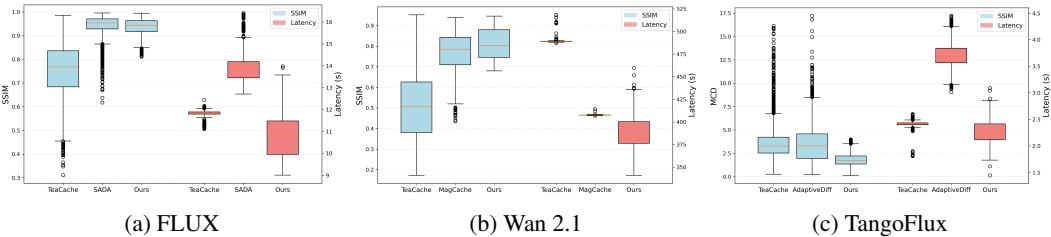

(a) FLUX      (b) Wan 2.1      (c) TangoFlux

Figure 6: Boxplots of SSIM and latency metrics across different tasks.

**Effectiveness of error control** To investigate error accumulation from approximate noise, we computed differences between approximated and actual noise at each timestep, along with corresponding latent bias. As shown in Figure 7, noise error exhibits two distinct patterns: fluctuating increase and steady increase. Tea-Cache demonstrates steady error growth due to its simplistic approximation using adjacent historical noise differences, introducing significant bias during high-fluctuation initial phases that propagates throughout denoising. Our method follows the fluctuating pattern but accumulates error slower than MagCache and Adap-tiveDiff. While SADA exhibits a similar error accumulation pattern to ours, it achieves lower accumulation rates by sacrificing inference speed.

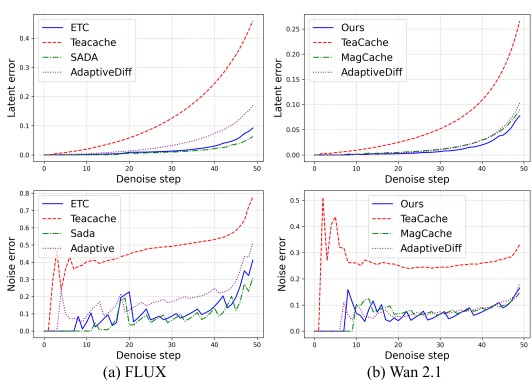

Figure 7: Accumulation of denoising errors.

In contrsast, our approach maintains high acceleration while effectively controlling error accumulation, demonstrating that our trend approximation strategy preserves denoising trajectory consistency during multiple approximations.

Table 2: Variation in speed and generation quality under different expansion threshold settings.

| | | | | | | | | | | |
|---|---|---|---|---|---|---|---|---|---|---|
| **FLUX** | $\sigma$ | 0.4429 | 0.2435 | 0.2263 | 0.1921 | 0.1269 | 0.1027 | 0.0798 | 0.0597 | 0.0392 |
| | Latency (s) ↓ | 8.64 | 9.06 | 9.61 | 9.62 | 10.86 | 11.37 | 11.95 | 13.69 | 14.66 |
| | SSIM ↑ | 0.823 | 0.851 | 0.887 | 0.893 | 0.926 | 0.927 | 0.942 | 0.965 | 0.980 |
| | CLIP ↑ | 25.679 | 25.437 | 25.817 | 25.769 | 25.983 | 26.281 | 26.169 | 26.323 | 26.494 |
| **Wan 2.1** | $\sigma$ | 0.4225 | 0.2762 | 0.2102 | 0.1523 | 0.1181 | 0.0869 | 0.0761 | 0.0678 | 0.0487 |
| | Latency (s) ↓ | 293.76 | 293.78 | 342.91 | 369.97 | 389.89 | 496.15 | 505.88 | 536.82 | 538.68 |
| | SSIM ↑ | 0.637 | 0.637 | 0.708 | 0.762 | 0.806 | 0.821 | 0.843 | 0.889 | 0.912 |
| | VBench ↑ | 80.61% | 80.61% | 80.25% | 80.56% | 80.71% | 80.72% | 80.78% | 80.95% | 81.10% |

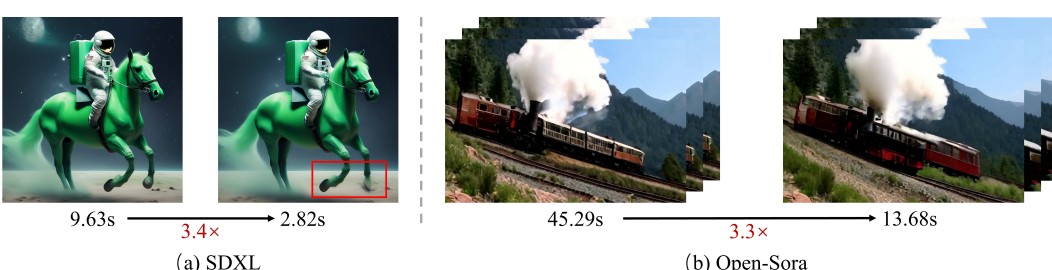

9.63s ——3.4×—→ 2.82s    45.29s ——3.3×—→ 13.68s

(a) SDXL    (b) Open-Sora

Figure 8: Even with less error control for faster acceleration, our method preserves overall structural consistency, with only detail variations.

**Effectiveness of threshold selection** As shown in Table 2, we evaluate the acceleration performance of our method under different expansion thresholds. When accelerating different base models, generation quality declines sharply before the optimal threshold but stabilizes afterward. This trend matches the denoising deviation patterns observed in our threshold search, showing that our method effectively captures each model's tolerance to deviation. Moreover, our method achieves higher SSIM than those obtained during threshold search at high thresholds, indicating that our smoothed trend estimation helps counteract the instability of denoising under high thresholds.

**Fixed error threshold** Table 3 shows the generation results when the error threshold searched from Open-Sora is applied as a fixed threshold across different models. As the gap from the optimal threshold varies, different models exhibit varying degrees of quality degradation. Therefore, employing model-specific error thresholds ensures consistent performance of ETC across different models. Furthermore, as shown in Table 2 and Figure 8, when employing larger error thresholds for over 3× acceleration, our method still maintains reasonable similarity (SSIM > 0.65). This demonstrates that even with less error control, our consistent trend estimation preserves the overall structural integrity of the generated results.

Table 3: Generated results using a fixed error threshold.

| Model | SDXL | FLUX | Open-Sora 1.2 | Wan 2.1 |
|---|---|---|---|---|
| SSIM↑ | 0.742 | 0.891 | 0.849 | 0.719 |
| Latency (s)↓ | 2.78 | 9.58 | 21.07 | 348.72 |
| Speedup↑ | 3.46 | 3.01 | 2.15 | 2.78 |

**Sensitivity of $n$ and $\alpha$** Table 4 presents the SSIM variations under different settings of $n$ and $\alpha$. For the parameter $n$, initiating dilation approximation during early denoising stages with high volatility leads to degraded generation consistency. Conversely, larger $n$ provide more stable initial trend estimation but lead to marginal improvements in SSIM. This indicates that permitting a few initial denoising steps helps establish a more stable estimate of the denoising trajectory. Additionally, the results demonstrate minimal sensitivity to different $\alpha$, with SSIM variations remaining within a narrow range across all tested configurations. This robustness indicates that our trend smoothing design effectively maintains the accelerated denoising trajectory aligned with the original sampling path, preventing significant deviation from the intended generation process.

Table 4: SSIM results at different $n$ and $\alpha$.

| $n$ | 2 | 4 | 6 | 7 | 8 |
|---|---|---|---|---|---|
| FLUX | 0.907 | 0.920 | 0.926 | 0.925 | 0.927 |
| Wan 2.1 | 0.722 | 0.786 | 0.806 | 0.806 | 0.810 |
| $\alpha$ | 0.3 | 0.4 | 0.5 | 0.6 | 0.7 |
| FLUX | 0.923 | 0.925 | 0.926 | 0.927 | 0.921 |
| Wan 2.1 | 0.798 | 0.806 | 0.806 | 0.803 | 0.791 |

**Acceleration performance across different solvers** Since ETC accelerates inference by estimating the model outputs, our approach is compatible with different solvers. To investigate the performance of our method under different solvers, and to examine whether the threshold is universally applicable across solvers, we conduct experiments using multiple solvers for each model.

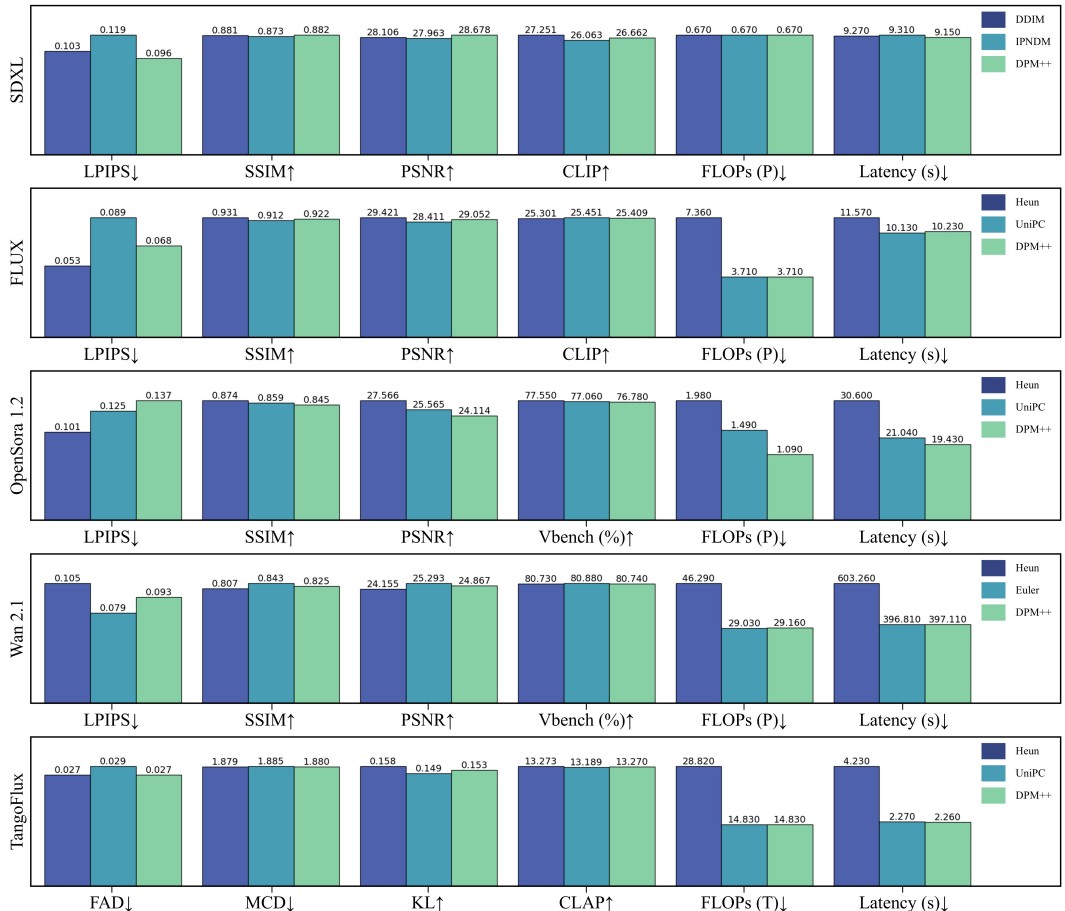

Figure 9: Quantitative results under different solvers.

For SDXL, we evaluate our method using DDIM (Song et al., 2020a), IPNDM (Liu et al., 2022) and DPM++ (Lu et al., 2025) as additional solvers. For FLUX, OpenSora1.2, and TangoFlux, we evaluate our method with Heun (Karras et al., 2022), UniPC (Zhao et al., 2023) and DPM++ as the alternative solvers. For Wan 2.1, we test our method with Euler, Heun and DPM++. All models use the thresholds specified in Table 1 and do not perform additional searches across different solvers.

As shown in Figure 9, our method maintains consistent acceleration and generation quality across different solvers without re-tuning the threshold. This indicates that the threshold we search for captures the error tolerance that prevents trajectory deviation in the model's outputs. Since this property is intrinsic to the model rather than to the choice of solver, the same threshold remains applicable as long as the model itself is unchanged. More experimental details and comprehensive results are provided in Appendix A.8.3.

# 5 CONCLUSION AND DISCUSSION

In this work, we proposed ETC, a training-free diffusion acceleration framework that project all historical model outputs into consistent future trends and distributing them across multiple steps within the model's tolerance limits. Experiments show that ETC preserves generative fidelity while providing substantial speedup. However, a key limitation lies in determining the maximum approximation steps per iteration, where we currently adopt a conservative adjustment strategy based only on the previous round. This restricts the attainable acceleration. A promising direction for future work is to estimate the maximum feasible approximations by evaluating the gap between accumulated errors and model-specific tolerance boundaries.

## 6 ETHICS STATEMENT

This work adheres to the ICLR Code of Ethics. In this study, no human subjects or animal experimentation was involved. All datasets used were sourced in compliance with relevant usage guidelines, ensuring no violation of privacy. We have taken care to avoid any biases or discriminatory outcomes in our research process. No personally identifiable information was used, and no experiments were conducted that could raise privacy or security concerns. We are committed to maintaining transparency and integrity throughout the research process.

## 7 REPRODUCIBILITY STATEMENT

We have made every effort to ensure that the results presented in this paper are reproducible. All code been made publicly available in an anonymous repository to facilitate replication and verification. The experimental setup, including model configurations and hardware details, is described in detail in the paper. We have also provided a full description of ETC, to assist others in reproducing our experiments. We believe these measures will enable other researchers to reproduce our work and further advance the field.

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

# A APPENDIX

## A.1 ERROR ESTIMATION INDUCED BY CONSISTENT TREND PREDICTOR

### A.1.1 A GENERALIZED FORMULA FOR MODEL INFERENCE K STEPS

Suppose we inference once to obtain model output $\epsilon_\theta(x_t, t, c)$ from the latent $x_t$, the denoised formula is as follows:

$$x_{t-1} = f(t-1) \cdot x_t - g(t-1) \cdot \epsilon_\theta(x_t, t, c) \tag{9}$$

Similarly, we can obtain the following formula after the second denoising and substituting Equation 9:

$$\begin{aligned} x_{t-2} &= f(t-2) \cdot x_{t-2} - g(t-2) \cdot \epsilon_\theta(x_{t-1}, t-1, c) \\ &= f(t-2) \cdot f(t-1) \cdot x_t - f(t-2) \cdot g(t-1) \cdot \epsilon_\theta(x_t, t, c) - g(t-2) \cdot \epsilon_\theta(x_{t-1}, t-1, c). \end{aligned} \tag{10}$$

By analogy, we can obtain the following results after sampling k times using the model output:

$$\begin{aligned} x_{t-k} &= \prod_{j=t-k}^{t-1} f(j) \cdot x_t - g(t-k) \cdot \epsilon_\theta(x_{t-k+1}, t-k+1, c) \\ &- \sum_{m=0}^{k-2} \left( \left( \prod_{j=t-k}^{t-k+m} f(j) \right) \cdot g(t-k+m+1) \cdot \epsilon_\theta(x_{t-k+m+2}, t-k+m+2, c) \right). \end{aligned} \tag{11}$$

### A.1.2 CUMULATIVE ERROR CAUSED BY USING ESTIMATED TRENDS FOR THE FIRST TIME

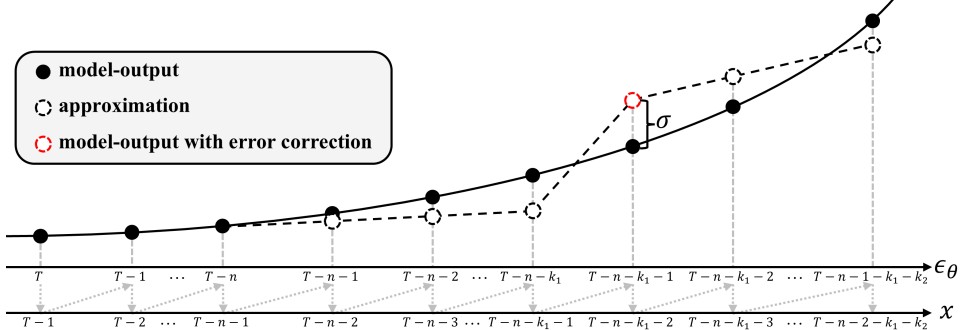

Figure 10: Error accumulation for $k$ times approximation used after model inference $n$ times.

**Assumptions** As shown in Figure 10, we make the assumptions that:

(1) The timestep of the denoising process decreases from T to 0;

(2) Let $d_{t_2}^{t_1} = \epsilon_\theta(x_{t_2}, t_2, c) - \epsilon_\theta(x_{t_1}, t_1, c)$;

(3) Estimated trend is utilized for denoising after performing model inference $n$ times.

**Initial approximate trend** When the model performs inference twice first and then the outputs are used to calculate the estimated trend, the formula is as follows:

$$\Delta_{T-2} = \epsilon_\theta(x_{T-1}, T-1, c) - \epsilon_\theta(x_T, T, c) = d_{T-1}^T. \tag{12}$$

The second estimated trend formulation is as follows:

$$\Delta_{T-3} = (1-\alpha) \cdot \Delta_{T-2} + \alpha \cdot d_{T-2}^{T-1} = (1-\alpha) \cdot d_{T-1}^T + \alpha \cdot d_{T-2}^{T-1}. \tag{13}$$

The third estimated trend formulation is as follows:

$$\begin{aligned} \Delta_{T-4} &= (1-\alpha) \cdot \Delta_{T-3} + \alpha \cdot d_{T-3}^{T-2} = (1-\alpha) \cdot ((1-\alpha) \cdot d_{T-1}^T + \alpha \cdot d_{T-2}^{T-1}) + \alpha \cdot d_{T-3}^{T-2} \\ &= (1-\alpha)^2 \cdot d_{T-1}^T + \alpha \cdot (1-\alpha) \cdot d_{T-2}^{T-1} + \alpha \cdot d_{T-3}^{T-2} \end{aligned} \tag{14}$$

By analogy, we can obtain the following estimated trend for the $n^{th}$ iteration:

$$\Delta_{T-n-1} = (1-\alpha)^{n-1} \cdot d_{T-1}^T + \alpha \sum_{m=0}^{n-2} (1-\alpha)^m \cdot d_{T-n+m}^{T-n+m+1}. \tag{15}$$

**Cumulative error**  To improve readability, we set $\epsilon_\theta(x_t) = \epsilon_\theta(x_t, t, c)$. Starting from $x_{T-n-1}$, the formulation for $x_{T-n-1-k_1}$ based on the estimated trend and Equation 11 is as follows:

$$x'_{T-n-1-k_1} = \prod_{j=T-n-1-k_1}^{T-n-2} f(j) \cdot x_{T-n-1} - g(T-n-1-k_1) \cdot (\epsilon_\theta(x_{T-n}) + \Delta_{T-n-1})$$

$$- \sum_{m=0}^{k_1-2} ((\prod_{j=T-n-1-k_1}^{T-n-1-k_1+m} f(j)) \cdot g(T-n-k_1+m) \cdot (\epsilon_\theta(x_{T-n}) + \frac{k_1-m-1}{k_1}\Delta_{T-n-1})). \tag{16}$$

The cumulative error is as follows:

$$error = \|x_{T-n-1-k_1} - x'_{T-n-1-k_1}\|$$

$$= \|g(T-n-1-k_1) \cdot (\Delta_{T-n-1} - d_{T-n-k_1}^{T-n})$$

$$+ \sum_{m=0}^{k_1-2} ((\prod_{j=T-n-1-k_1}^{T-n-1-k_1+m} f(j)) \cdot g(T-n-k_1+m) \cdot (\frac{k_1-m-1}{k_1}\Delta_{T-n-1} - d_{T-n-k_1+m+1}^{T-n}))\|. \tag{17}$$

Since $f \leq 1$ and $g \leq 1$, we can get the formulation below:

$$error \leq \|(\Delta_{T-n-1} - d_{T-n-k_1}^{T-n}) + (\frac{k_1-m-1}{k_1}\Delta_{T-n-1} - d_{T-n-k_1+m+1}^{T-n})\|$$

$$= \|\sum_{m=0}^{k_1-1} (\frac{k_1-m}{k_1}\Delta_{T-n-1} - d_{T-n-k_1+m}^{T-n})\|. \tag{18}$$

Substituting Equation 15 into Equation 18, we obtain the following upper bound on the error:

$$error \leq \|\sum_{m=0}^{k_1-1} (\frac{(k_1-m)(1-\alpha)^{n-1}}{k_1} \cdot d_{T-1}^T + \frac{\alpha(k_1-m)}{k_1} \sum_{j=0}^{n-2} (1-\alpha)^j \cdot d_{T-n+j}^{T-n+j+1} - d_{T-n-k_1+m}^{T-n})\|. \tag{19}$$

### A.1.3  ERROR ACCUMULATION IN THE NEXT ROUND

**Assumptions**  As shown in Figure 10, we make the assumptions that:

(1) To correct the cumulative error, the model output $\epsilon_\theta^*(x_{T-n-k_1-1}) = \epsilon_\theta(x_{T-n-k_1-1}) + \sigma_1$. After a single model inference, the obtained $x_{T-n-k_1-2}$ exhibits no error.

**Updated approximate trend**  After correcting the cumulative error using a single model inference, we update the formula for estimated trend as follows:

$$\Delta_{T-n-2-k_1} = (1-\alpha) \cdot \Delta_{T-n-1} + \alpha \cdot ((\epsilon_\theta(x_{T-n-k_1-1}) + \sigma_1) - (\epsilon_\theta(x_{T-n}) + \Delta_{T-n-1}))$$

$$= (1-2\alpha) \cdot \Delta_{T-n-1} + \alpha \cdot (d_{T-n-k_1-1}^{T-n} + \sigma_1). \tag{20}$$

**Cumulative error**  To improve readability, we omit the arguments of $f$ and $g$. Starting from $x_{T-n-2-k_1}$, the formulation for $x_{T-n-2-k_1-k_2}$ based on the estimated trend and Equation 11 is:

$$x'_{T-n-2-k_1-k_2} = \prod f \cdot x_{T-n-2-k_1} - g \cdot (\epsilon_\theta(x_{T-n-1-k_1}) + \sigma_1 + \Delta_{T-n-2-k_1})$$

$$- \sum_{m=0}^{k_2-2} ((\prod f) \cdot g \cdot (\epsilon_\theta(x_{T-n-1-k_1}) + \sigma_1 + \frac{k_2-m-1}{k_2}\Delta_{T-n-2-k_1})). \tag{21}$$

The cumulative error is as follows:

$$error = \|x_{T-n-2-k_1-k_2} - x'_{T-n-2-k_1-k_2}\| = \|g \cdot (\sigma_1 + \Delta_{T-n-2-k_1} - d_{T-n-1-k_1-k_2}^{T-n-1-k_1})$$

$$+ \sum_{m=0}^{k_2-2} ((\prod f) \cdot g \cdot (\sigma_1 + \frac{k_2-m-1}{k_2}\Delta_{T-n-2-k_1} - d_{T-n-k_1-k_2+m}^{T-n-1-k_1}))\|. \tag{22}$$

Since $f \leq 1$ and $g \leq 1$, we can get the formulation below:

$$error \leq \|(\sigma_1 + \Delta_{T-n-2-k_1} - d_{T-n-1-k_1-k_2}^{T-n-1-k_1}) + (\sigma_1 + \frac{k_2 - m - 1}{k_2}\Delta_{T-n-2-k_1} - d_{T-n-k_1-k_2+m}^{T-n-1-k_1})\|$$

$$= \|\sum_{m=0}^{k_2-1}(\sigma_1 + \frac{k_2 - m}{k_2}\Delta_{T-n-2-k_1} - d_{T-n-1-k_1-k_2+m}^{T-n-1-k_1})\|. \tag{23}$$

Substituting Equation 20 into Equation 23, we obtain the following upper bound on the error:

$$error \leq \|\sum_{m=0}^{k_2-1}(\frac{(k_2 - m)(1 - 2\alpha)(1 - \alpha)^{n-1}}{k_2} \cdot d_{T-1}^T + \frac{\alpha(k_2 - m)(1 - 2\alpha)}{k_2}\sum_{j=0}^{n-2}(1 - \alpha)^j \cdot d_{T-n+j}^{T-n+j+1}$$

$$+ \frac{\alpha(k_2 - m)}{k_2} \cdot d_{T-n-k_1-1}^{T-n} + \frac{k_2 + \alpha(k_2 - m)}{k_2} \cdot \sigma_1 - d_{T-n-1-k_-k_2+m}^{T-n-1-k_1})\|. \tag{24}$$

### A.1.4 ACCUMULATION OF ERRORS THROUGHOUT THE PROCESS

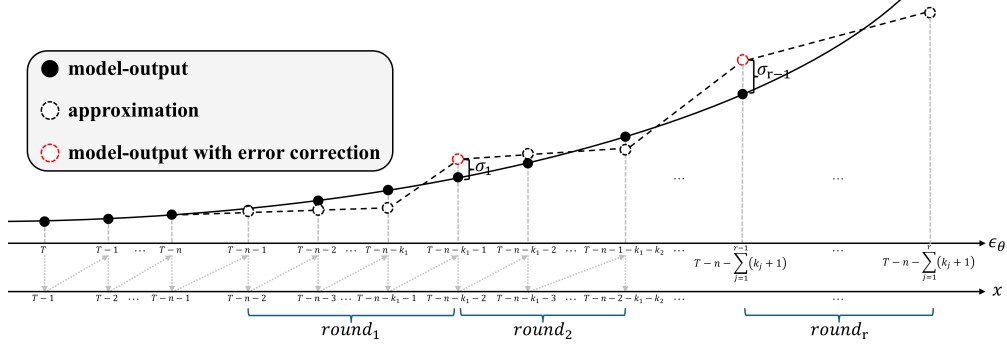

Figure 11: Error accumulation throughout the denoising process.

**Assumptions** As shown in Figure 11, we make the assumptions that:

(1) A total of $r$ rounds of approximations were performed, with $k_i$ approximations per round, and the difference between the outputs obtained using model inference and the model outputs under no error for each round is $\sigma_{i-1}$, where $i = 1, ..., r$.

**Approximate trend at each round** By analogy, we can obtain the following formulation for the estimated trend used in the $r^{th}$ round:

$$\Delta_{T-n-1-\sum_{m=1}^{r-1}(k_m+1)} = (1 - 2\alpha)^{r-1}\Delta_{T-n-1} + \alpha\sum_{m=1}^{r-1}((1 - 2\alpha)^{r-m-1}(d_{T-n-\sum_{j=1}^{m}k_j+1}^{T-n-\sum_{j=1}^{m-1}(k_j+1)} + \sigma_m)) \tag{25}$$

**Cumulative error** By analogy, we can obtain the following formulation for the cumulative error in the $r^{th}$ round:

$$error_r \leq \|\sum_{m=0}^{k_r-1}(\sigma_{r-1} + \frac{k_r - m}{k_r}\Delta_{T-n-1-\sum_{j=1}^{r-1}(k_j+1)} - d_{T-n-\sum_{j=1}^{r-1}(k_j+1)-k_r+m}^{T-n-\sum_{j=1}^{r-1}(k_j+1)})\|$$

$$= \|\sum_{m=0}^{k_r-1}(\sigma_{r-1} + \frac{(k_r - m)(1 - 2\alpha)^{r-1}}{k_r}\Delta_{T-n-1} \tag{26}$$

$$+ \frac{\alpha(k_r - m)}{k_r}\sum_{j=1}^{r-1}((1 - 2\alpha)^{r-j-1}(d_{T-n-\sum_{l=1}^{j}(k_l+1)}^{T-n-\sum_{l=1}^{j-1}(k_l+1)} + \sigma_j)) - d_{T-n-\sum_{j=1}^{r-1}(k_j+1)-k_r+m}^{T-n-\sum_{j=1}^{r-1}(k_j+1)})\|$$

Substituting Equation 15 into Equation 26, we obtain the following upper bound on the error:

$$error_r \leq \| \sum_{m=0}^{k_r-1} ((\sigma_{r-1} + \frac{\alpha(k_r - m)}{k_r} \sum_{j=1}^{r-1} (1 - 2\alpha)^{r-j-1} \cdot \sigma_j)$$

$$+ (\frac{(k_r - m)(1 - 2\alpha)^{r-1}(1 - \alpha)^{n-1}}{k_r} \cdot d_{T-1}^T + \frac{\alpha(k_r - m)(1 - 2\alpha)^{r-1}}{k_r} \sum_{j=0}^{n-2} (1 - \alpha)^j \cdot d_{T-n+j}^{T-n+j+1}$$

$$+ \frac{\alpha(k_r - m)}{k_r} \sum_{j=1}^{r-1} (1 - 2\alpha)^{r-j-1} \cdot d_{T-n-\sum_{l=1}^{j}(k_l+1)}^{T-n-\sum_{l=1}^{j-1}(k_l+1)} - d_{T-n-\sum_{j=1}^{m}(k_j+1)-k_r+m}^{T-n-\sum_{j=1}^{m-1}(k_j+1)})\|$$

$$(27)$$

As can be seen, we approximate the future trend at each step by a weighted combination of the historical trend and the error-correction.

## A.2 TRAJECTORY CONTINUITY OF DIFFUSION PROCESSES

### A.2.1 CONTINUITY OF THE FORWARD PROCESS

By the definition of the forward process in diffusion models, the variable at timestep $t$ is given by:

$$x_t = \alpha(t) \cdot x_0 + \beta(t) \cdot \epsilon, \tag{28}$$

where $\alpha(t)$ and $\beta(t)$ are noise scheduling coefficients. Taking expectation over $x_0 \sim p(x_0)$, $\epsilon \sim \mathcal{N}(0, I)$, and $t \sim \text{Uniform}(1, ..., T)$, we get:

$$\mathbb{E}_{x_0, \epsilon, t} = \mathbb{E}[\mathbb{E}_{x_0}[\alpha(t) \cdot x_0] + \mathbb{E}_\epsilon[\beta(t) \cdot \epsilon]]. \tag{29}$$

Since $\epsilon \sim \mathcal{N}(0, I)$, we have $\mathbb{E}[\epsilon] = 0$, and thus:

$$\mathbb{E}_{x_0, \epsilon, t} = \mathbb{E}_t[\alpha(t) \cdot \mathbb{E}_{x_0}[x_0]] = \alpha(t) \cdot \mathbb{E}_{x_0}[x_0]. \tag{30}$$

From Equation 30, we know that the trajectory $x_t$ is continuous in expectation. Consequently, the empirical average $x_t$ also exhibits continuity by the law of large numbers (Feller et al., 1971).

### A.2.2 CONTINUITY OF THE BACKWARD PROCESS

**Different mathematical definitions of the denoising process** The denoising process can be defined through various mathematical frameworks. Stochastic differential equation (SDE) (Song et al., 2020b) defines the process as follows:

$$dx = \left[ -\frac{1}{2}\beta(t)x - \nabla_{x_t} \log p_t(x_t) \right] dt + \sqrt{\beta(t)} d\bar{w}, \tag{31}$$

where $\nabla_{x_t} \log p_t(x_t)$ denotes the log-likelihood gradient and $d\bar{w}$ represents the standard Wiener process. Ordinary differential equation (ODE) removes the term $d\bar{w}$ from Equation 31, offering a more stable denoising process that mitigates fluctuations inherent in stochastic processes. Flow matching (Lipman et al., 2022) proposes a deterministic approach to denoising by learning data flows, mathematically described as follows:

$$dx = v(x_t, t)dt, \tag{32}$$

where $v(x_t, t)$ represents the flow function that governs the evolution of data toward the target distribution over time.

**$\epsilon$-prediction in sde and ode.** The noise prediction model (Ho et al., 2020; Song et al., 2020a) is trained to estimate the noise contained in $x_t$, which is proved to be equivalent to estimating the log-likelihood gradient by Ho et al. (2020). Given the origin data $x_0$, the forward noising process is defined as:

$$x_t = \sqrt{\bar{\alpha}_t} x_0 + \sqrt{1 - \bar{\alpha}_t} \epsilon, \quad \bar{\alpha}_t = \prod_{i=1}^{t} (1 - \beta_i), \tag{33}$$

where $\beta \in (0, 1)$ and $\epsilon \sim N(0, I)$. Then the network $\epsilon_\theta(x, t)$ is trained with the standard mean-squared error (MSE) objective:

$$\mathcal{L}(\theta) = \mathbb{E}_{x_0, \epsilon, t}[||\epsilon - \epsilon_\theta(x_t, t)||^2], \quad x_t = \sqrt{\bar{\alpha}_t} x_0 + \sqrt{1 - \bar{\alpha}_t} \epsilon. \tag{34}$$

During the denoising stage, the next time-step $x_{t-1}$ can be obtained from $x_t$ as:

$$x_{t-1} = \sqrt{\bar{\alpha}_{t-1}}\left(\frac{x_t - \sqrt{1-\bar{\alpha}_t}\epsilon_\theta(x_t,t)}{\sqrt{\bar{\alpha}_t}}\right) + \sqrt{1-\bar{\alpha}_t}\epsilon_\theta(x_t,t). \tag{35}$$

Taking expectation over $x_t \sim p(x_t)$ and $t \sim$ Uniform$(1, ..., T)$, we get:

$$\mathbb{E}_{x_t,\epsilon_\theta,t} = \mathbb{E}\left[\mathbb{E}_{x_t}\left[\frac{\sqrt{\bar{\alpha}_{t-1}}}{\sqrt{\bar{\alpha}_t}}x_t\right] + \mathbb{E}_{\epsilon_\theta}\left[\sqrt{1-\bar{\alpha}_t}\left(1 - \frac{\sqrt{\bar{\alpha}_{t-1}}}{\sqrt{\bar{\alpha}_t}}\right)\epsilon_\theta(x_t,t)\right]\right]$$
$$= \frac{\sqrt{\bar{\alpha}_{t-1}}}{\sqrt{\bar{\alpha}_t}} \cdot \mathbb{E}_{x_t}[x_t] + \sqrt{1-\bar{\alpha}_t}\left(1 - \frac{\sqrt{\bar{\alpha}_{t-1}}}{\sqrt{\bar{\alpha}_t}}\right) \cdot \mathbb{E}_{\epsilon_\theta}[\epsilon_\theta(x_t,t)] \tag{36}$$

Suppose $\theta$ minimizes $\mathcal{L}$, and the training is sufficiently converged, then $\epsilon_\theta(x_t,t) \approx \epsilon \sim \mathcal{N}(0,I)$. Since $\epsilon_\theta \sim \mathcal{N}(0,I)$, we have $\mathbb{E}[\epsilon_\theta] = 0$, and thus:

$$\mathbb{E}_{x_t,\epsilon_\theta,t} = \mathbb{E}_t\left[\frac{\sqrt{\bar{\alpha}_{t-1}}}{\sqrt{\bar{\alpha}_t}} \cdot \mathbb{E}_{x_t}[x_t]\right] = \frac{\sqrt{\bar{\alpha}_{t-1}}}{\sqrt{\bar{\alpha}_t}} \cdot \mathbb{E}_{x_t}[x_t] \tag{37}$$

**v-prediction in flow matching.** Flow matching defines a vector field $v$ that guides the data from the initial distribution $x_0$ toward the target distribution $x_T$. Specifically, the linear interpolation path from $x_0$ to $x_T$ is:

$$x_t = (1 - \frac{t}{T})x_0 + \frac{t}{T}x_T, \quad x_T \sim \mathcal{N}(0,I), \quad t \sim \text{Uniform}(1, ..., T). \tag{38}$$

The vector field $v$ is defined as:

$$v = \frac{dx}{dt} = -x_0 + x_T = x_T - x_0. \tag{39}$$

Then the network $v_\theta(x,t)$ is trained with the standard mean-squared error (MSE) objective:

$$\mathcal{L}(\theta) = \mathbb{E}_{x_0,x_T,v,t}[||v - v_\theta(x_t,t)||^2], \quad v = x_T - x_0. \tag{40}$$

During the denoising stage, the next time-step $x_{t-1}$ can be obtained from $x_t$ as:

$$x_{t-1} = x_t - \left(\frac{t}{T} - \frac{t-1}{T}\right)v_\theta(x_t,t) = x_t - \frac{1}{T}v_\theta(x_t,t), \tag{41}$$

Taking expectation over $x_t \sim p(x_t)$ and $t \sim$ Uniform$(1, ..., T)$, we get:

$$\mathbb{E}_{x_t,v_\theta,t} = \mathbb{E}\left[\mathbb{E}_{x_t}[x_t] - \mathbb{E}_{v_\theta}\left[\frac{1}{T}v_\theta\right]\right] = \mathbb{E}_{x_t}[x_t] - \frac{1}{T}\mathbb{E}_{v_\theta}[v_\theta(x_t,t)] \tag{42}$$

Suppose $\theta$ minimizes $\mathcal{L}$, and the training is sufficiently converged, then $v_\theta(x_t,t) \approx v$.

$$\mathbb{E}_{x_t,v_\theta,t} = \mathbb{E}\left[\mathbb{E}_{x_t}[x_t] - \mathbb{E}_{v_\theta}\left[\frac{1}{T}v_\theta\right]\right] = \mathbb{E}\left[\mathbb{E}_{x_t}[x_t] - \mathbb{E}\left[\frac{1}{T}(x_T - x_0)\right]\right]$$
$$= \mathbb{E}\left[\mathbb{E}_{x_t}[x_t] - \mathbb{E}\left[\frac{1}{T}\left(x_T - \frac{T}{T-t}(x_t - \frac{t}{T}x_T)\right)\right]\right] \tag{43}$$

Since $x_T \sim \mathcal{N}(0,I)$, we have $\mathbb{E}[x_T] = 0$, and thus:

$$\mathbb{E}_{x_t,v_\theta,t} = \mathbb{E}\left[\mathbb{E}_{x_t}[x_t] + \mathbb{E}_{x_t}\left[\frac{1}{T-t}x_t\right]\right] = \mathbb{E}_t\left[(1 + \frac{1}{T-t})\mathbb{E}_{x_t}[x_t]\right] = \frac{T-t+1}{T-t} \cdot \mathbb{E}_{x_t}[x_t]. \tag{44}$$

From Equation 37 and Equation 44, a well-trained denoiser permits the sampling trajectory from $x_t$ to $x_0$ continuous in expectation. Consequently, the empirical average $x_t$ also exhibits continuity by the law of large numbers (Feller et al., 1971).

However, as shown in Figure 12 (b) and (c), the mean of the model's output varies across timesteps during inference. This occurs because different timesteps correspond to different noise levels and contain different amounts of information, leading to variations in the predicted values. Consequently, as implied by Equations Equation 36 and Equation 42, under a fixed scheduler, the differences in the denoising trajectory are determined by the model $\theta$.

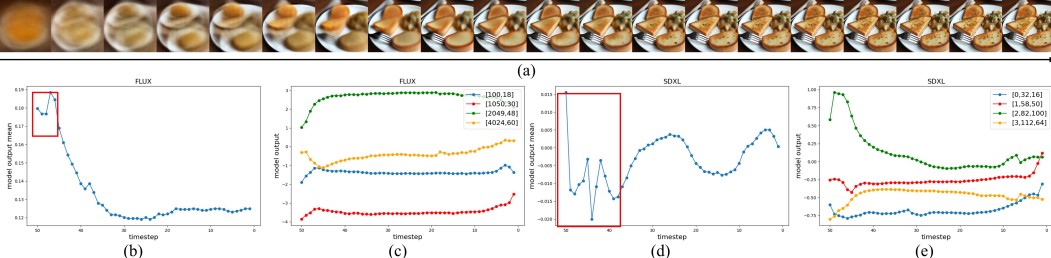

Figure 12: Visualization of fluctuation and stabilization phases during denoising. Subfigure (a) shows the results obtained at each timestep by directly sampling $x_0$ from model outputs. Subfigures (b) and (d) present the mean output curves of FLUX and SDXL across timesteps. Subfigures (c) and (e) show the output trajectories at four selected latent points for FLUX and SDXL, respectively.

### A.3 FLUCTUATION AND STABILIZATION PHASES OF MODEL OUTPUTS DURING DENOISING

Figure 12(a) visualizes the denoising process during the semantic planning phase and the fidelity refinement phase. At the beginning of the denoising process, the model gradually constructs the main semantic components of the image, for example, first the plate appears, then the bread. Once the global structure aligns with the semantic intent, subsequent denoising shifts toward refining details, such as adding scallions on the bread.

As analyzed in A.2.2, the evolution of the denoising process is determined by the model outputs. Therefore, we visualize the timestep-wise mean of the model outputs for SDXL ($\epsilon$-prediction) and FLUX ($v$-prediction). As shown in Figure 12(b) and (d), the mean outputs exhibit pronounced fluctuations in the early denoising steps and then gradually stabilize, which is consistent with the behavior observed in Figure 12(a).

Figure 12(c) and (e) further plot the output differences between adjacent timesteps at four selected latent positions for FLUX and SDXL. In the refinement phase, these differences become stable, indicating a consistent output trend. This motivates us to apply approximation-based acceleration during this stable stage.

### A.4 PROOF OF LIPSCHITZ CONTINUITY OF $\epsilon_\theta$

Based on Equation 28, we obtain the following formulation for the noise at timestep $t$:

$$\epsilon = \frac{1}{\beta(t)}x_t - \frac{\alpha(t)}{\beta(t)}x_0. \tag{45}$$

Assuming that under the same conditions $c$, the corresponding $x_0$ remains identical across timesteps, and the trained model's outputs $\epsilon_\theta(x_t, t, c)$ at different timesteps follow Equation 45, we obtain the following formulation for the model output difference between timestep $t$ and timestep $t'$:

$$\begin{aligned}
\epsilon_\theta(x_t, t, c) - \epsilon_\theta(x_{t'}, t', c) &= \frac{x_t - \alpha(t)x_0}{\beta(t)} - \frac{x_{t'} - \alpha(t')x_0}{\beta(t')} \\
&= \frac{x_t - x_{t'}}{\beta(t)} + \frac{x_{t'} - \alpha(t')x_0}{\beta(t)} - \frac{x_{t'} - \alpha(t')x_0}{\beta(t')} + \frac{-\alpha(t)x_0 + \alpha(t')x_0}{\beta(t)} \\
&= \frac{1}{\beta(t)}(x_t - x_{t'}) + (\frac{1}{\beta(t)} - \frac{1}{\beta(t')})(x_{t'} - \alpha(t')x_0) + \frac{\alpha(t') - \alpha(t)}{\beta(t)}x_0
\end{aligned} \tag{46}$$

Take an upper bound of the norm:

$$\begin{aligned}
||\epsilon_\theta(x_t, t, c) - \epsilon_\theta(x_{t'}, t', c)|| &\leq \frac{1}{\beta(t)}||x_t - x_{t'}|| \\
&+ ||x_{t'} - \alpha(t')x_0|| \, |\frac{1}{\beta(t)} - \frac{1}{\beta(t')}| + ||x_0||\frac{|\alpha(t') - \alpha(t)|}{|\beta(t)|}
\end{aligned} \tag{47}$$

Since $\alpha$ and $\beta$ vary with time $t$ within the interval $[0, 1]$, both $\alpha$ and $\beta$ are Lipschitz continuous:

$$|\alpha(t) - \alpha(t')| \leq K_\alpha |t - t'|, \quad |\frac{1}{\beta(t)} - \frac{1}{\beta(t')}| \leq K_\beta |t - t'|. \tag{48}$$

To avoid potential issues with infinite Lipschitz (Yang et al., 2023) constants near the boundaries, we skip the last time steps. Therefore, $\beta$ has a lower bound that $\beta(t) \geq \beta_{min} > 0$. We can get the formulation below:

$$
\begin{aligned}
||\epsilon_\theta(x_t, t, c) - \epsilon(x_{t'}, t', c)|| &\leq \frac{1}{\beta_{min}}||x_t - x_{t'}|| \\
&+ ||x_{t'} - \alpha(t')x_0||K_\beta|t - t'| + ||x_0||\frac{K_\alpha}{\beta_{min}}|t - t'|)
\end{aligned}
\tag{49}
$$

In generation tasks, $x_0$ is often bounded (e.g., in image generation tasks, pixel values are normalized between 0 and 1), thus we defind $x_0 \leq K_0$. Since $x_t'$ lies along the denoising path from $x_T$ to $x_0$, it follows that $||x_{t'} - f(t')x_0|| \leq ||x_T - x_0||$. Furthermore, since $x_T \sim \mathcal{N}(0, I)$, we also have $||x_T - x_0|| \leq K_1$. We can get the formulation:

$$
||\epsilon_\theta(x_t, t, c) - \epsilon_\theta(x_{t'}, t', c)|| \leq L_x||x_t - x_{t'}|| + L_t||t - t'||, L_x = \frac{1}{\beta_{min}}, L_t = K_1K_\beta + \frac{K_0K\alpha}{\beta_{min}}
\tag{50}
$$

Therefore, the model output $\epsilon_\theta(x_t, t, c)$ is jointly Lipschitz continuous in both $x_t$ and $t$, exhibiting structured and predictable variations across time steps.

## A.5 ACCUMULATED ERROR WHEN EACH TIMESTEP INTRODUCES A FIXED ERROR

Assuming that a fixed error $\beta$ is introduced at every timestep. The accumulated error after a single sampling step can then be expressed as follows:

$$
x'_{T-1} = f(T-1) \cdot x_T - g(T-1) \cdot (\epsilon_\theta(x_T, T, c) - \beta)
\tag{51}
$$

Similarly, we can obtain the following formula after the second sampling using the approximated noise and substituting Equation 11:

$$
\begin{aligned}
x'_{T-2} &= f(T-2) \cdot x'_{T-1} - g(T-2) \cdot (\epsilon_\theta(x_{T-2}, T-2, c) - \beta) \\
&= f(T-2) \cdot f(T-1) \cdot x_T - f(T-2) \cdot g(T-1) \cdot \epsilon_\theta(x_T, T, c) \\
&+ (f(T-2) \cdot g(T-1) + g(T-2)) \cdot \beta - g(T-2) \cdot \epsilon_\theta(x_{T-1}, T-1, c).
\end{aligned}
\tag{52}
$$

By analogy, we can obtain the following results after sampling T times:

$$
\begin{aligned}
x'_0 &= \prod_{j=0}^{T-1} f(j) \cdot x_T - g(0) \cdot \epsilon_\theta(x_1, 1, c) - \sum_{m=0}^{T-2} \left( \left( \prod_{j=0}^{T-2-m} f(j) \right) \cdot g(T-1-m) \cdot \epsilon_\theta(x_{T-m}, T-m, c) \right) \\
&+ \underbrace{\sum_{m=0}^{T-2} \left( \left( \prod_{j=0}^{T-2-m} f(j) \right) \cdot g(T-1-m) \cdot \beta \right) + g(0) \cdot \beta}_{cumulative\ error}.
\end{aligned}
\tag{53}
$$

It can be seen that there is an additional error accumulation term on top of Equation 11 obtained by normal denoising. The error at each step is amplified throughout the sampling process. Therefore, we must ensure that the accumulated error within a single approximation round does not cause the trajectory to deviate; otherwise, the deviation would become even larger after multiple rounds of accumulation.

## A.6 ALGORITHM OF CONSISTENT TREND PREDICTOR

Algorithm 1 illustrates the workflow of our consistent trend predictor. The consistent trend predictor is applied during the stable phase of the denoising process, leveraging the smooth continuity of diffusion trajectories, projecting historical de- noising patterns into stable future directions and progressively distributing them across multiple approximation steps to achieve acceleration without deviating.

---

**Algorithm 1** Consistent Trend Predictor.

---

**Input:** Diffusion Model $\epsilon_\theta$, Sampling Scheduler $\phi$, Decoder $D$, Sample Step $T$, Conditional Embedding $c$,
  Pre-Inference step $n$, Smoothing Factor $\alpha$, Error Threshold $\sigma$;
1: Initialize Random Noise $x$, Future Trend $\Delta = $ None, Approximation Step $k = 0$, Previous Model Output
  $P = $ None.
2: **for** $t = T$ to $T - n$ **do**
3:     **if** $t = T - 1$ **then**
4:         $\Delta = \epsilon_\theta(x, t, c) - P$;
5:     **end if**
6:     **if** $t < T - 1$ **then**
7:         $\Delta = (1 - \alpha)\Delta + \alpha(\epsilon_\theta(x, t, c) - P)$;
8:     **end if**
9:     $P = \epsilon_\theta(x, t, c)$;
10:     Compute $\phi(x, \epsilon_\theta(x, t, c))$ by Eq. (1);
11: **end for**
12: **while** $t > 1$ **do**
13:     **for** $j = 1$ to $k$ **do**
14:         $\epsilon_\theta^{'}(x, t - j, c) \leftarrow P + \frac{\Delta}{k}$;
15:         $P = \epsilon_\theta^{'}(x, t - j, c)$;
16:         Compute $\phi(x, \epsilon_\theta^{'}(x, t - j, c))$ by Eq. (1);
17:     **end for**
18:     $t \leftarrow t - k - 1$;
19:     **if** $\epsilon_\theta(x, t, c) - P - \Delta \geq \sigma$ **then**
20:         **if** $k > 0$ **then**
21:             $k = k - 1$;
22:         **end if**
23:     **end if**
24:     **if** $\epsilon_\theta(x, t, c) - P - \Delta < \sigma$ **then**
25:         $k = k + 1$;
26:     **end if**
27:     $\Delta = (1 - \alpha)\Delta + \alpha(\epsilon_\theta(x, t, c) - P)$;
28:     $P = \epsilon_\theta(x, t, c)$;
29:     Compute $\phi(x, \epsilon_\theta(x, t, c))$ by Eq. (1);
30: **end while**
31: Compute $\phi(x, \epsilon_\theta(x, t - 1, c))$ by Eq. (1);
32: **return** $D(x)$.

---

## A.7 ALGORITHM OF CONSISTENT TREND PREDICTOR

---

**Algorithm 2** Model-Specific Error Tolerance Search Mechanism.

---

**Input:** Diffusion Model $\epsilon_\theta$, Sampling Scheduler $\phi$, Decoder $D$, Sample Step $T$, Conditional Embedding Set
  $S_c$, Trend Inflection Point Analysis Model $M$, Similarity Metrics $Sim$;
1: Initialize Similarity List S=[0]*$(T - 1)$.
2: **for** $c$ in $S_c$ **do**
3:     Initialize Random Noise $x$, Previous Output $P = $ None, Model Output Differences List $L$=[];
4:     **for** $t = T$ to $0$ **do**
5:         **if** $t < T$ **then**
6:             $L$.append($(\epsilon_\theta(x, t, c) - P).abs().mean()$);
7:         **end if**
8:         $P = \epsilon_\theta(x, t, c)$;
9:         Compute $\phi(x, \epsilon_\theta(x, t, c))$ by Eq. (1);
10:     **end for**
11:     **for** $i = 0$ to $T - 1$ **do**
12:         Initialize Random Noise $x^{'}$;
13:         $x^{'} = x^{'}/x^{'}.mean() * L[i]$;
14:         S[i] = S[i]+$Sim(D(x + x^{'}), D(x))$;
15:     **end for**
16: **end for**
17: S = S/$(T$-1$)$;
18: $\sigma = M$(S); **return** $\sigma$

---

Table 5: Quantitative results compared with intra-layer feature reuse methods. Best performance in **bold**, and the second best underlined.

| Method | Visual Quality | | | | Efficiency | | |
|---|---|---|---|---|---|---|---|
| **Text to Image** | **LPIPS ↓** | **SSIM ↑** | **PSNR ↑** | **CLIP ↑** | **FLOPs (P) ↓** | **Speedup ↑** | **Latency (s) ↓** |
| SDXL ($T = 50$) | - | - | - | 27.253 | 0.67 | 1× | 9.63 |
| ETC ($\sigma = 0.0171$) | **0.105** | **0.876** | **28.017** | 27.103 | 0.26 | 2.12× | 4.53 |
| DeepCache | 0.205 | 0.791 | 23.871 | **27.055** | 0.25 | 2.17× | 4.33 |
| DeepCache+ETC | 0.231 | 0.782 | 22.547 | 27.049 | **0.14** | **4.29×** | **2.24** |
| FLUX ($T = 50$) | - | - | - | 26.319 | 3.64 | 1× | 28.85 |
| ETC | **0.068** | **0.926** | **29.176** | 25.983 | 1.31 | 2.65× | 10.86 |
| ToCa | 0.322 | 0.694 | 18.387 | 25.286 | 1.86 | 1.88× | 15.31 |
| ToCa+ETC | 0.351 | 0.687 | 18.187 | 25.711 | **1.01** | **3.43×** | **8.41** |
| **Text to Video** | **LPIPS ↓** | **SSIM ↑** | **PSNR ↑** | **VBench ↑** | **FLOPs (P) ↓** | **Speedup ↑** | **Latency (s) ↓** |
| Open-Sora 1.2 ($T = 30$) | - | - | - | 77.18% | 3.15 | 1× | 45.29 |
| Ours ($\sigma = 0.2101$) | **0.131** | **0.848** | **24.123** | 76.44% | 1.32 | 2.15× | 21.07 |
| FasterCache | 0.166 | 0.793 | 23.282 | 76.37% | 2.08 | 1.58× | 28.67 |
| FasterCache+ETC | 0.168 | 0.802 | 23.389 | 76.38% | **1.03** | **2.62×** | **17.28** |

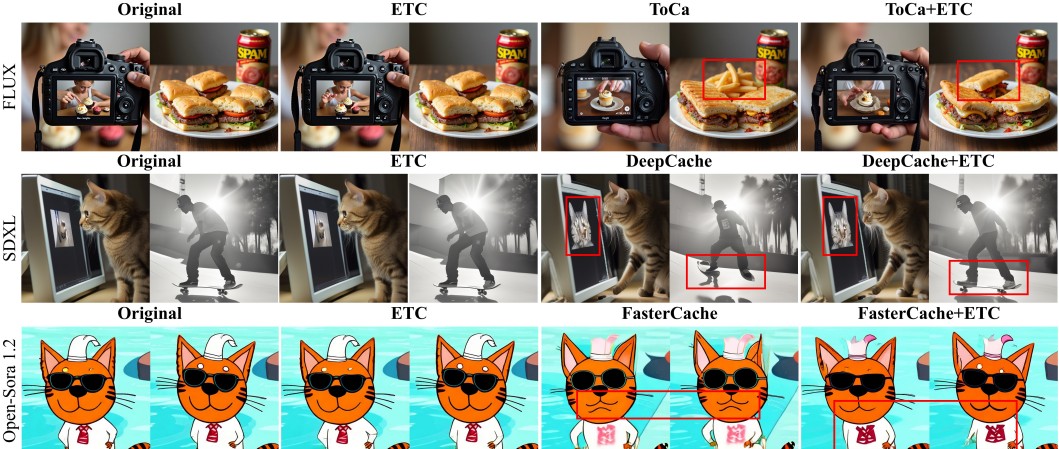

Figure 13: Comparison of visual quality with the intra-layer feature reuse method.

## A.8 MORE EXPERIMENTAL RESULTS

### A.8.1 COMPARED WITH INTRA-LAYER FEATURE REUSE METHODS

For SDXL, we adopt DeepCache (Ma et al., 2024b), a feature-reuse method designed for U-Net (Ronneberger et al., 2015) that accelerates inference by reusing model features across down-sampling and upsampling blocks. For FLUX, we employ ToCa (Zou et al., 2024), which analyzes token-level importance within DiT (Peebles & Xie, 2023) architectures and reuses low-impact tokens to reduce the computational cost of attention. For Open-Sora, we leverage FasterCache (Lv et al., 2024), a method that accelerates inference by dynamically reusing features within attention modules.For Wan and TangoFlux, although the relevant acceleration techniques are theoretically applicable, no compatible implementations are currently available. Therefore, we do not include experiments for these models. Additionally, our method is compatible with intra-layer feature-reuse approaches, and we include comparative experiments to demonstrate this compatibility.

As shown in Figure 13, our method achieves higher acceleration as well as better generation consistency and quality compared with intra-layer feature–reuse approaches. Furthermore, when combined with intra-layer feature reuse, our method provides additional speedup, while the quality degradation is minimal relative to using intra-layer reuse alone. In some cases, the quality even improves, for example, the structural restoration of a person on a skateboard.

### A.8.2 COMPARED WITH TRAINING-BASED FEW STEP MODELS

We also compare our method with trained few-step models. For SDXL, we use SDXL-Lightning (Lin et al., 2024), and for FLUX, we use Flux-Turbo (Alibaba, 2024). Since widely

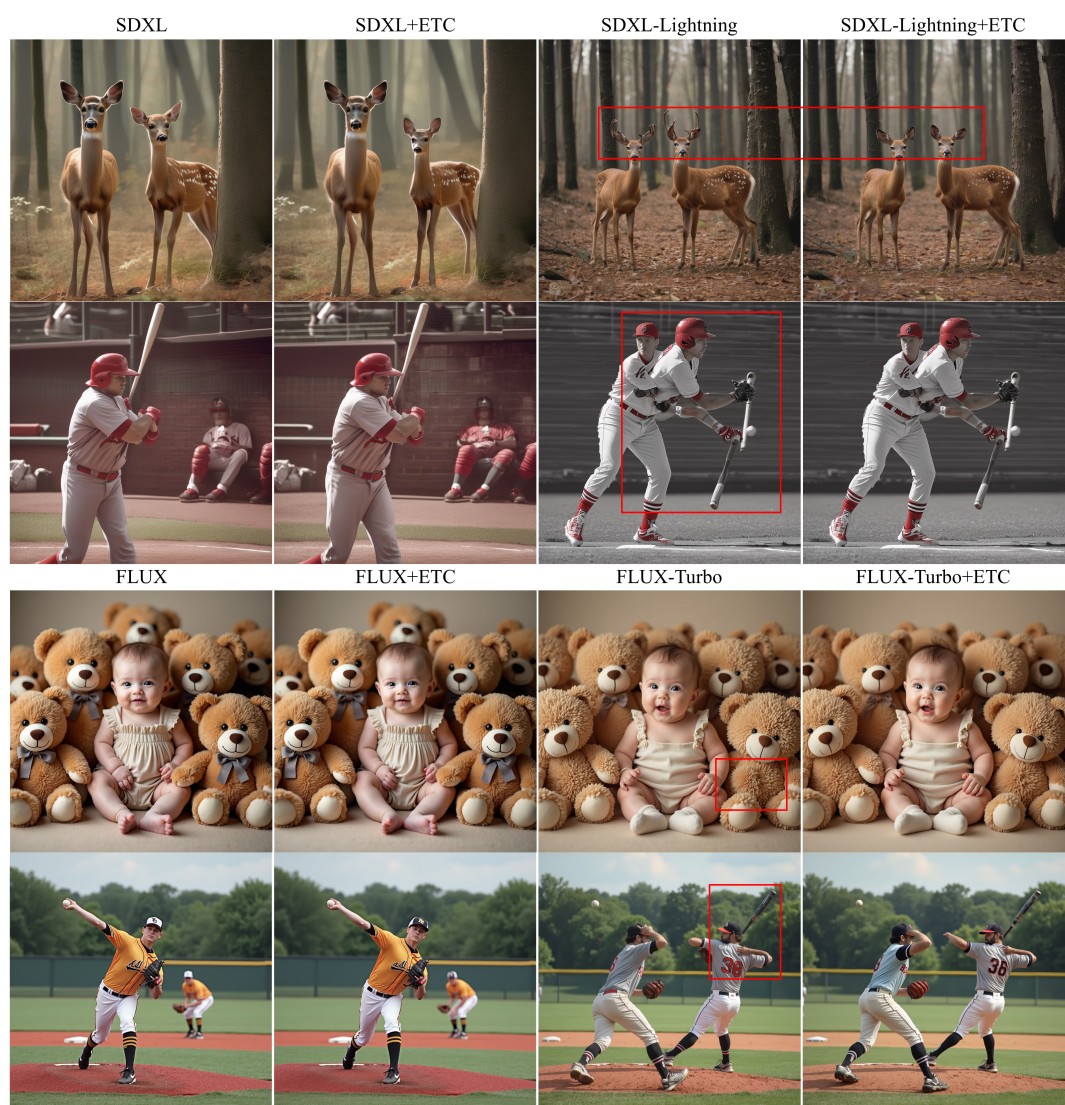

| SDXL | SDXL+ETC | SDXL-Lightning | SDXL-Lightning+ETC |

| FLUX | FLUX+ETC | FLUX-Turbo | FLUX-Turbo+ETC |

Figure 14: Comparison of visual quality with the trained few-step method.

adopted few-step models for OpenSora, Wan and TangoFlux are not yet publicly available, we omit those experiments for now. In addition, we introduce the MPS (Zhang et al., 2024) metric to evaluate the quality of generated images in terms of attributes such as color, style, and texture.

As shown in Table 6, although the trained few-step models achieve substantial improvements in generation speed, they exhibit a noticeable decrease in CLIP scores compared with their multi-step counterparts. While SDXL-Lightning emphasizes aesthetics during training and exceeds the multi-step model on the MPS metric, it also introduces structural distortions—for example, merged human limbs or mixed deer antlers and ears, as illustrated in Figure 14. FLUX-Turbo shows similar issues, such as abnormal hand generation. Moreover, training-based approaches require significant hardware resources. For instance, SDXL-Lightning is trained on 64

Table 6: Quantitative results compared with trained few-step methods.

| Method | MPS ↑ | CLIP ↑ | Latency (s) ↓ | Speedup ↑ |
|---|---|---|---|---|
| SDXL ($T = 50$) | 10.768 | **27.253** | 9.63 | 1× |
| ETC ($T = 50$) | 10.701 | 27.103 | 4.53 | 2.12× |
| SDXL-Lightning ($T = 8$) | **12.003** | 26.517 | 1.41 | 6.82× |
| SDXL-Lightning +ETC ($T = 8$) | 11.957 | 26.458 | **1.08** | **8.91×** |
| FLUX ($T = 50$) | **12.941** | **26.319** | 28.85 | 1× |
| ETC ($T = 50$) | 12.898 | 25.983 | 10.86 | 2.65× |
| FLUX-turbo ($T = 8$) | 12.884 | 25.968 | 4.87 | 5.92× |
| FLUX-turbo +ETC ($T = 8$) | 12.852 | 25.891 | **3.42** | **8.43×** |

Table 7: Quantitative results using different solvers.

| Method | Visual Quality | | | | Efficiency | | |
|---|---|---|---|---|---|---|---|
| **SDXL** | **LPIPS ↓** | **SSIM ↑** | **PSNR ↑** | **CLIP ↑** | **FLOPs (P) ↓** | **Speedup ↑** | **Latency (s) ↓** |
| DDIM | - | - | - | 27.278 | 0.67 | 1× | 9.27 |
| ETC w/DDIM | 0.103 | 0.881 | 28.106 | 27.251 | 0.27 | 2.15× | 4.31 |
| IPNDM | - | - | - | 26.121 | 0.67 | 1× | 9.31 |
| ETC w/IPNDM | 0.119 | 0.873 | 27.963 | 26.063 | 0.26 | 2.01× | 4.65 |
| DPM++ | - | - | - | 26.668 | 0.67 | 1× | 9.15 |
| ETC w/DPM++ | 0.096 | 0.882 | 28.678 | 26.662 | 0.24 | 2.15× | 4.26 |
| **FLUX** | **LPIPS ↓** | **SSIM ↑** | **PSNR ↑** | **CLIP ↑** | **FLOPs (P) ↓** | **Speedup ↑** | **Latency (s) ↓** |
| Heun | - | - | - | 25.522 | 7.36 | 1× | 50.21 |
| ETC w/Heun | 0.053 | 0.931 | 29.421 | 25.301 | 1.56 | 4.33× | 11.57 |
| UniPC | - | - | - | 25.629 | 3.71 | 1× | 28.54 |
| ETC w/UniPC | 0.089 | 0.912 | 28.411 | 25.451 | 1.12 | 2.81× | 10.13 |
| DPM++ | - | - | - | 25.734 | 3.71 | 1× | 28.47 |
| ETC w/DPM++ | 0.068 | 0.922 | 29.052 | 25.409 | 1.26 | 2.78× | 10.23 |
| **Open-Sora 1.2** | **LPIPS ↓** | **SSIM ↑** | **PSNR ↑** | **VBench ↑** | **FLOPs (P) ↓** | **Speedup ↑** | **Latency (s) ↓** |
| Heun | - | - | - | 77.79% | 5.96 | 1× | 82.76 |
| ETC w/Heun | 0.101 | 0.874 | 27.566 | 77.55% | 1.98 | 2.71× | 30.60 |
| UniPC | - | - | - | 77.43% | 3.08 | 1× | 43.81 |
| ETC w/UniPC | 0.125 | 0.859 | 25.565 | 77.06% | 1.49 | 2.08× | 21.04 |
| DPM++ | - | - | - | 77.39% | 3.08 | 1× | 43.71 |
| ETC w/DPM++ | 0.137 | 0.845 | 24.114 | 76.78% | 1.09 | 2.24× | 19.43 |
| **Wan 2.1** | **LPIPS ↓** | **SSIM ↑** | **PSNR ↑** | **VBench ↑** | **FLOPs (P) ↓** | **Speedup ↑** | **Latency (s) ↓** |
| Heun | - | - | - | 81.01% | 150.68 | 1× | 1906.29 |
| ETC w/Heun | 0.105 | 0.807 | 24.155 | 80.73% | 46.29 | 3.16× | 603.26 |
| Euler | - | - | - | 81.09% | 76.57 | 1× | 966.22 |
| ETC w/Euler | 0.079 | 0.843 | 25.293 | 80.88% | 29.03 | 2.43× | 396.81 |
| DPM++ | - | - | - | 81.03% | 75.39 | 1× | 958.85 |
| ETC w/DPM++ | 0.093 | 0.825 | 24.867 | 80.74% | 29.16 | 2.42× | 397.11 |
| **TangoFlux** | **FAD ↓** | **MCD ↓** | **KL ↓** | **CLAP ↑** | **FLOPs (T) ↓** | **Speedup ↑** | **Latency (s) ↓** |
| Heun | - | - | - | 13.285 | 92.85 | 1× | 10.98 |
| ETC w/Heun | 0.027 | 1.879 | 0.158 | 13.273 | 28.82 | 2.59× | 4.23 |
| UniPC | - | - | - | 13.209 | 46.86 | 1× | 5.56 |
| ETC w/UniPC | 0.029 | 1.885 | 0.149 | 13.189 | 14.83 | 2.45× | 2.27 |
| DPM++ | - | - | - | 13.276 | 46.86 | 1× | 5.50 |
| ETC w/DPM++ | 0.027 | 1.880 | 0.153 | 13.270 | 14.83 | 2.43× | 2.26 |

A100 80GB GPUs with a batch size of 512. In contrast, our method is friendly to consumer-level hardware while preserving both generation quality and inference speed. Additionally, our approach can be combined with few-step models to achieve an extra 20%–30% acceleration while maintaining consistent outputs, demonstrating its effectiveness and general applicability.

### A.8.3 ACCELERATION PERFORMANCE ACROSS DIFFERENT SOLVERS

**Solvers detail** All solvers are using the schedulers provided by the Diffusers library[2]. For FLUX, OpenSora, TangoFlux and Wan, where the model output is $v$, we specify $prediction\_type = flow\_prediction$ when using the scheduler.

**Quantitative and qualitative results** As shown in Table 7 and Figure 15, our method maintains consistent acceleration and generation quality across different solvers without re-tuning the threshold. This indicates that the threshold we search for captures the error tolerance that prevents trajectory deviation in the model's outputs. Since this property is intrinsic to the model rather than to the choice of solver, the same threshold remains applicable as long as the model itself is unchanged.

### A.8.4 ACCELERATION PERFORMANCE ACROSS DIFFERENT RESOLUTIONS

To further evaluate the robustness of our acceleration method, we evaluate ETC under different resolutions without changing the threshold. For image generation models, we use resolutions of 768×1344 (9:16) and 1152×896 (5:4). For video generation models, OpenSora is tested with 360p

---

[2]https://github.com/huggingface/diffusers/

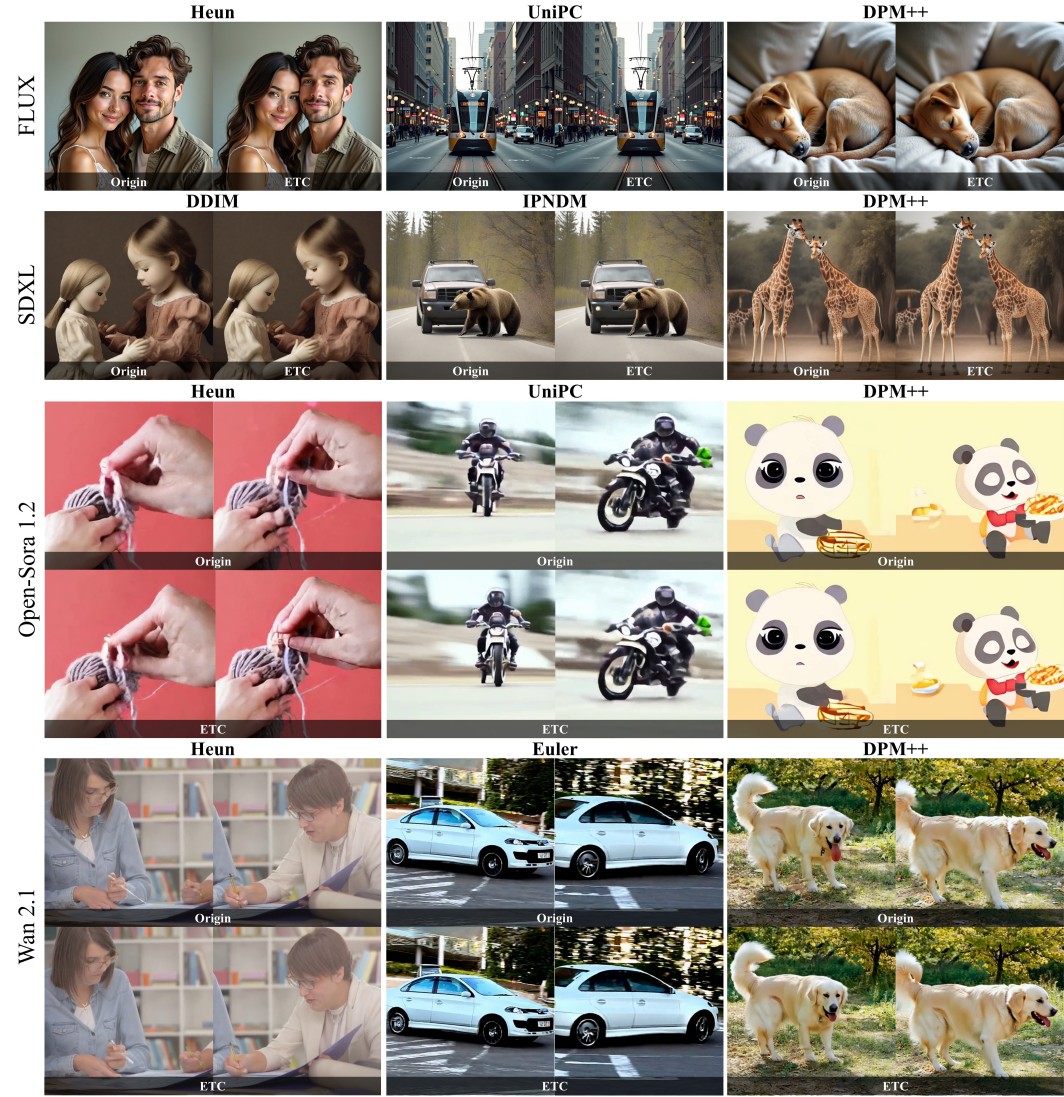

Figure 15: Comparison of visual quality using different solvers.

Table 8: Quantitative results across different resolutions.

| Method | Visual Quality | | | | Efficiency | | |
|---|---|---|---|---|---|---|---|
| **Text to Image** | **LPIPS ↓** | **SSIM ↑** | **PSNR ↑** | **CLIP ↑** | **FLOPs (P) ↓** | **Speedup ↑** | **Latency (s) ↓** |
| SDXL (768×1344) | - | - | - | 27.228 | 0.67 | 1× | 9.28 |
| ETC | 0.114 | 0.873 | 27.937 | 27.045 | 0.24 | 2.11× | 4.38 |
| SDXL (1152×896) | - | - | - | 27.221 | 0.67 | 1× | 9.31 |
| ETC | 0.103 | 0.879 | 28.113 | 27.048 | 0.23 | 2.12× | 4.37 |
| FLUX (768×1344) | - | - | - | 26.429 | 3.67 | 1× | 28.65 |
| ETC | 0.071 | 0.914 | 29.093 | 26.151 | 1.26 | 2.69× | 10.68 |
| FLUX (1152×896) | - | - | - | 26.139 | 3.67 | 1× | 28.59 |
| ETC | 0.069 | 0.919 | 29.189 | 26.058 | 1.25 | 2.68× | 10.62 |
| **Text to Video** | **LPIPS ↓** | **SSIM ↑** | **PSNR ↑** | **VBench ↑** | **FLOPs (P) ↓** | **Speedup ↑** | **Latency (s) ↓** |
| Open-Sora (360p,4:3) | - | - | - | 76.89% | 1.76 | 1× | 25.86 |
| ETC | 0.125 | 0.868 | 25.657 | 76.63% | 0.76 | 2.14× | 12.16 |
| Open-Sora (720p,1:1) | - | - | - | 77.22% | 6.75 | 1× | 105.88 |
| ETC | 0.151 | 0.853 | 24.624 | 76.98% | 2.02 | 2.23× | 46.52 |
| Wan 2.1 (720×1280) | - | - | - | 82.79% | 182.95 | 1× | 3456.53 |
| ETC | 0.089 | 0.853 | 26.763 | 82.56% | 68.83 | 2.78× | 1218.98 |

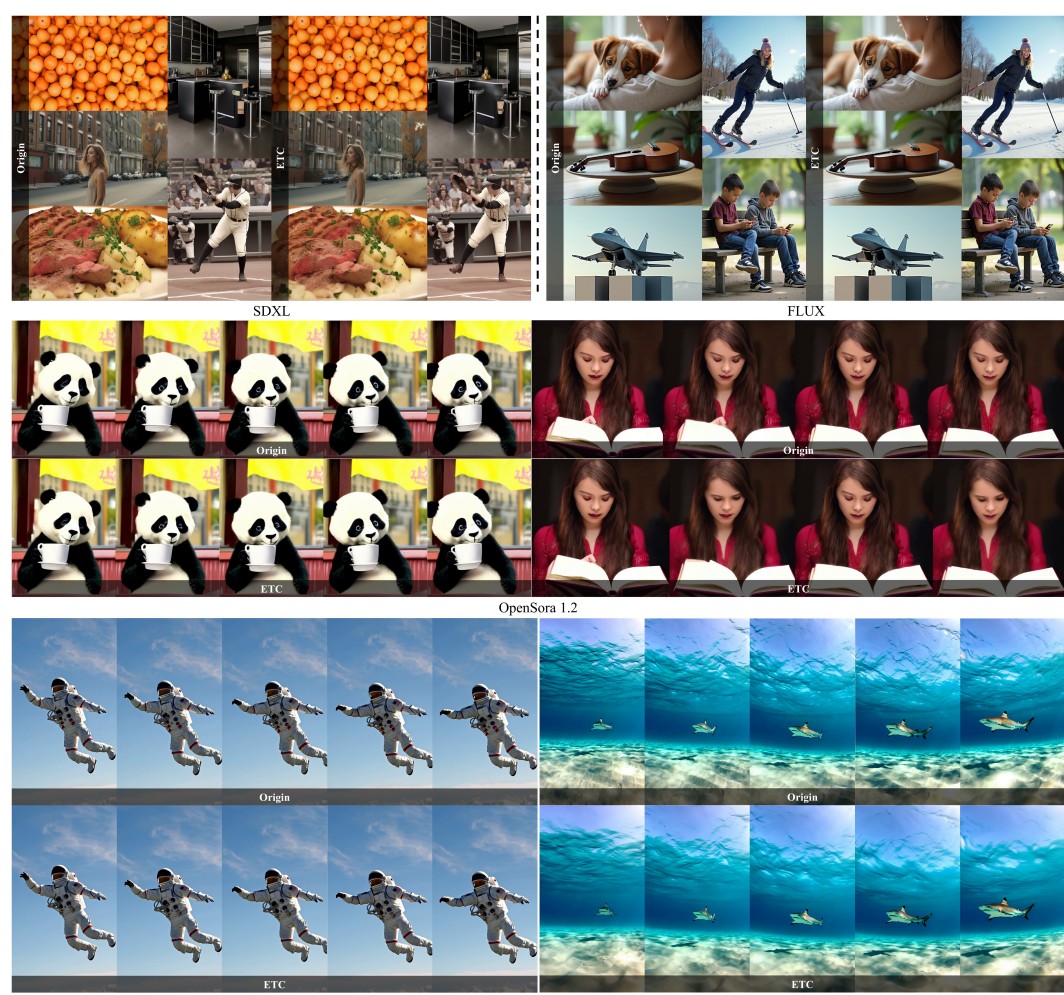

Figure 16: Comparison of visual quality at different resolutions.

Table 9: Quantitative results of dense text generation.

| Method | Visual Quality | | | | Efficiency | | |
|---|---|---|---|---|---|---|---|
| Text to Image | LPIPS ↓ | SSIM ↑ | PSNR ↑ | CLIP ↑ | FLOPs (P) ↓ | Speedup ↑ | Latency (s) ↓ |
| SDXL | - | - | - | 30.584 | 0.67 | 1× | 9.82 |
| ETC | 0.109 | 0.865 | 27.958 | 30.893 | 0.28 | 1.98× | 5.08 |
| FLUX | - | - | - | 32.547 | 3.64 | 1× | 29.19 |
| ETC | 0.083 | 0.916 | 28.344 | 32.286 | 1.31 | 2.51× | 11.49 |

(4:3) and 720p (1:1) settings, while Wan is tested at a resolution of 720×1280. As shown in Table 8 and Figure 16, our method maintains consistent acceleration and stable generation quality across different resolution settings.

### A.8.5 ACCELERATION PERFORMANCE IN OUT-OF-DISTRIBUTION PROMPTS

Our threshold search is conducted on a small set of prompts, raising the question of whether the method can maintain consistent generation performance under out-of-distribution prompts—especially in scenarios involving high-density text generation—and whether a smaller threshold is required to ensure final quality. Therefore, we evaluate our method on the TextAtlas5M (Wang et al., 2025) evaluation set. As shown in Table 9 and Figure 17, while our method maintains consistent generation quality, the generation time increases. This is because complex sce-

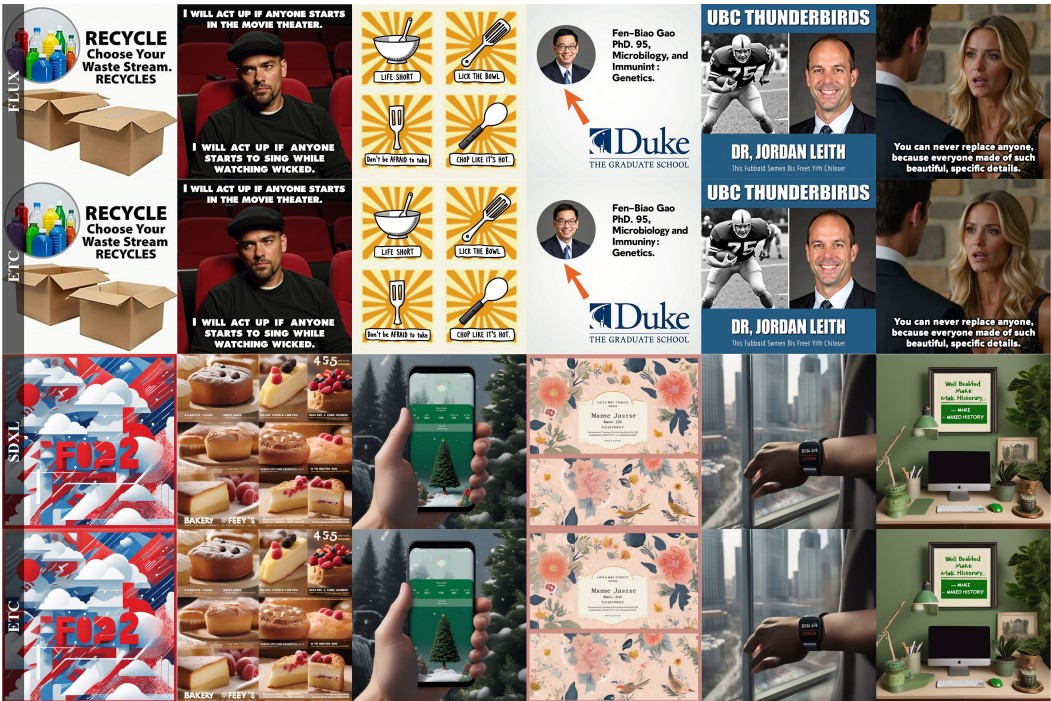

Figure 17: Comparison of visual quality at different resolutions.

Table 10: Quantitative results with more aggressive acceleration.

| Method | Visual Quality | | | | Efficiency | | |
|---|---|---|---|---|---|---|---|
| **Text to Image** | **LPIPS** ↓ | **SSIM** ↑ | **MPS** ↑ | **CLIP** ↑ | **FLOPs (P)** ↓ | **Speedup** ↑ | **Latency (s)** ↓ |
| SDXL | - | - | 10.768 | 27.253 | 0.67 | 1× | 9.63 |
| ETC | 0.105 | 0.876 | 10.701 | 27.103 | 0.26 | 2.12× | 4.53 |
| ETC-fast | 0.336 | 0.694 | 10.679 | 26.898 | 0.12 | 4.91× | 1.88 |
| FLUX | - | - | 12.941 | 26.319 | 3.64 | 1× | 28.85 |
| ETC | 0.068 | 0.926 | 12.898 | 25.983 | 1.31 | 2.65× | 10.86 |
| ETC-fast | 0.303 | 0.766 | 12.871 | 25.676 | 0.75 | 4.63× | 6.25 |
| **Text to Video** | **LPIPS** ↓ | **SSIM** ↑ | **PSNR** ↑ | **VBench** ↑ | **FLOPs (P)** ↓ | **Speedup** ↑ | **Latency (s)** ↓ |
| Open-Sora | - | - | - | 77.18% | 3.15 | 1× | 45.29 |
| ETC | 0.131 | 0.848 | 24.123 | 76.44% | 1.32 | 2.15× | 21.07 |
| ETC-fast | 0.591 | 0.492 | 10.967 | 76.29% | 0.797 | 4.06× | 10.67 |
| Wan 2.1 ($T = 50$) | - | - | - | 81.01% | 76.71 | 1× | 970.49 |
| ETC | 0.103 | 0.806 | 24.078 | 80.71% | 28.63 | 2.49× | 389.89 |
| ETC-fast | 0.299 | 0.603 | 18.193 | 80.48% | 17.58 | 4.49× | 215.67 |

narios are more prone to trajectory deviation, resulting in more frequent application of the threshold constraint. This indicates that the threshold derived from a small data set maintains robustness.

## A.9 MORE AGGRESSIVE ACCELERATION

To assess the stability of generation quality under higher acceleration, we apply a larger number of approximation times in the first round of consistent trend predict ($k = 4$). As shown in Table 10, our method achieves a 4–5× speedup. Although consistency drops significantly under this higher acceleration setting, the overall generation quality remains stable—both the CLIP score and the VBench score stay at high levels. Figure 18 presents a good-case example under high acceleration, demonstrating that our method does not suffer from noticeable quality collapse even in this regime, indicating robustness and acceleration potential.

**Problems arising from trajectory deviation**  Although our method maintains a certain level of generation quality under high acceleration, the lack of strict constraints causes deviations in the denoising trajectory, which in turn leads to several issues. In addition to the decreases observed

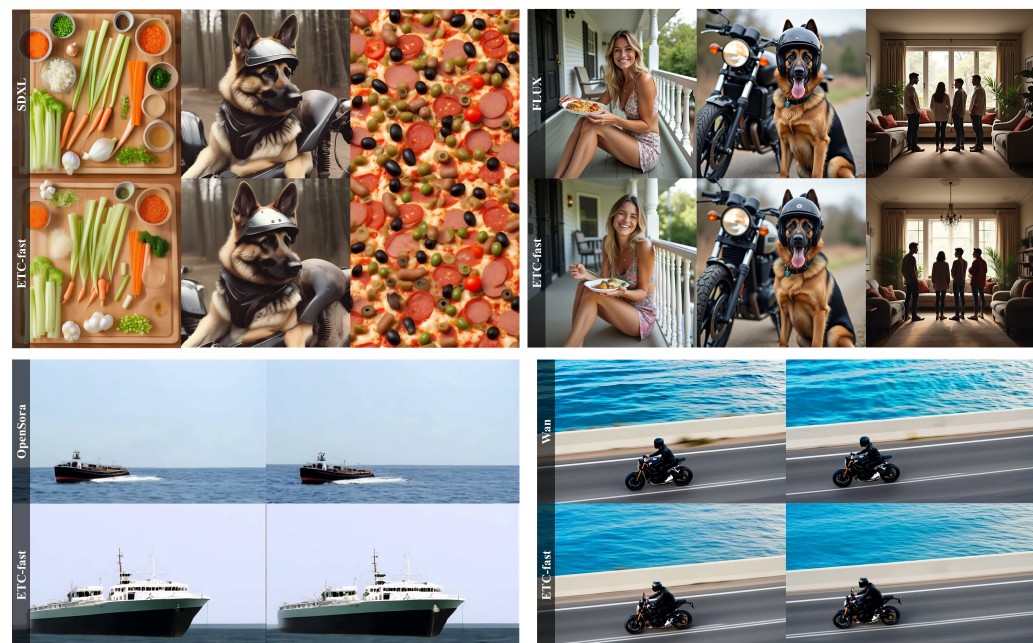

Figure 18: Visual quality under aggressive acceleration.

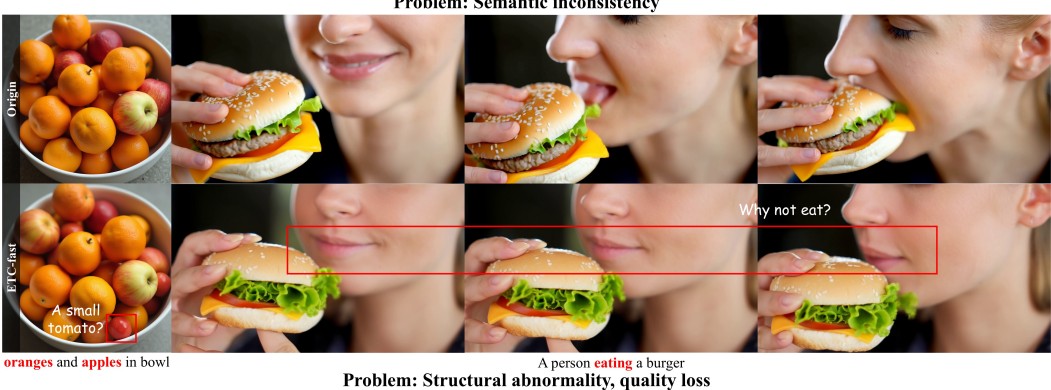

Figure 19: Problems arising from trajectory deviation.

in CLAP, MPS, and VBench metrics in Table 10, we also identify two common problems illustrated in Figure 19. For semantic inconsistency, the generated content does not fully align with the prompt—for example, generating a person with a hamburger but without the intended eating action. For structural abnormalities, highly fine-grained and structured regions such as hands or text often exhibit distortion or deformation. Therefore, our designs for trend consistency and error control aim to minimize deviations from the target generation trajectory and preserve quality under acceleration.

Table 11: Comparison of computational cost between our method and the full-history trend storage approach. Best performance in **bold**.

| Method | Visual Quality | | | | Efficiency | | |
|---|---|---|---|---|---|---|---|
| **SDXL** | **LPIPS** ↓ | **SSIM** ↑ | **PSNR** ↑ | **CLIP** ↑ | **FLOPs (M)** ↓ | **VRAM (MB)** ↓ | **Latency (s)** ↓ |
| ETC | 0.105 | **0.876** | 28.017 | 27.103 | **9.61** | **0.75** | **4.53** |
| ETC w/all-trend | **0.103** | 0.875 | **28.025** | **27.105** | 35.14 | 3.44 | 4.82 |
| **FLUX** | **LPIPS** ↓ | **SSIM** ↑ | **PSNR** ↑ | **CLIP** ↑ | **FLOPs (M)** ↓ | **VRAM (MB)** ↓ | **Latency (s)** ↓ |
| ETC | **0.068** | **0.926** | 29.176 | **25.983** | 25.83 | **1.01** | 10.86 |
| ETC w/all-trend | 0.070 | 0.925 | **29.177** | 25.976 | 67.72 | 6.73 | 11.35 |
| **OpenSora** | **LPIPS** ↓ | **SSIM** ↑ | **PSNR** ↑ | **VBench** ↑ | **FLOPs (M)** ↓ | **VRAM** ↓ | **Latency (s)** ↓ |
| ETC | 0.131 | **0.848** | **24.123** | **76.44%** | **23.65** | **18.34** | **21.07** |
| ETC w/all-trend | **0.128** | 0.848 | 24.119 | 76.44% | 74.03 | 39.22 | 23.51 |
| **Wan 2.1** | **LPIPS** ↓ | **SSIM** ↑ | **PSNR** ↑ | **VBench** ↑ | **FLOPs (M)** ↓ | **VRAM** ↓ | **Latency (s)** ↓ |
| ETC | **0.103** | **0.806** | 24.078 | **80.71%** | **211.76** | **40.99** | **389.89** |
| ETC w/all-trend | 0.105 | 0.805 | **24.101** | 80.69% | 624.79 | 104.21 | 397.34 |
| **TangoFlux** | **FAD** ↓ | **MCD** ↓ | **KL** ↓ | **CLAP** ↑ | **FLOPs (M)** ↓ | **VRAM** ↓ | **Latency (s)** ↓ |
| ETC | **0.026** | **1.877** | 0.157 | 13.251 | **4.74** | **0.94** | **2.26** |
| ETC w/all-trend | 0.028 | 1.877 | **0.155** | **13.252** | 11.51 | 3.31 | 2.53 |

### A.9.1 COMPUTATIONAL COST OF OUR METHOD

To evaluate the computational overhead of our method, we measure both VRAM consumption and FLOPs. In addition, to assess the effectiveness of recursively computing trends, we compare our approach with an alternative method that estimates future trends by storing all historical trends. For this baseline, we apply an exponentially increasing weighting scheme in which older trends receive smaller weights and more recent trends receive larger weights, with all weights sum to 1.

As shown in Table 11, our method requires only a few of megabytes of additional VRAM during diffusion inference, and its computational cost is negligible compared with the overall diffusion model inference. This demonstrates that our approach is computation-efficient. Moreover, compared with the method that stores all historical trends to estimate future trends, our recursive formulation achieves comparable acceleration performance and generation quality while reducing both computational cost and memory consumption by more than 50%, thereby significantly mitigating the overhead introduced by trend computation.

### A.9.2 A FEASIBLE STRATEGY FOR DYNAMICALLY ADAPTING $\alpha$ AND $n$

**Adaptive $n$** The purpose of setting $n$ is to ensure that our acceleration method is applied only after the diffusion model reaches a stable denoising phase. The threshold $\sigma$ is obtained by analyzing the transition point between the semantic planning stage and the stable quality-improvement stage of the diffusion process. Therefore, by checking whether the difference between model outputs at adjacent timesteps falls below this threshold, we can determine when to start acceleration and achieve a dynamic $n$.

**Adaptive $\alpha$** For $\alpha$, when the difference between the model output and the previous approximation is equal to the estimated trend (i.e., their difference is zero), it indicates that the estimated trend perfectly matches the original denoising trajectory. When the difference becomes larger, it suggests that the current $\alpha$ is insufficient and should be updated in the opposite direction. Conversely, when the difference becomes smaller, it indicates that $\alpha$ is being updated in the correct direction and should continue to move along that trajectory. Based on this, we can design an adaptive $\alpha$ update strategy driven by the gradient of the change. Additionally, $\alpha$ can be set as a matrix of the same size as latent, enabling distinct trend updates for each pixel. The pseudocode for adaptive $n$ and $\alpha$ is shown in Algorithm 3.

Table 12 presents the results of applying our method with adaptive $n$ and $\alpha$. We observe a slight improvement in generation quality after introducing the dynamic strategy. This suggests that although our method is generally insensitive to the choice of $\alpha$ (as shown by Table 4, where $\alpha = 0.5$ already yields well acceleration performance), adapting $\alpha$ to different levels of accumulated error can further enhance generation quality. Our simple adaptive mechanism already provides gains, indicating that future work could build on this direction by more deeply analyzing how model-output

Table 12: Quantitative results by using adaptive $n$ and $\alpha$. Best performance in **bold**.

| Method | Visual Quality | | | | Efficiency | | |
|---|---|---|---|---|---|---|---|
| **Text to Image** | **LPIPS** $\downarrow$ | **SSIM** $\uparrow$ | **PSNR** $\uparrow$ | **CLIP** $\uparrow$ | **FLOPs (P)** $\downarrow$ | **Speedup** $\uparrow$ | **Latency (s)** $\downarrow$ |
| SDXL ($T = 50$) | - | - | - | 27.253 | 0.67 | $1\times$ | 9.63 |
| ETC | 0.105 | 0.876 | **28.017** | 27.103 | 0.26 | $2.12\times$ | 4.53 |
| ETC-adaptive | **0.101** | **0.883** | 28.009 | **27.129** | **0.26** | **2.16**$\times$ | **4.44** |
| FLUX | - | - | - | 26.319 | 3.64 | $1\times$ | 28.85 |
| ETC | 0.068 | 0.926 | 29.176 | 25.983 | 1.31 | $2.65\times$ | 10.86 |
| ETC-adaptive | **0.061** | **0.928** | **29.211** | **25.989** | **1.29** | **2.74**$\times$ | **10.51** |
| **Text to Video** | **LPIPS** $\downarrow$ | **SSIM** $\uparrow$ | **PSNR** $\uparrow$ | **VBench** $\uparrow$ | **FLOPs (P)** $\downarrow$ | **Speedup** $\uparrow$ | **Latency (s)** $\downarrow$ |
| Open-Sora 1.2 ($T = 30$) | - | - | - | 77.18% | 3.15 | $1\times$ | 45.29 |
| ETC | **0.131** | 0.848 | 24.123 | 76.44% | 1.32 | $2.15\times$ | 21.07 |
| ETC-adaptive | 0.132 | **0.852** | **24.235** | **76.58%** | **1.32** | **2.15**$\times$ | **21.02** |
| Wan 2.1 ($T = 50$) | - | - | - | 81.01% | 76.71 | $1\times$ | 970.49 |
| ETC | **0.103** | 0.806 | 24.078 | 80.71% | 28.63 | $2.49\times$ | 389.89 |
| ETC-adaptive | 0.104 | **0.812** | **24.101** | **80.92%** | **28.54** | **2.50**$\times$ | **388.34** |
| **Text to Audio** | **FAD** $\downarrow$ | **MCD** $\downarrow$ | **KL** $\downarrow$ | **CLAP** $\uparrow$ | **FLOPs (T)** $\downarrow$ | **Speedup** $\uparrow$ | **Latency (s)** $\downarrow$ |
| TangoFlux ($T = 50$) | - | - | - | 13.286 | 46.86 | $1\times$ | 5.49 |
| ETC | 0.026 | 1.877 | 0.157 | 13.251 | 14.83 | $2.43\times$ | 2.26 |
| AdaptiveDiffusion | **0.022** | **1.796** | **0.156** | **13.263** | **14.76** | **2.43**$\times$ | **2.25** |

---

**Algorithm 3** Adaptive $n$ and $\alpha$.

---

**Input:** Diffusion Model $\epsilon_\theta$, Sample Step $T$, Conditional Embedding $c$, Error Threshold $\sigma$;

1: Future Trend $\Delta = $ None, Previous Model Output $P = 0$, Smoothing matrix $\alpha$, Previous smoothing matrix $\alpha_p$, Previous trend estimation error $E_t$, Start to acceleration $s = $ False.
2: **for** $t = T$ to $T - n$ **do**
3:    **if** not $s$ **then**
4:       **if** $\epsilon_\theta(x, t, c) - P < \sigma$ **then**
5:          $s = $True;
6:       **end if**
7:       Update $P$ and $\Delta$;
8:    **else**
9:       Acceleration by using consistent trend predictor;
10:       $grad = (abs(\epsilon_\theta(x, t, c) - P - \delta) - E_t)/(\alpha - \alpha_p + 1e^{-8})$;
11:       $\alpha_p = \alpha$;
12:       $E_t = abs(\epsilon_\theta(x, t, c) - P - \delta)$;
13:       $\alpha = \alpha - grad$;
14:       $\alpha$.clamp(0.1,0.9);
15:    **end if**
16: **end for**
17: **return** $D(x)$.

---

deviations under different error accumulations influence future denoising trajectories. Such analysis could guide more principled adjustments of the trend-update magnitude.

### A.10 ONLINE THRESHOLD SEARCH METHOD FOR UNSEEN MODELS

Our threshold search is performed offline on a small amount of dataset. However, when encountering new data (e.g. different resolutions or prompts with varying complexity), the offline threshold may still have room for improvement. To investigate this, we design an online threshold search method and compare it with the offline approach. Specifically, we randomly sample and combine seven conditions across different samplers, resolutions, and prompts. For each condition, we perform threshold searching and store the resulting value. Whenever a new threshold is obtained, we update the global threshold by taking the average of all collected thresholds, and then evaluate the acceleration performance under this updated threshold.

As shown in Table 13, small adjustments to the threshold do not lead to significant changes in acceleration performance. As more online search results are accumulated, the threshold gradually converges to a universal value, which also lies near the one obtained by our offline method. This

Table 13: Quantitative results by using online search $\sigma$.

| Model | $\sigma$ update round | 1 | 2 | 3 | 4 | 5 | 6 | 7 |
|---|---|---|---|---|---|---|---|---|
| SDXL | $\sigma$-search | 0.0096 | 0.0149 | 0.0108 | 0.0187 | 0.0193 | 0.0203 | 0.0151 |
| | $\sigma$-mean | 0.0096 | 0.0122 | 0.0115 | 0.0151 | 0.0172 | 0.0187 | 0.0169 |
| | SSIM↑ | 0.883 | 0.879 | 0.880 | 0.878 | 0.876 | 0.874 | 0.876 |
| | CLIP↑ | 27.159 | 27.122 | 27.129 | 27.110 | 27.105 | 27.103 | 27.110 |
| | Latency (s) ↓ | 5.05 | 4.79 | 4.82 | 4.65 | 4.55 | 4.31 | 4.56 |
| FLUX | $\sigma$-search | 0.0986 | 0.1284 | 0.1032 | 0.1411 | 0.1314 | 0.1102 | 0.1417 |
| | $\sigma$-mean | 0.0986 | 0.1135 | 0.1083 | 0.1247 | 0.1281 | 0.1191 | 0.1304 |
| | SSIM↑ | 0.928 | 0.927 | 0.927 | 0.926 | 0.926 | 0.926 | 0.925 |
| | CLIP↑ | 26.279 | 26.087 | 26.281 | 25.983 | 25.982 | 25.983 | 25.981 |
| | Latency (s) ↓ | 11.89 | 11.02 | 11.78 | 10.82 | 10.85 | 10.83 | 10.56 |
| OpenSora | $\sigma$-search | 0.1841 | 0.2327 | 0.1747 | 0.2505 | 0.1960 | 0.2171 | 0.2006 |
| | $\sigma$-mean | 0.1841 | 0.2084 | 0.1915 | 0.2210 | 0.2085 | 0.2128 | 0.2067 |
| | SSIM↑ | 0.852 | 0.848 | 0.850 | 0.845 | 0.848 | 0.848 | 0.848 |
| | VBench↑ | 76.63% | 76.44% | 76.52% | 76.40% | 76.44% | 76.44% | 76.44% |
| | Latency (s) ↓ | 22.23 | 21.09 | 21.96 | 20.98 | 21.10 | 21.09 | 21.05 |
| Wan | $\sigma$-search | 0.0879 | 0.1139 | 0.1016 | 0.1361 | 0.1289 | 0.0977 | 0.1248 |
| | $\sigma$-mean | 0.0879 | 0.1009 | 0.1012 | 0.1186 | 0.1237 | 0.1107 | 0.1177 |
| | SSIM↑ | 0.821 | 0.810 | 0.810 | 0.806 | 0.793 | 0.806 | 0.806 |
| | VBench↑ | 80.72% | 80.71% | 80.71% | 80.71% | 80.61% | 80.71% | 80.71% |
| | Latency (s) ↓ | 495.21 | 403.55 | 403.67 | 392.22 | 382.19 | 391.02 | 390.12 |
| TangoFlux | $\sigma$-search | 0.0595 | 0.0756 | 0.0621 | 0.0641 | 0.0701 | 0.0621 | 0.0658 |
| | $\sigma$-mean | 0.0595 | 0.0675 | 0.0648 | 0.0644 | 0.0672 | 0.0646 | 0.0652 |
| | KL↓ | 0.148 | 0.157 | 0.155 | 0.155 | 0.157 | 0.155 | 0.156 |
| | CLAP↑ | 13.260 | 13.251 | 13.252 | 13.252 | 13.251 | 13.252 | 13.251 |
| | Latency (s) ↓ | 2.41 | 2.23 | 2.32 | 2.34 | 2.25 | 2.33 | 2.32 |

indicates that providing an approximate range for the threshold is sufficient. Consequently, the threshold estimated from a small number of prompts exhibits robustness comparable to that of the online approach.

## A.11 WORKFLOW OF THE TREND INFLECTION POINT ANALYSIS MODEL

For the analysis of trend inflection point, we use the Bottom-up (Truong et al., 2020) method from the ruptures library[3]. Bottom-up segmentation, denoted BotUp, starts by splitting the original signal in many small sub-signals and sequentially merges them until there remain only $K$ change points. At every step, all potential change points (indexes separating adjacent sub-segments) are ranked by the discrepancy measure $d$ between the segments they separate. Change points with the lowest discrepancy are then deleted, meaning that the segments they separate are merged. A schematic view of the algorithm is displayed on Figure 20 and an implementation is provided in Algorithm 4. Its benefits are its linear computational complexity and conceptual simplicity. For the discrepancy measure function $d$, we select Least Absolute Deviation (CostL1). Formally, for a signal $\{y_t\}_t$ on an interval $I$:

$$c(y_I) = \sum_{t \in I} ||y_t - \overline{y}||_1, \tag{54}$$

where $\overline{y}$ is the componentwise median of $\{y_t\}_{t \in I}$.

---

[3]https://github.com/deepcharles/ruptures

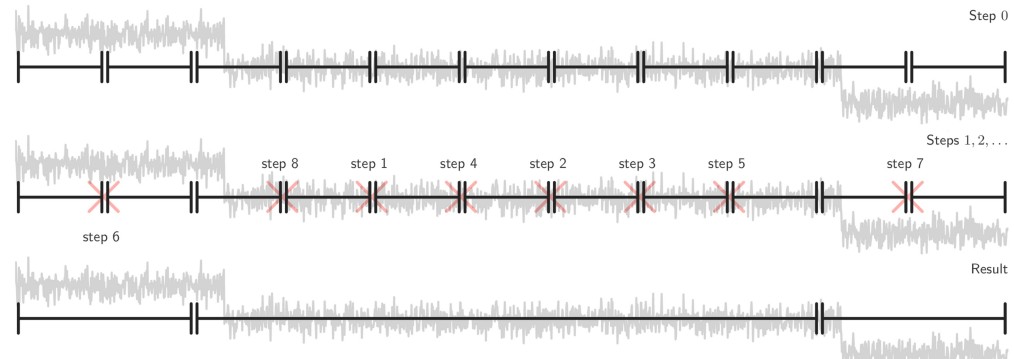

Figure 20: Schematic view of Bottom-up (Truong et al., 2020).

---

**Algorithm 4** Algorithm Bottom-up.

---

**Input:** signal $\{y_t\}_{t=1}^T$, cost function $c(\cdot)$, stopping criterion, grid size $\delta > 2$.
1: Initialize $L \leftarrow \delta, 2\delta, \ldots, (\lfloor T/\delta \rfloor - 1)\delta$.
2: **repeat**
3:    $k \leftarrow |L|$.
4:    $t_0 \leftarrow 0$ and $t_{k+1} \leftarrow T$
5:    Denote by $t_i (i = 1, \ldots, k)$ the elements (in ascending order) of $L$, ie $L = \{t_1, \ldots, t_k\}$.
6:    Initialize $G$, a $(k-1)$-long array.
7:    **for** $i = 1, \ldots, k-1$ **do**
8:       $G[i-1] \leftarrow c(y_{t_{i-1}..t_{i+1}}) - [c(y_{t_{i-1}..t_i}) + c(y_{t_i..t_{i+1}})]$.
9:    **end for**
10:    $\hat{i} \leftarrow argmin_i G[i]$
11:    Remove $t_{\hat{i}+1}$ from $L$.
12: **until** stopping criterion is met.
13: **return** set $L$ of estimated breakpoint indexes.

---

## B  LLM USAGE

Large Language Models (LLMs) were used to aid in the writing and polishing of the manuscript. Specifically, we used an LLM to assist in refining the language, improving readability, and ensuring clarity in various sections of the paper. The model helped with tasks such as sentence rephrasing, grammar checking, and enhancing the overall flow of the text.

It is important to note that the LLM was not involved in the ideation, research methodology, or experimental design. All research concepts, ideas, and analyses were developed and conducted by the authors. The contributions of the LLM were solely focused on improving the linguistic quality of the paper, with no involvement in the scientific content or data analysis.

The authors take full responsibility for the content of the manuscript, including any text generated or polished by the LLM. We have ensured that the LLM-generated text adheres to ethical guidelines and does not contribute to plagiarism or scientific misconduct.

