# OpenReview forum: "ETC: training-free diffusion models acceleration with Error-aware Trend Consistency"
_ICLR.cc/2026/Conference — Submitted to ICLR 2026_

### Official Review · Reviewer_cWaK · 2025-10-29

**Soundness:** 2
**Presentation:** 2
**Contribution:** 2
**Rating:** 4
**Confidence:** 4

**Summary:**

This paper proposes a diffusion model acceleration framework called Error-aware Trend Consistency (ETC), which accelerates diffusion sampling without training while maintaining generation fidelity. ETC consists of a consistent trend predictor and a model-specific error tolerance search mechanism: the consistent trend predictor estimates a stable future denoising direction through historical model outputs, and the model-specific error tolerance search mechanism determines a safely reusable threshold based on denoising dynamics. Experiments show that ETC achieves a maximum speedup of 2.65× in image, video, and audio tasks, with negligible quality degradation.

**Strengths:**

1. The experiments are relatively comprehensive, covering mainstream quantitative comparison results and qualitative visualization results. Additionally, ablation experiments are conducted to verify the effectiveness of error control, statistical robustness, and the role of key parameters $n$ and $\alpha$.
2. The qualitative visualization results show that ETC maintains better generation consistency and avoids the structural distortion and detail loss issues present in the baseline methods.
3.  This paper features detailed derivations, which reduces the difficulty of understanding the proposed method. The provided illustrations and algorithm pseudocode further facilitate the comprehension of this paper.

**Weaknesses:**

1. It has not been compared with other training-free acceleration methods, such as methods based on intra-layer feature reuse and methods based on solvers.
2. The description of the search for model-specific error tolerance is relatively vague, which is only illustrated through Algorithm 2. Specific workflows are not provided for models like the Trend Inflection Point Analysis Model.
3. The important hyperparameters $\alpha$ and $n$ depend on being specified manually.
4. Some parameter names are incorrect. For example, in the title of Table 4, $p$ should be $n$.

**Questions:**

1. What is the difference in computational overhead between storing all historical trends and the recursive historical trend weighting method proposed in this paper?
2. Is the ETC proposed in this paper compatible with DDIM[1]? And is it compatible with other high-order samplers such as iPNDM[2] and DPM-Solver++[3]?
3. Does an adaptive optimization strategy for $\alpha$ exist?

[1] Song J, Meng C, Ermon S. Denoising diffusion implicit models[J]. arXiv preprint arXiv:2010.02502, 2020.

[2] Luping Liu, Yi Ren, Zhijie Lin, and Zhou Zhao. Pseudo numerical methods for diffusion models on
 manifolds. arXiv preprint arXiv:2202.09778, 2022.

[3] Lu C, Zhou Y, Bao F, et al. Dpm-solver++: Fast solver for guided sampling of diffusion probabilistic models[J]. Machine Intelligence Research, 2025: 1-22.

---

> ### Author Response · Authors · 2025-11-30
>
> >Q1: Compared with intra-layer feature reuse methods and methods based on solvers.
>
> A1: Thank you for raising this important point. Our method is designed to accelerate inference by approximating the step-wise model outputs. Therefore, the main experiments in the paper primarily compare against other step-wise acceleration methods. Nevertheless, we agree that comparing with intra-layer feature reuse approaches can provide a more comprehensive evaluation of our method’s effectiveness.
>
> To address this, we have conducted additional quantitative experiments with intra-layer feature reuse methods: DeepCache[1] for SDXL, ToCa[2] for FLUX, and Faster-Cache[3] for Open-Sora. For Wan and TangoFlux, the corresponding inference implementations are not publicly available, so we are unable to include these baselines. The results are presented below.
>
> ||LPIPS↓|SSIM↑|PSNR↑|CLIP↑|FLOPs(p)↓|Speedup↑|Latency(s)↓|
> |:------: | :------: |:------: |:------: | :------: |:------: |:------: | :------: |
> |SDXL|-|-|-|27.253|0.67|1×|9.63|
> |ETC|**0.105**|**0.876**|**28.017**|**27.103**|0.26|2.12×|4.53|
> |DeepCache|0.205|0.791|23.871|27.055|0.25|2.17×|4.33|
> |DeepCache+ETC|0.231|0.782|22.547|27.049|**0.14**|**4.29×**|**2.24**|
>
> ||LPIPS↓|SSIM↑|PSNR↑|CLIP↑|FLOPs(p)↓|Speedup↑|Latency(s)↓|
> |:------: | :------: |:------: |:------: | :------: |:------: |:------: | :------: |
> |FLUX|-|-|-|26.319|3.64|1×|28.85|
> |ETC|**0.068**|**0.926**|**29.176**|**25.983**|1.31|2.65×|10.86|
> |ToCa|0.322|0.694|18.387|25.286|1.86|1.88×|15.31|
> |ToCa+ETC|0.351|0.687|18.187|25.711|**1.01**|**3.43×**|**8.41**|
>
> ||LPIPS↓|SSIM↑|PSNR↑|VBench↑|FLOPs(p)↓|Speedup↑|Latency(s)↓|
> |:------: | :------: |:------: |:------: | :------: |:------: |:------: | :------: |
> |Open-Sora|-|-|-|77.18%|3.15|1×|45.29|
> |ETC|**0.131**|**0.848**|**24.123**|**76.44%**|1.32|2.15×|21.07|
> |ToCa|0.166|0.793|23.282|76.37%|2.08|1.58×|28.67|
> |ToCa+ETC|0.168|0.802|23.389|76.38%|**1.03**|**2.62×**|**17.28**|
>
> The additional experiments show that our method achieves higher speedups while maintaining better generation consistency and quality compared with these methods. Moreover, our approach can be combined with intra-layer feature reuse to obtain even greater acceleration with only a minor quality drop. This demonstrates the generality and compatibility of our framework. We also include more visual results in Appendix A.8.1.
>
> What's more, since our method focuses on approximating the model output at each step, it can be applied to different solvers. We will analyze the performance of our method under various solvers in Q2.
>
> [1] Ma X, Fang G, Wang X. Deepcache: Accelerating diffusion models for free[C]//Proceedings of the IEEE/CVF conference on computer vision and pattern recognition. 2024: 15762-15772.
>
> [2] Zou C, Liu X, Liu T, et al. Accelerating Diffusion Transformers with Token-wise Feature Caching[C]//The Thirteenth International Conference on Learning Representations.
>
> [3] Lv Z, Si C, Song J, et al. FasterCache: Training-Free Video Diffusion Model Acceleration with High Quality[C]//The Thirteenth International Conference on Learning Representations.

---

> > ### Author Response · Authors · 2025-11-30
> >
> > >Q2: Is ETC compatible with DDIM and other high-order samplers such as iPNDM and DPM-Solver++.
> >
> > A2: Since our method focuses on approximating the model output at each step, it can be applied to different solvers. Specifically, for SDXL, we test our method with DDIM, iPNDM, and DPM-Solver++. For other models like FLUX, OpenSora, Wan, and TangoFlux, which predict $v$ rather than noise $\epsilon$, we select Heun[4], UniPC[5], Euler, and DPM++, with the *prediction_type* set to *flow_prediction*. Detailed solver settings can be found in Appendix A.8.3. Additionally, we did not re-tune the threshold values, and instead used the values specified in Table 1 in paper. The quantitative results are shown below.
> >
> > |SDXL|LPIPS↓|SSIM↑|PSNR↑|CLIP↑|FLOPs(p)↓|Speedup↑|Latency(s)↓|
> > |:------: | :------: |:------: |:------: | :------: |:------: |:------: | :------: |
> > |DDIM|-|-|-|27.278|0.67|1×|9.27|
> > |ETC w/DDIM|0.103|0.881|28.106|27.251|0.27|2.15×|4.31|
> > |IPNDM|-|-|-|26.121|0.67|1×|9.31|
> > |ETC w/IPNDM|0.119|0.873|27.963|26.063|0.26|2.01×|4.65|
> > |DPM++|-|-|-|26.668|0.67|1×|9.15|
> > |ETC w/DPM++|0.096|0.882|28.678|26.662|0.24|2.15×|4.26|
> >
> > |FLUX|LPIPS↓|SSIM↑|PSNR↑|CLIP↑|FLOPs(p)↓|Speedup↑|Latency(s)↓|
> > |:------: | :------: |:------: |:------: | :------: |:------: |:------: | :------: |
> > |Heun|-|-|-|25.522|7.36|1×|50.21|
> > |ETC w/Heun|0.053|0.931|29.421|25.301|1.56|4.33×|11.57|
> > |UniPC|-|-|-|25.629|3.71|1×|28.54|
> > |ETC w/UniPC|0.089|0.912|28.411|25.451|1.12|2.81×|10.13|
> > |DPM++|-|-|-|25.734|3.71|1×|28.47|
> > |ETC w/DPM++|0.068|0.922|29.052|25.409|1.26|2.78×|10.23|
> >
> > |OpenSora|LPIPS↓|SSIM↑|PSNR↑|VBench↑|FLOPs(p)↓|Speedup↑|Latency(s)↓|
> > |:------: | :------: |:------: |:------: | :------: |:------: |:------: | :------: |
> > |Heun|-|-|-|77.79%|5.96|1×|82.76|
> > |ETC w/Heun|0.101|0.874|27.566|77.55%|1.98|2.71×|30.60|
> > |UniPC|-|-|-|77.43%|3.08|1×|43.81|
> > |ETC w/UniPC|0.125|0.859|25.565|77.06%|1.49|2.08×|21.04|
> > |DPM++|-|-|-|77.39%|3.08|1×|43.71|
> > |ETC w/DPM++|0.137|0.845|24.114|76.78%|1.09|2.24×|19.43|
> >
> > |Wan|LPIPS↓|SSIM↑|PSNR↑|VBench↑|FLOPs(p)↓|Speedup↑|Latency(s)↓|
> > |:------: | :------: |:------: |:------: | :------: |:------: |:------: | :------: |
> > |Heun|-|-|-|81.01%|150.68|1×|1906.29|
> > |ETC w/Heun|0.105|0.807|24.155|80.73%|46.29|3.16×|603.26|
> > |Euler|-|-|-|81.09%|76.57|1×|966.22|
> > |ETC w/Euler|0.079|0.843|25.293|80.88%|29.03|2.43×|396.81|
> > |DPM++|-|-|-|81.03%|75.39|1×|958.85|
> > |ETC w/DPM++|0.093|0.825|24.867|80.74%|29.16|2.42×|397.11|
> >
> > |TangoFlux|FAD↓|MCD↓|KL↓|CLAP↑|FLOPs(T)↓|Speedup↑|Latency(s)↓|
> > |:------: | :------: |:------: |:------: | :------: |:------: |:------: | :------: |
> > |Heun|-|-|-|13.285|92.85|1×|10.98|
> > |ETC w/Heun|0.027|1.879|0.158|13.273|28.82|2.59×|4.23|
> > |UniPC|-|-|-|13.209|46.86|1×|5.56|
> > |ETC w/UniPC|0.029|1.885|0.149|13.189|14.83|2.45×|2.27|
> > |DPM++|-|-|-|13.276|46.86|1×|5.50|
> > |ETC w/DPM++|0.027|1.880|0.153|13.270|14.83|2.43×|2.26|
> >
> > Our method maintains consistent acceleration and generation quality across different solvers without re-tuning the threshold. This indicates that the threshold we search for captures the error tolerance that prevents trajectory deviation in the model’s outputs. Since this property is intrinsic to the model rather than to the choice of solver, the same threshold remains applicable as long as the model itself is unchanged. More **visual results are provided in Appendix A.8.3**.
> >
> > [4] Karras T, Aittala M, Aila T, et al. Elucidating the design space of diffusion-based generative models[J]. Advances in neural information processing systems, 2022, 35: 26565-26577.
> >
> > [5] Zhao W, Bai L, Rao Y, et al. Unipc: A unified predictor-corrector framework for fast sampling of diffusion models[J]. Advances in Neural Information Processing Systems, 2023, 36: 49842-49869.

---

> > > ### Author Response · Authors · 2025-11-30
> > >
> > > >Q3: Difference in computational overhead between storing all historical trends and the recursive historical trend weighting method.
> > >
> > > A3: We thank the reviewer for raising this important point. Storing all historical trends may provide more detailed trend estimation. However, as diffusion models rely on multi-step denoising, both the computation and memory overhead grow linearly with the number of denoising steps. Let the computational cost of computing the change between two trends be $C$, and the memory required to store a single trend be $M$. At timestep $t$ there are $t-1$ stored historical trends, so the memory footprint is $M(t−1)$. If we estimates the future trend by a weighted sum over all historical trends, the computation required at that timestep becomes $C(t−2)$. In contrast, our recursive historical trend weighting method requires only constant overhead: at any timestep the computation needed to estimate the future trend is $C$, and the memory footprint is fixed at $2M$. This shows that our recursive method reduces both time and space overhead from linear in t to constant.
> > >
> > > To evaluate the computational overhead of our method, we measure both VRAM consumption and FLOPs. In addition, to assess the effectiveness of recursively computing trends, we compare our approach with an alternative method that estimates future trends by storing all historical trends. For this baseline, we apply an exponentially increasing weighting scheme in which older trends receive smaller weights and more recent trends receive larger weights, with all weights sum to 1. The quantitative results are shown below.
> > >
> > > |SDXL|LPIPS↓|SSIM↑|PSNR↑|CLIP↑|FLOPs(M)↓|VRAM(MB)↓|Latency(s)↓|
> > > |:------: | :------: |:------: |:------: | :------: |:------: |:------: | :------: |
> > > |ETC|0.105|**0.876**|28.017|27.103|**9.61**|**0.75**|**4.53**|
> > > |ETC w/all-trend|**0.103**|0.875|**28.025**|**27.105**|35.14|3.44|4.82|
> > >
> > > |FLUX|LPIPS↓|SSIM↑|PSNR↑|CLIP↑|FLOPs(M)↓|VRAM(MB)↓|Latency(s)↓|
> > > |:------: | :------: |:------: |:------: | :------: |:------: |:------: | :------: |
> > > |ETC|**0.068**|**0.926**|29.176|**25.983**|**25.83**|**1.01**|**10.86**|
> > > |ETC w/all-trend|0.070|0.925|**29.177**|25.976|67.72|6.73|11.35|
> > >
> > > |OpenSora|LPIPS↓|SSIM↑|PSNR↑|VBench↑|FLOPs(M)↓|VRAM(MB)↓|Latency(s)↓|
> > > |:------: | :------: |:------: |:------: | :------: |:------: |:------: | :------: |
> > > |ETC|0.131|**0.848**|**24.123**|**76.44%**|**23.65**|**18.34**|**21.07**|
> > > |ETC w/all-trend|**0.128**|0.848|24.119|76.44%|74.03|39.22|23.51|
> > >
> > > |Wan|LPIPS↓|SSIM↑|PSNR↑|VBench↑|FLOPs(M)↓|VRAM(MB)↓|Latency(s)↓|
> > > |:------: | :------: |:------: |:------: | :------: |:------: |:------: | :------: |
> > > |ETC|**0.103**|**0.806**|24.078|**80.71%**|**211.76**|**40.99**|**389.89**|
> > > |ETC w/all-trend|0.105|0.805|**24.101**|80.69%|624.79|104.21|397.34|
> > >
> > > |TangoFlux|FAD↓|MCD↓|KL↓|CLAP↑|FLOPs(M)↓|VRAM(MB)↓|Latency(s)↓|
> > > |:------: | :------: |:------: |:------: | :------: |:------: |:------: | :------: |
> > > |ETC|**0.026**|**1.877**|0.157|13.251|**4.74**|**0.94**|**2.26**|
> > > |ETC w/all-trend|0.028|1.877|**0.155**|**13.252**|11.51|3.31|2.53|
> > >
> > > Our method requires only a few of megabytes of additional VRAM during diffusion inference, and its computational cost is negligible compared with the overall diffusion model inference. This demonstrates that our approach is computation-efficient. Moreover, compared with the method that stores all historical trends to estimate future trends, our recursive formulation achieves comparable acceleration performance and generation quality while reducing both computational cost and memory consumption by more than 50\%, mitigating the overhead introduced by trend computation.

---

> > > > ### Author Response · Authors · 2025-11-30
> > > >
> > > > >Q4: Hyper parameters $\alpha$ and $n$ depend on being specified manually. Does an adaptive optimization strategy for $\alpha$ exist.
> > > >
> > > > A4: Thank you for your insightful question. Since the hyper parameters $\alpha$ and $n$ are manually specified, the optimal values may vary under different generation settings (e.g., different prompts). Therefore, we also explored adaptive strategies for $\alpha$ and $n$.
> > > >
> > > > The purpose of setting $n$ is to ensure that our acceleration method is applied only after the diffusion model reaches a stable denoising phase. The threshold $\sigma$ is obtained by analyzing the transition point between the semantic planning stage and the stable quality-improvement stage of the diffusion process. Therefore, by checking whether the difference between model outputs at adjacent timesteps falls below this threshold, we can determine when to start acceleration and achieve a dynamic $n$.
> > > >
> > > > For $\alpha$, when the difference between the model output and the previous approximation is equal to the estimated trend (i.e., their difference is zero), it indicates that the estimated trend perfectly matches the original denoising trajectory. When the difference becomes larger, it suggests that the current $\alpha$ is insufficient and should be updated in the opposite direction. Conversely, when the difference becomes smaller, it indicates that $\alpha$ is being updated in the correct direction and should continue to move along that trajectory. Based on this, we can design an adaptive $\alpha$ update strategy driven by the gradient of the change. Additionally, $\alpha$ can be set as a matrix of the same size as latent, enabling distinct trend updates for each pixel. The pseudocode for adaptive $n$ and $\alpha$ is provided in Algorithm 3 in Appendix.
> > > >
> > > > ||LPIPS↓|SSIM↑|PSNR↑|CLIP↑|FLOPs(P)↓|SpeedUp↑|Latency(s)↓|
> > > > |:------: | :------: |:------: |:------: | :------: |:------: |:------: | :------: |
> > > > |SDXL|-|-|-|27.253|0.67|1×|9.63|
> > > > |ETC|0.105|0.876|**28.017**|27.103|0.26|2.12×|4.53|
> > > > |ETC-adaptive|**0.101**|**0.883**|28.009|**27.129**|**0.26**|**2.16×**|**4.44**|
> > > >
> > > > ||LPIPS↓|SSIM↑|PSNR↑|CLIP↑|FLOPs(P)↓|SpeedUp↑|Latency(s)↓|
> > > > |:------: | :------: |:------: |:------: | :------: |:------: |:------: | :------: |
> > > > |FLUX|-|-|-|26.319|3.64|1×|28.85|
> > > > |ETC|0.068|0.926|29.176|25.983|1.31|2.65×|10.86|
> > > > |ETC-adaptive|**0.061**|**0.928**|29.211|**25.989**|**1.29**|**2.74×**|**10.51**|
> > > >
> > > > ||LPIPS↓|SSIM↑|PSNR↑|VBench↑|FLOPs(P)↓|SpeedUp↑|Latency(s)↓|
> > > > |:------: | :------: |:------: |:------: | :------: |:------: |:------: | :------: |
> > > > |OpenSora|-|-|-|77.18%|3.15|1×|45.29|
> > > > |ETC|**0.131**|0.848|24.123|76.44%|**1.32**|**2.15×**|21.07|
> > > > |ETC-adaptive|0.132|**0.852**|24.235|**76.58%**|1.32|2.15×|**21.02**|
> > > >
> > > > ||LPIPS↓|SSIM↑|PSNR↑|VBench↑|FLOPs(P)↓|SpeedUp↑|Latency(s)↓|
> > > > |:------: | :------: |:------: |:------: | :------: |:------: |:------: | :------: |
> > > > |Wan|-|-|-|81.01%|76.71|1×|970.49|
> > > > |ETC|**0.103**|0.806|24.078|80.71%|28.63|2.49×|389.89|
> > > > |ETC-adaptive|0.104|**0.812**|**24.101**|**80.92%**|**28.54**|**2.50×**|**388.34**|
> > > >
> > > > ||FAD↓|MCD↓|KL↓|CLAP↑|FLOPs(T)↓|SpeedUp↑|Latency(s)↓|
> > > > |:------: | :------: |:------: |:------: | :------: |:------: |:------: | :------: |
> > > > |TangoFlux|-|-|-|13.286|46.86|1×|5.49|
> > > > |ETC|0.026|1.877|0.157|13.251|14.83|**2.43×**|2.26|
> > > > |ETC-adaptive|**0.022**|**1.796**|**0.156**|**13.263**|**14.76**|2.43×|**2.25**|
> > > >
> > > > We observe a slight improvement in generation quality after introducing the dynamic strategy. This suggests that although our method is generally insensitive to the choice of $\alpha$ (as shown by Table 4 in the paper, where $\alpha=0.5$ already yields well acceleration performance), adapting $\alpha$ to different levels of accumulated error can further enhance generation quality. Our simple adaptive mechanism already provides gains, indicating that future work could build on this direction by more deeply analyzing how model-output deviations under different error accumulations influence future denoising trajectories. Such analysis could guide more principled adjustments of the trend-update magnitude.

---

> > > > > ### Author Response · Authors · 2025-11-30
> > > > >
> > > > > >Q5: The description of the search for model-specific error tolerance is relatively vague. Specific workflows are not provided for models like the Trend Inflection Point Analysis Model.
> > > > >
> > > > > A5: Thank you for pointing out this issue. We have updated the Method section of the paper to more clearly explain the motivation and underlying principles of our model-specific error tolerance strategy. In addition, we provide the complete workflow of the Trend Inflection Point Analysis Model in Appendix A.11, detailing each step of the procedure. We appreciate the reviewer’s suggestion, which has helped us improve the clarity and completeness of the manuscript.
> > > > >
> > > > > >Q6: Some parameter names are incorrect. For example, in the title of Table 4, $p$ should be $n$.
> > > > >
> > > > > A6: Thanks for your careful and accurate reading of our paper. We have fixed the errors in the revised version.

---

### Official Review · Reviewer_sJqw · 2025-10-31

**Soundness:** 3
**Presentation:** 3
**Contribution:** 2
**Rating:** 4
**Confidence:** 4

**Summary:**

The proposed ETC (Error-aware Trend Consistency) is a training-free method that speeds up diffusion-model sampling by (i) recursively smoothing all past noise predictions to form a Consistent Trend Predictor, then evenly applying this forecast across the next k timesteps to skip costly forward passes, and (ii) searching a model-specific error threshold by locating the breakpoint between the semantic-planning and quality-refinement phases of denoising, which adaptively controls the skip window. Tested on SDXL and Flux, Open-Sora 1.2 and Wan 2.1, and TangoFlux, ETC delivers up to 2.65× faster inference with virtually no perceptual degradation.

**Strengths:**

1.	The paper is clearly written.

2.	The approach requires no retraining and no architectural changes, yet shows consistent gains across diverse generation task.

**Weaknesses:**

1.	The error threshold is obtained through offline search. Its robustness to new resolutions, schedulers, or unseen models is unclear. An online or learned adaptation mechanism may strengthen the claim of generality.

2.	The proposed trend-reuse idea is conceptually close to TeaCache and SADA, and its two-phase treatment of the denoising trajectory ( semantic-planning vs. quality-refinement ) resembles the block-specific reuse strategy in BlockDance. However, the paper gives little theoretical justification or quantitative ablation that clarifies how ETC departs from— or improves upon— these predecessors, leaving the boundary of novelty and the precise source of its gains insufficiently articulated.

3.	This paper lacks comparison with training-based accelerators (e.g., progressive/step distillation), making it hard to judge how much of ETC’s benefit stems from being training-free. Moreover, all experiments cap the speed-up at ~2–2.6 ×; there is no analysis of more aggressive settings (3–5 ×), so it remains unclear whether ETC experiences significant quality collapse under higher acceleration.

**Questions:**

N/A

---

> ### Author Response · Authors · 2025-11-30
>
> # Table 1. Quantitative results using different schedulers.
>
> |SDXL|LPIPS↓|SSIM↑|PSNR↑|CLIP↑|FLOPs(p)↓|Speedup↑|Latency(s)↓|
> |:------: | :------: |:------: |:------: | :------: |:------: |:------: | :------: |
> |DDIM|-|-|-|27.278|0.67|1×|9.27|
> |ETC w/DDIM|0.103|0.881|28.106|27.251|0.27|2.15×|4.31|
> |IPNDM|-|-|-|26.121|0.67|1×|9.31|
> |ETC w/IPNDM|0.119|0.873|27.963|26.063|0.26|2.01×|4.65|
> |DPM++|-|-|-|26.668|0.67|1×|9.15|
> |ETC w/DPM++|0.096|0.882|28.678|26.662|0.24|2.15×|4.26|
>
> |FLUX|LPIPS↓|SSIM↑|PSNR↑|CLIP↑|FLOPs(p)↓|Speedup↑|Latency(s)↓|
> |:------: | :------: |:------: |:------: | :------: |:------: |:------: | :------: |
> |Heun|-|-|-|25.522|7.36|1×|50.21|
> |ETC w/Heun|0.053|0.931|29.421|25.301|1.56|4.33×|11.57|
> |UniPC|-|-|-|25.629|3.71|1×|28.54|
> |ETC w/UniPC|0.089|0.912|28.411|25.451|1.12|2.81×|10.13|
> |DPM++|-|-|-|25.734|3.71|1×|28.47|
> |ETC w/DPM++|0.068|0.922|29.052|25.409|1.26|2.78×|10.23|
>
> |OpenSora|LPIPS↓|SSIM↑|PSNR↑|VBench↑|FLOPs(p)↓|Speedup↑|Latency(s)↓|
> |:------: | :------: |:------: |:------: | :------: |:------: |:------: | :------: |
> |Heun|-|-|-|77.79%|5.96|1×|82.76|
> |ETC w/Heun|0.101|0.874|27.566|77.55%|1.98|2.71×|30.60|
> |UniPC|-|-|-|77.43%|3.08|1×|43.81|
> |ETC w/UniPC|0.125|0.859|25.565|77.06%|1.49|2.08×|21.04|
> |DPM++|-|-|-|77.39%|3.08|1×|43.71|
> |ETC w/DPM++|0.137|0.845|24.114|76.78%|1.09|2.24×|19.43|
>
> |Wan|LPIPS↓|SSIM↑|PSNR↑|VBench↑|FLOPs(p)↓|Speedup↑|Latency(s)↓|
> |:------: | :------: |:------: |:------: | :------: |:------: |:------: | :------: |
> |Heun|-|-|-|81.01%|150.68|1×|1906.29|
> |ETC w/Heun|0.105|0.807|24.155|80.73%|46.29|3.16×|603.26|
> |Euler|-|-|-|81.09%|76.57|1×|966.22|
> |ETC w/Euler|0.079|0.843|25.293|80.88%|29.03|2.43×|396.81|
> |DPM++|-|-|-|81.03%|75.39|1×|958.85|
> |ETC w/DPM++|0.093|0.825|24.867|80.74%|29.16|2.42×|397.11|
>
> |TangoFlux|FAD↓|MCD↓|KL↓|CLAP↑|FLOPs(T)↓|Speedup↑|Latency(s)↓|
> |:------: | :------: |:------: |:------: | :------: |:------: |:------: | :------: |
> |Heun|-|-|-|13.285|92.85|1×|10.98|
> |ETC w/Heun|0.027|1.879|0.158|13.273|28.82|2.59×|4.23|
> |UniPC|-|-|-|13.209|46.86|1×|5.56|
> |ETC w/UniPC|0.029|1.885|0.149|13.189|14.83|2.45×|2.27|
> |DPM++|-|-|-|13.276|46.86|1×|5.50|
> |ETC w/DPM++|0.027|1.880|0.153|13.270|14.83|2.43×|2.26|
>
> # Table 2. Quantitative results across different resolutions.
>
> |Text to Image|LPIPS↓|SSIM↑|PSNR↑|CLIP↑|FLOPs(p)↓|Speedup↑|Latency(s)↓|
> |:------: | :------: |:------: |:------: | :------: |:------: |:------: | :------: |
> |SDXL(768×1344)|-|-|-|27.228|0.67|1×|9.28|
> |ETC|0.114|0.873|27.937|27.045|0.24|2.11×|4.38|
> |SDXL(1152×896)|-|-|-|27.221|0.67|1×|9.31|
> |ETC|0.103|0.879|28.113|27.048|0.23|2.12×|4.37|
> |FLUX(768×1344)|-|-|-|26.429|3.67|1×|28.65|
> |ETC|0.071|0.914|29.093|26.151|1.26|2.69×|10.68|
> |FLUX(1152×896)|-|-|-|26.139|3.67|1×|28.59|
> |ETC|0.069|0.919|29.189|26.058|1.25|2.68×|10.62|
>
> |Text to Video|LPIPS↓|SSIM↑|PSNR↑|VBench↑|FLOPs(p)↓|Speedup↑|Latency(s)↓|
> |:------: | :------: |:------: |:------: | :------: |:------: |:------: | :------: |
> |OpenSora(360p,4:3)|-|-|-|76.89%|1.76|1×|25.86|
> |ETC|0.125|0.868|25.657|76.63%|0.76|2.14×|12.16|
> |OpenSora(720p,1:1)|-|-|-|77.22%|6.75|1×|105.88|
> |ETC|0.151|0.853|24.624|76.98%|2.02|2.23×|46.52|
> |Wan(720×1280)|-|-|-|82.79%|182.95|1×|3456.53|
> |ETC|0.089|0.853|26.763|82.56%|68.83|2.78×|1218.98|
>
> # Table 3. Quantitative results by using $\sigma$ online search .
>
> |Model|$\sigma$ update round|1|2|3|4|5|6|7|
> |:-----:|:-----:|:-----:|:-----:|:-----:|:-----:|:-----:|:-----:|:-----:|
> |SDXL|$\sigma$-search|0.0096|0.0149|0.0108|0.0187|0.0193|0.0203|0.0151|
> ||$\sigma$-mean|0.0096|0.0122|0.0115|0.0151|0.0172|0.0187|0.0169|
> ||SSIM↑|0.883|0.879|0.880|0.878|0.876|0.874|0.876|
> ||CLIP↑|27.159|27.122|27.129|27.110|27.105|27.103|27.110|
> ||Latency(s)|5.05|4.79|4.82|4.65|4.55|4.31|4.56|
> |FLUX|$\sigma$-search|0.0986|0.1284|0.1032|0.1411|0.1314|0.1102|0.1417|
> ||$\sigma$-mean|0.0986|0.1135|0.1083|0.1247|0.1281|0.1191|0.1304|
> ||SSIM↑|0.928|0.927|0.927|0.926|0.926|0.926|0.925|
> ||CLIP↑|26.279|26.087|26.281|25.983|25.982|25.983|25.981|
> ||Latency(s)|11.89|11.02|11.78|10.82|10.85|10.83|10.56|
> |OpenSora|$\sigma$-search|0.1841|0.2327|0.1747|0.2505|0.1960|0.2171|0.2006|
> ||$\sigma$-mean|0.1841|0.2084|0.1915|0.2210|0.2085|0.2128|0.2067|
> ||SSIM↑|0.852|0.848|0.850|0.845|0.848|0.848|0.848|
> ||VBench↑|76.63|76.44%|76.52%|76.40%|76.44%|76.44%|76.44|
> ||Latency(s)|22.23|21.09|21.96|20.98|21.10|21.09|21.05|
> |Wan|$\sigma$-search|0.0879|0.1139|0.1016|0.1361|0.1289|0.0977|0.1248|
> ||$\sigma$-mean|0.0879|0.1009|0.1012|0.1186|0.1237|0.1107|0.1177|
> ||SSIM↑|0.821|0.810|0.810|0.806|0.793|0.806|0.806|
> ||VBench↑|80.72%|80.71%|80.71%|80.71%|80.61%|80.71%|80.71%|
> ||Latency(s)|495.21|403.55|403.67|392.22|382.19|391.02|390.12|
> |TangoFlux|$\sigma$-search|0.0595|0.0756|0.0621|0.0641|0.0701|0.0621|0.0658|
> ||$\sigma$-mean|0.0595|0.0675|0.0648|0.0644|0.0672|0.0646|0.0652|
> ||KL↓|0.148|0.157|0.155|0.155|0.157|0.155| 0.156|
> ||CLAP↑|13.260|13.251|13.252|13.252|13.251|13.252|13.251|
> ||Latency(s)|2.41|2.23|2.32|2.34|2.25|2.33|2.32|

---

> > ### Author Response · Authors · 2025-11-30
> >
> > >Q1: Robustness of error threshold to new resolutions, schedulers and unseen models, and an online or learned adaptation mechanism of error threshold.
> >
> > A1: We thank the reviewer for raising this valuable question. We agree that evaluating our method under different resolutions and schedulers provides a more comprehensive assessment of its robustness. Therefore, we also conduct experiments under different resolution settings and different schedulers.
> >
> > For SDXL, we test our method with DDIM[1], iPNDM[2], and DPM-Solver++[3]. For FLUX, OpenSora, Wan, and TangoFlux, which predict $v$ rather than noise $\epsilon$, we select Heun[4], UniPC[5], Euler, and DPM++, with the *prediction_type* set to *flow_prediction*. Detailed solver settings can be found in Appendix A.8.3. Additionally, we did not re-tune the threshold values, and instead used the values specified in Table 1 in paper.
> >
> > For image generation models, we use resolutions of 768×1344 and 1152×896. For video generation models, OpenSora is tested with 360p (4:3) and 720p (1:1) settings, while Wan is tested at a resolution of 720×1280.
> >
> > As shown in Table 1 and Table 2, our approach consistently maintains similar acceleration performance and generation quality across these varied settings. This indicates that the searched threshold effectively captures the model’s inherent error tolerance that prevents trajectory deviation in its outputs. Because this property is tied to the model itself rather than to any particular solver, the same threshold remains applicable as long as the underlying model is unchanged. We provide more visualization results in Appendices A.8.3 and A.8.4.
> >
> > For an unseen model, its error tolerance differs from that of previously evaluated models, so the threshold must be re-searched accordingly. However, our experiments also show that the threshold found from a small amount of data is robust. This suggests that our method can be efficiently and reliably deployed on new models with minimal additional effort. Therefore, we designed an online method to validate the deployability of our threshold search approach. Specifically, we randomly sample and combine seven conditions across different samplers, resolutions, and prompts. For each condition, we perform threshold searching and store the resulting value. Whenever a new threshold is obtained, we update the global threshold by taking the average of the thresholds, and then evaluate the acceleration performance under this updated threshold.
> >
> > As shown in Table 3, the threshold obtained from the first data already provides a general acceleration effect and generation quality across models. As more online search results are accumulated, the threshold gradually converges to a universal value, which also lies near the one obtained by our offline method. This indicates that threshold search performed on a small amount of data is robust, enabling rapid deployment on new unseen models.
> >
> > [1] Song J, Meng C, Ermon S. Denoising diffusion implicit models[J]. arXiv preprint arXiv:2010.02502, 2020.
> >
> > [2] Luping Liu, Yi Ren, Zhijie Lin, and Zhou Zhao. Pseudo numerical methods for diffusion models on manifolds. arXiv preprint arXiv:2202.09778, 2022.
> >
> > [3] Lu C, Zhou Y, Bao F, et al. Dpm-solver++: Fast solver for guided sampling of diffusion probabilistic models[J]. Machine Intelligence Research, 2025: 1-22.
> >
> > [4] Karras T, Aittala M, Aila T, et al. Elucidating the design space of diffusion-based generative models[J]. Advances in neural information processing systems, 2022, 35: 26565-26577.
> >
> > [5] Zhao W, Bai L, Rao Y, et al. Unipc: A unified predictor-corrector framework for fast sampling of diffusion models[J]. Advances in Neural Information Processing Systems, 2023, 36: 49842-49869.

---

> > > ### Author Response · Authors · 2025-11-30
> > >
> > > >Q2: Idea is conceptually close to TeaCache and SADA, and two-phase treatment resembles BlockDance.
> > >
> > > A2: We thank the reviewer for raising this important question. Although our method shares a high-level motivation with TeaCache and SADA—accelerating diffusion inference through step-wise reuse of model output trends—our approach differs from these methods in two key aspects:
> > > (1) how to obtain a more robust estimate of future trends to ensure that approximation errors do not diverge under high acceleration, and (2) how to evaluate the maximum allowable approximation steps so as to achieve the highest possible acceleration without incurring significant quality degradation.
> > >
> > > For TeaCache, the future trend is estimated only from the difference between two adjacent timesteps, giving it a window of size two. Consequently, short-term fluctuations can be amplified, causing trajectory deviations (Figure 1(a) in paper) and quality drops (Figure 5 in paper). In contrast, our method captures the overall evolution across all historical trends, which helps counteract such short-term fluctuations and yields a more stable prediction of future trends. We replace TeaCache’s trend estimator with our consistent trend predictor, and the quantitative results are shown below.
> > >
> > > ||LPIPS↓|SSIM↑|PSNR↑|CLIP↑|FLOPs(P)↓|Speedup↑|Latency(s)↓|
> > > |:-:|:-:|:-:|:-:|:-:|:-:|:-:|:-:|
> > > |FLUX|-|-|-|26.319|3.64|1×|28.85|
> > > |TeaCache-fast|0.281|0.753|18.845|25.924|1.45|2.45×|11.78|
> > > |ETC|0.068|0.926|29.176|25.983|1.31|2.65×|10.86|
> > > |TeaCache-fast+ETC-trend|0.066|0.928|29.185|25.992|1.69|2.21×|13.15|
> > >
> > > We observe that integrating our module leads to noticeable improvements in TeaCache’s generation quality and consistency. However, our method is applied during the denoising stability stage, whereas TeaCache uses time embeddings to indicate stability and does not evaluate whether additional trend reuse is feasible for achieving higher acceleration. As a result, the combined system does not gain substantial speed improvements. In contrast, our approach offers a different perspective on exploring the limits of acceleration. By assessing whether the accumulated approximation error remains within the range that the model can still correct, we dynamically adjust the number of approximations. This allows us to reach the highest possible acceleration while maintaining reliable output quality.
> > >
> > > For SADA, the method relies on a third-order finite-difference scheme to assess trend stability and perform approximations. This restricts the number of reusable steps to at most four, which preserves generation quality but results in only limited acceleration. To address this limitation, we replace SADA’s stability-checking module with our error threshold mechanism. The experimental results are presented below.
> > >
> > > ||LPIPS↓|SSIM↑|PSNR↑|CLIP↑|FLOPs(P)↓|Speedup↑|Latency(s)|
> > > |:-:|:-:|:-:|:-:|:-:|:-:|:-:|:-:|
> > > |SDXL|-|-|-|27.253|0.67|1×|9.63|
> > > |SADA|0.096|0.882|28.881|27.055|0.35|1.81×|5.07|
> > > |ETC|0.105|0.876|28.017|27.103|0.26|2.12×|4.53|
> > > |SADA+ETC-threshold|0.102|0.880|28.832|27.076|0.32|1.95×|4.93|
> > >
> > > ||LPIPS↓|SSIM↑|PSNR↑|CLIP↑|FLOPs(P)↓|Speedup↑|Latency(s)|
> > > |:-:|:-:|:-:|:-:|:-:|:-:|:-:|:-:|
> > > |FLUX|-|-|-|26.319|3.64|1×|28.85|
> > > |SADA|0.062|0.923|29.342|25.979|1.74|2.07×|13.94|
> > > |ETC|0.068|0.926|29.176|25.983|1.31|2.65×|10.86|
> > > |SADA+ETC-threshold|0.064|0.922|29.287|25.976|1.65|2.22×|12.97|
> > >
> > > We observe that the acceleration improves without introducing noticeable quality degradation, indicating that under SADA’s third-order finite-difference approximation, the accumulated error does not exceed the level that would cause significant quality collapse. This suggests that additional speedup is still possible. However, because the third-order finite-difference scheme incorporates the approximation errors from earlier reused steps, the estimated trend becomes increasingly inaccurate as more steps are approximated, which ultimately limits the achievable acceleration. In contrast, our method computes the trend solely from the historical model outputs—without including any approximated values—allowing it to produce more reliable trend estimates. As a result, the accumulated error across multiple approximations remains within the acceptable error tolerance, enabling substantially higher acceleration.
> > >
> > > Regarding the similarity to BlockDance’s reuse strategy, although both approaches accelerate sampling during the stable phase of denoising, our method operates at the model-output level, while BlockDance reuses internal features within a DiT model. Consequently, our approach is architecture-agnostic and can be applied to various diffusion backbones. Since our two-stage formulation is based on observations rather than being a core contribution, we have moved this discussion to the preliminary section for clarity. We thank the reviewer for the helpful comment.

---

> > > > ### Author Response · Authors · 2025-11-30
> > > >
> > > > >Q3: Comparison with training-based accelerators, and quality stable under higher acceleration.
> > > >
> > > > A3: We thank the reviewer for suggesting this important experiment. Since a trained model cannot guarantee generation consistency with the original model, we focus our evaluation on generation quality and inference speed. For SDXL, we use SDXL-Lightning[6], and for FLUX, we use Flux-Turbo[7]. Since widely adopted few-step models for OpenSora, Wan and TangoFlux are not yet publicly available, we omit those experiments for now. In addition, we introduce the MPS[8] metric to evaluate the quality of generated images in terms of attributes such as color, style, and texture. The quantitative results are shown below, and more visualizations are provided in **Appendix A.8.2**.
> > > >
> > > > |Method|MPS↑|CLIP↑|Latency~(s)↓|Speedup↑|
> > > > |:------:|:------:|:------:|:------:|:------:|
> > > > |SDXL (T=50)|10.768|**27.253**|9.63|1×|
> > > > |ETC (T=50)|10.701|27.103|4.53|2.12×|
> > > > |SDXL-Lightning (T=8)|**12.003**|26.517|1.41|6.82×|
> > > > |SDXL-Lightning+ETC (T=8)|11.957|26.458|**1.08**|**8.91×**|
> > > > |FLUX (T=50)|**12.941**|**26.319**|28.85|1×|
> > > > |ETC (T=50)|12.898|25.983|10.86|2.65×|
> > > > |FLUX-turbo (T=8)|12.884|25.968|4.87|5.92×|
> > > > |FLUX-turbo+ETC (T=8)|12.852|25.891|**3.42**|**8.43×**|
> > > >
> > > > Although the trained few-step models achieve substantial improvements in generation speed, they exhibit a noticeable decrease in CLIP scores compared with their multi-step counterparts. While SDXL-Lightning emphasizes aesthetics during training and exceeds the multi-step model on the MPS metric, it also introduces structural distortions—for example, merged human limbs or mixed deer antlers and ears, as illustrated in Figure 14 in Appendix. FLUX-Turbo shows similar issues, such as abnormal hand generation. Moreover, training-based approaches require significant hardware resources. For instance, SDXL-Lightning is trained on 64 A100 80GB GPUs with a batch size of 512. In contrast, our method is friendly to consumer-level hardware while preserving both generation quality and inference speed. Additionally, our approach can be combined with few-step models to achieve an extra 20%–30% acceleration while maintaining consistent outputs, demonstrating its effectiveness and general applicability.
> > > >
> > > > For the quality degradation that may occur under high acceleration settings, we investigate this in the fixed error threshold ablation study. By increasing the threshold, we achieve more than a 3× speedup, and as shown in Figure 8 of the paper, our method still preserves overall structural consistency, with differences appearing mainly in fine details. Building on this, we increase the approximation steps in the first trend-approximation round (raising $k$ from 1 to 4) to obtain even higher acceleration. The quantitative results are shown below.
> > > >
> > > > ||LPIPS↓|SSIM↑|MPS↑|CLIP↑|FLOPs(P)↓|Speedup↑|Latency(s)|
> > > > |:-----:|:-----:|:-----:|:-----:|:-----:|:-----:|:-----:|:-----:|
> > > > |SDXL|-|-|10.768|27.253|0.67|1×|9.63|
> > > > |ETC|0.105|0.876|10.701|27.103|0.26|2.12×|4.53|
> > > > |ETC-fast|0.336|0.694|10.679|26.898|0.12|4.91×|1.88|
> > > >
> > > > ||LPIPS↓|SSIM↑|MPS↑|CLIP↑|FLOPs(P)↓|Speedup↑|Latency(s)|
> > > > |:-----:|:-----:|:-----:|:-----:|:-----:|:-----:|:-----:|:-----:|
> > > > |FLUX|-|-|12.941|26.319|3.64|1×|28.85|
> > > > |ETC|0.068|0.926|12.898|25.983|1.31|2.65×|10.86|
> > > > |ETC-fast|0.303|0.766|12.871|25.676|0.75|4.63×|6.25|
> > > >
> > > > ||LPIPS↓|SSIM↑|MPS↑|VBench↑|FLOPs(P)↓|Speedup↑|Latency(s)|
> > > > |:-----:|:-----:|:-----:|:-----:|:-----:|:-----:|:-----:|:-----:|
> > > > |Open-Sora|-|-|-|77.18%|3.15|1×|45.29|
> > > > |ETC|0.131|0.848|24.123|76.44%|1.32|2.15×|21.07|
> > > > |ETC-fast|0.591|0.492|10.967|76.29%|0.797|4.06×|10.67|
> > > >
> > > > ||LPIPS↓|SSIM↑|MPS↑|VBench↑|FLOPs(P)↓|Speedup↑|Latency(s)|
> > > > |:-----:|:-----:|:-----:|:-----:|:-----:|:-----:|:-----:|:-----:|
> > > > |Wan|-|-|-|81.01%|76.71|1×|970.49|
> > > > |ETC|0.103|0.806|24.078|80.71%|28.63|2.49×|389.89|
> > > > |ETC-fast|0.299|0.603|18.193|80.48%|17.58|4.49×|215.67|
> > > >
> > > > Although consistency drops significantly under this higher acceleration setting, the overall generation quality remains stable—both the CLIP score and the VBench score stay at high levels. **Figure 18 in the Appendix** shows the generated results under high acceleration., demonstrating that our method does not suffer from noticeable quality collapse even in this regime, indicating robustness and acceleration potential.
> > > >
> > > > [6]Shanchuan Lin, Anran Wang, and Xiao Yang. Sdxl-lightning: Progressive adversarial diffusion distillation, 2024.
> > > >
> > > > [7]Alibaba. Flux turbo. https://huggingface.co/alimama-creative/FLUX.1-Turbo-Alpha, 2024.
> > > >
> > > > [8]Sixian Zhang, Bohan Wang, Junqiang Wu, Yan Li, Tingting Gao, Di Zhang, and Zhongyuan Wang. Learning multi-dimensional human preference for text-to-image generation. In Proceedings of the IEEE/CVF Conference on Computer Vision and Pattern Recognition, pp. 8018–8027, 2024.

---

### Official Review · Reviewer_XXmn · 2025-10-31

**Soundness:** 3
**Presentation:** 3
**Contribution:** 3
**Rating:** 6
**Confidence:** 3

**Summary:**

This paper proposes ETC (Error-aware Trend Consistency), a training-free method to accelerate diffusion/flow model sampling by skipping some denoiser calls while preserving output consistency. ETC (i) aggregates historical inter-step model output changes with exponential smoothing to form a trend predictor and progressively distributes the predicted delta across a short window of skipped steps, and (ii) uses a model-specific error/tolerance search to adaptively decide when and how far to glide (skip) before re-evaluating the denoiser. Experiments on images (e.g., SDXL/FLUX), video (e.g., Open-Sora/Wan), and audio (e.g., TangoFlux) show ~2×–2.6× speedups with small drops in performance, and favorable comparisons against training-free reuse based baselines (TeaCache, SADA, MagCache, AdaptiveDiffusion).

**Strengths:**

- Practical, training-free modification; no retraining or distillation required.
- Conceptual advance over step-wise reuse: uses all-history, smoothed trend prediction rather than just the latest pair; progressive distribution mitigates drift during multi-step reuse.
- Adaptive skip length via a model-specific tolerance, more robust than fixed global thresholds.
- Cross-modality and cross-backbone evaluation (image/video/audio) with consistent speed vs. quality improvements; strong empirical appeal.
- Sensible ablations (tolerance sweeps, stability/error accumulation, sensitivity) and clear intuition for early/mid vs. late-step behavior.

**Weaknesses:**

- It seems that the main results of the paper rely on the assumption that $\epsilon_\theta(x,t,c)$ changes smoothly in $x$ and $t$, but this isn’t formally proven here. That’s fine for an empirical paper, yet it means there’s no hard guarantee, but only evidence from experiments.
- Algorithm/equation mismatch: There is a mismatch on the usage on the usage of delta in text and in the pseudo algorithm. It is not clear which one is implemented and it seems that it would make a difference in results depending on the option.
- Minor polish: Notation/indexing inconsistencies for $d$

**Questions:**

1) If you pick $\sigma$ on one small prompt set, does it transfer to out-of-distribution prompts (e.g., dense text or multi-object scenes) without retuning?

2) At equal perceptual quality (e.g. same FID score), which is faster: ETC or simply reducing steps with for example DDIM solver?

3) Table  4 shows SSIM sensitivity to $n$ but the latency impact of different $n$ is not reported. Could you add (or at least state) the corresponding wall‑clock changes for different values and give a simple "rule‑of‑thumb" for picking $n$?

---

> ### Author Response · Authors · 2025-11-30
>
> >Q1: Proven of $\epsilon_\theta(x,t,c)$ smoothly in $x$ and $t$.
>
> A1:
> We thank the reviewer for raising this important point. Since the forward diffusion process injects noise of different magnitudes into $x$ as time $t$ increases, we can prove that the model’s reverse-process output $\epsilon_{\theta}$is jointly Lipschitz continuous in both $x_t$ and $t$. Assuming the forward process is given by $x_t = \alpha(t)\cdot x_0+\beta(t)\cdot \epsilon$, the noise at timestep $t$ during denoising can be written as:
>
> $\epsilon = \frac{1}{\beta(t)}x_t-\frac{\alpha(t)}{\beta(t)}x_0$ (1).
>
> Assume that the model predicts the noise accurately at timestep $t$:
>
> $\epsilon_{\theta} = \frac{1}{\beta(t)}x_t-\frac{\alpha(t)}{\beta(t)}x_0$  (2).
>
> Then for two timesteps $t$ and $t'$, the difference between outputs satisfies:
>
> $||\epsilon_\theta(x_{t},t,c)-\epsilon_\theta(x_{t'},t',c)|| = ||\frac{x_{t}-\alpha(t)x_0}{\beta(t)}-\frac{x_{t'}-\alpha(t')x_0}{\beta(t')}||\leq L_x||x_t-x_{t'}||+L_t||t-t'||$ (3),
>
> where $L_x=\frac{1}{\beta_{min}},L_t=K_1K_\beta+\frac{K_0K\alpha}{\beta_{min}}.$ The full derivation is provided in Appendix A.4. Equation (3) implies that $\epsilon_{\theta}(x_t,t,c)$ is Lipschitz continuous in both $x_t$ and $t$, confirming its smooth evolution throughout the denoising process. We appreciate the reviewer’s suggestion, as this clarification improves the rigor of our analysis. We have incorporated the formal statement of this Lipschitz continuity assumption into the preliminary section of the paper.
>
> >Q2: Can $\sigma$ transfer to out-of-distribution prompts (e.g., dense text or multi-object) without retuning.
>
> We thank the reviewer for this insightful question. Evaluating the searched threshold $\sigma$ on out-of-distribution prompts is important for understanding its robustness. To examine this, we conducted quantitative experiments on the evaluation set of TextAtlas5M[1], a benchmark specifically designed for dense-text generation. The results are shown below.
>
> ||LPIPS↓|SSIM↑|PSNR↑|CLIP↑|FLOPs(P)↓|Speedup↑|Latency(s)↓|
> |:-:|:-:|:-:|:-:|:-:|:-:|:-:|:-:|
> |SDXL|-|-|-|30.584|0.67|1×|9.82|
> |ETC|0.109|0.865|27.958|30.893|0.28|1.98×|5.08|
> |FLUX|-|-|- |32.547|3.64|1×|29.19|
> |ETC|0.083|0.916|28.344|32.286|1.31|2.51×|11.49|
>
> While our method maintains consistent generation quality, the generation time increases slightly. This is because complex scenarios are more prone to trajectory deviation, resulting in more frequent application of the threshold constraint. This indicates that the threshold derived from a small data set maintains robustness. We provide additional **visual results in Appendix A.8.5**, which show that our method preserves text accuracy even under accelerated settings.
>
> [1] Alex Jinpeng Wang, Dongxing Mao, Jiawei Zhang, Weiming Han, Zhuobai Dong, Linjie Li, Yiqi Lin, Zhengyuan Yang, Libo Qin, Fuwei Zhang, et al. Textatlas5m: A large-scale dataset for dense text image generation. arXiv preprint arXiv:2502.07870, 2025.

---

> ### Author Response · Authors · 2025-11-30
>
> >Q2: At equal perceptual quality (e.g. same FID score), which is faster: ETC or simply reducing steps with for example DDIM solver.
>
> A2: Since our method focuses on approximating the model output at each step, it can be **applied to different solvers**. This means ETC can further accelerate methods such as DDIM even after their step-reduction. To evaluate whether our approach provides consistent speedup across solvers without causing noticeable quality degradation, we conduct quantitative experiments using multiple solvers across different models. Specifically, for SDXL, we test our method with DDIM, iPNDM, and DPM-Solver++. For other models like FLUX, OpenSora, Wan, and TangoFlux, which predict $v$ rather than noise $\epsilon$, we select Heun[2], UniPC[3], Euler, and DPM++, with the *prediction_type* set to *flow_prediction*. Detailed solver settings can be found in Appendix A.8.3. Additionally, we did not re-tune the threshold values, and instead used the values specified in Table 1 in paper. The quantitative results are shown below.
>
> |SDXL|LPIPS↓|SSIM↑|PSNR↑|CLIP↑|FLOPs(p)↓|Speedup↑|Latency(s)↓|
> |:------: | :------: |:------: |:------: | :------: |:------: |:------: | :------: |
> |DDIM|-|-|-|27.278|0.67|1×|9.27|
> |ETC w/DDIM|0.103|0.881|28.106|27.251|0.27|2.15×|4.31|
> |IPNDM|-|-|-|26.121|0.67|1×|9.31|
> |ETC w/IPNDM|0.119|0.873|27.963|26.063|0.26|2.01×|4.65|
> |DPM++|-|-|-|26.668|0.67|1×|9.15|
> |ETC w/DPM++|0.096|0.882|28.678|26.662|0.24|2.15×|4.26|
>
> |FLUX|LPIPS↓|SSIM↑|PSNR↑|CLIP↑|FLOPs(p)↓|Speedup↑|Latency(s)↓|
> |:------: | :------: |:------: |:------: | :------: |:------: |:------: | :------: |
> |Heun|-|-|-|25.522|7.36|1×|50.21|
> |ETC w/Heun|0.053|0.931|29.421|25.301|1.56|4.33×|11.57|
> |UniPC|-|-|-|25.629|3.71|1×|28.54|
> |ETC w/UniPC|0.089|0.912|28.411|25.451|1.12|2.81×|10.13|
> |DPM++|-|-|-|25.734|3.71|1×|28.47|
> |ETC w/DPM++|0.068|0.922|29.052|25.409|1.26|2.78×|10.23|
>
> |OpenSora|LPIPS↓|SSIM↑|PSNR↑|VBench↑|FLOPs(p)↓|Speedup↑|Latency(s)↓|
> |:------: | :------: |:------: |:------: | :------: |:------: |:------: | :------: |
> |Heun|-|-|-|77.79%|5.96|1×|82.76|
> |ETC w/Heun|0.101|0.874|27.566|77.55%|1.98|2.71×|30.60|
> |UniPC|-|-|-|77.43%|3.08|1×|43.81|
> |ETC w/UniPC|0.125|0.859|25.565|77.06%|1.49|2.08×|21.04|
> |DPM++|-|-|-|77.39%|3.08|1×|43.71|
> |ETC w/DPM++|0.137|0.845|24.114|76.78%|1.09|2.24×|19.43|
>
> |Wan|LPIPS↓|SSIM↑|PSNR↑|VBench↑|FLOPs(p)↓|Speedup↑|Latency(s)↓|
> |:------: | :------: |:------: |:------: | :------: |:------: |:------: | :------: |
> |Heun|-|-|-|81.01%|150.68|1×|1906.29|
> |ETC w/Heun|0.105|0.807|24.155|80.73%|46.29|3.16×|603.26|
> |Euler|-|-|-|81.09%|76.57|1×|966.22|
> |ETC w/Euler|0.079|0.843|25.293|80.88%|29.03|2.43×|396.81|
> |DPM++|-|-|-|81.03%|75.39|1×|958.85|
> |ETC w/DPM++|0.093|0.825|24.867|80.74%|29.16|2.42×|397.11|
>
> |TangoFlux|FAD↓|MCD↓|KL↓|CLAP↑|FLOPs(T)↓|Speedup↑|Latency(s)↓|
> |:------: | :------: |:------: |:------: | :------: |:------: |:------: | :------: |
> |Heun|-|-|-|13.285|92.85|1×|10.98|
> |ETC w/Heun|0.027|1.879|0.158|13.273|28.82|2.59×|4.23|
> |UniPC|-|-|-|13.209|46.86|1×|5.56|
> |ETC w/UniPC|0.029|1.885|0.149|13.189|14.83|2.45×|2.27|
> |DPM++|-|-|-|13.276|46.86|1×|5.50|
> |ETC w/DPM++|0.027|1.880|0.153|13.270|14.83|2.43×|2.26|
>
> Our method maintains consistent acceleration and generation quality across different solvers without re-tuning the threshold. This indicates that the threshold we search for captures the error tolerance that prevents trajectory deviation in the model’s outputs. Since this property is intrinsic to the model rather than to the choice of solver, the same threshold remains applicable as long as the model itself is unchanged. More **visual results are provided in Appendix A.8.3**.
>
> [2] Karras T, Aittala M, Aila T, et al. Elucidating the design space of diffusion-based generative models[J]. Advances in neural information processing systems, 2022, 35: 26565-26577.
>
> [3] Zhao W, Bai L, Rao Y, et al. Unipc: A unified predictor-corrector framework for fast sampling of diffusion models[J]. Advances in Neural Information Processing Systems, 2023, 36: 49842-49869.

---

> > ### Author Response · Authors · 2025-11-30
> >
> > >Q3: Wall‑clock changes for different values of $n$ and give a simple "rule‑of‑thumb" for picking $n$.
> >
> > A3: We thank the reviewer for raising this important question. The results showing how the latency varies with respect to $n$ are provided below.
> >
> > |n for FLUX|2|4|6|7|8|9|10|11|12|
> > |:-:|:-:|:-:|:-:|:-:|:-:|:-:|:-:|:-:|:-:|
> > |SSIM↑|0.907|0.920|**0.926**|0.925|0.927|0.927|0.928|0.929|0.929|
> > |Latency(s)↓|10.37|10.55|**10.86**|11.20|11.88|12.45|12.82|13.21|13.92|
> >
> > |n for Wan|2|4|6|7|8|9|10|11|12|
> > |:-:|:-:|:-:|:-:|:-:|:-:|:-:|:-:|:-:|:-:|
> > |SSIM↑|0.722|0.786|**0.806**|0.806|0.810|0.811|0.811|0.813|0.813|
> > |Latency(s)↓|375.34|381.56|**389.89**|403.39|420.81|440.31||||
> >
> > As shown in the results, increasing $n$ beyond our selected value does not lead to significant improvement in SSIM, while latency continues to grow. This indicates that estimating the trend at the beginning of the stable phase already yields sufficiently small approximation error, further increasing $n$ may produce a slightly more accurate trend estimate, but the quality gain is negligible, whereas latency increases due to additional model evaluations.
> >
> > Conversely, when reduce $n$ below our selected value, the generation quality drops noticeably, yet the latency does not decrease much. This is because performing trend approximation too early—during the fluctuation phase—introduces larger errors, which quickly exceed the model’s error-tolerance threshold. As a result, fewer approximations can be performed, limiting the theoretical speedup.
> >
> > Therefore, choosing $n$ at the very beginning of the stable phase provides the optimal balance. A simple rule of thumb for selecting $n$ is to choose the denoising step at the transition between the fluctuation and stable phases, which can be directly identified through our threshold-search mechanism.
> >
> > >Q4: Algorithm/equation mismatch and notation/indexing inconsistencies for $d$.
> >
> > A4: Thanks for your careful and accurate reading of our paper. We have fixed the errors in the revised version.

---

### Official Review · Reviewer_iKFu · 2025-11-03

**Soundness:** 3
**Presentation:** 1
**Contribution:** 2
**Rating:** 2
**Confidence:** 3

**Summary:**

This submission focuses on the problem of generation time in diffusion models with multiple denoising steps. A new approach is presented to improve the sampling speed without significantly reducing the quality of samples, where instead of using only the most recent model output, ETC looks at all past outputs to estimate the denoising trend and create a smoothed weighted projection of historical changes.

**Strengths:**

- The idea of leveraging historical denoising trajectories to stabilize trend prediction and adaptively control approximation frequency is well-motivated, addressing instability in multi-step reuse methods.
- The evaluation performed with several different diffusion models with different domains is extensive and impressive, showing the scalability of the proposed approach

**Weaknesses:**

- The contribution is incremental in nature, in this submission an acceleration method is proposed that combines several observations to reduce the number of inference steps necessary to denoise the output. At the same time, evaluation and comparison to other methods does not include more complex approaches as for example second order denoising methods (DPM solver).
- In the presented comparison it seems that the proposed approach slightly outperforms evaluated approaches, but often providing higher speed-up in expense for the lower quality of generations
- I am puzzled, what is the motivation by enforcing the same trajectory between the original and accelerated method? I fail to see a proper explanation in the submission. I guess that maybe trajectory deviation might be the reason for reduced diversity of generations, but only if different trajectories converge to the same one. I do not see any experiments evaluating that. Could authors elaborate on that?
(Presentation) The introduction is filled with novel terms introduced by the authors without proper explanation, which makes it harder to understand the main contribution of the paper.
- “Despite their differing formulations, these approaches [Different formulations of diffusion process] all ensure the smooth temporal evolution of data features.” - this is not obvious for me. To the best of my knowledge there are no implicit mechanisms enforcing smoothness between different timesteps in any of the diffusion process formulations. This might be an emergent effect, but there are numerous works showing that the process varies significantly between different timesteps rather than opposite.
- The recursive trend estimation, dynamic window, and tolerance search introduce extra computations. The paper doesn’t quantify this overhead, so the “training-free” claim may still imply nontrivial offline cost.
- It is unclear for me what is the memory footprint of the proposed solution.
- Y axis in Fig 1a is confusing. I guess this is the error between ground truth model prediction and accelerated version. SImilarly X axis for Fig 3a is not explained before reference. Also “As shown in Figure 3a, we subtract a fixed value from the latent to represent the approximation error” - where is it shown?

**Questions:**

See weaknesses

---

> ### Author Response · Authors · 2025-11-30
>
> # Table 1. Quantitative results using different solvers.
>
> |SDXL|LPIPS↓|SSIM↑|PSNR↑|CLIP↑|FLOPs(p)↓|Speedup↑|Latency(s)↓|
> |:------: | :------: |:------: |:------: | :------: |:------: |:------: | :------: |
> |DDIM|-|-|-|27.278|0.67|1×|9.27|
> |ETC w/DDIM|0.103|0.881|28.106|27.251|0.27|2.15×|4.31|
> |IPNDM|-|-|-|26.121|0.67|1×|9.31|
> |ETC w/IPNDM|0.119|0.873|27.963|26.063|0.26|2.01×|4.65|
> |DPM++|-|-|-|26.668|0.67|1×|9.15|
> |ETC w/DPM++|0.096|0.882|28.678|26.662|0.24|2.15×|4.26|
>
> |FLUX|LPIPS↓|SSIM↑|PSNR↑|CLIP↑|FLOPs(p)↓|Speedup↑|Latency(s)↓|
> |:------: | :------: |:------: |:------: | :------: |:------: |:------: | :------: |
> |Heun|-|-|-|25.522|7.36|1×|50.21|
> |ETC w/Heun|0.053|0.931|29.421|25.301|1.56|4.33×|11.57|
> |UniPC|-|-|-|25.629|3.71|1×|28.54|
> |ETC w/UniPC|0.089|0.912|28.411|25.451|1.12|2.81×|10.13|
> |DPM++|-|-|-|25.734|3.71|1×|28.47|
> |ETC w/DPM++|0.068|0.922|29.052|25.409|1.26|2.78×|10.23|
>
> |OpenSora|LPIPS↓|SSIM↑|PSNR↑|VBench↑|FLOPs(p)↓|Speedup↑|Latency(s)↓|
> |:------: | :------: |:------: |:------: | :------: |:------: |:------: | :------: |
> |Heun|-|-|-|77.79%|5.96|1×|82.76|
> |ETC w/Heun|0.101|0.874|27.566|77.55%|1.98|2.71×|30.60|
> |UniPC|-|-|-|77.43%|3.08|1×|43.81|
> |ETC w/UniPC|0.125|0.859|25.565|77.06%|1.49|2.08×|21.04|
> |DPM++|-|-|-|77.39%|3.08|1×|43.71|
> |ETC w/DPM++|0.137|0.845|24.114|76.78%|1.09|2.24×|19.43|
>
> |Wan|LPIPS↓|SSIM↑|PSNR↑|VBench↑|FLOPs(p)↓|Speedup↑|Latency(s)↓|
> |:------: | :------: |:------: |:------: | :------: |:------: |:------: | :------: |
> |Heun|-|-|-|81.01%|150.68|1×|1906.29|
> |ETC w/Heun|0.105|0.807|24.155|80.73%|46.29|3.16×|603.26|
> |Euler|-|-|-|81.09%|76.57|1×|966.22|
> |ETC w/Euler|0.079|0.843|25.293|80.88%|29.03|2.43×|396.81|
> |DPM++|-|-|-|81.03%|75.39|1×|958.85|
> |ETC w/DPM++|0.093|0.825|24.867|80.74%|29.16|2.42×|397.11|
>
> |TangoFlux|FAD↓|MCD↓|KL↓|CLAP↑|FLOPs(T)↓|Speedup↑|Latency(s)↓|
> |:------: | :------: |:------: |:------: | :------: |:------: |:------: | :------: |
> |Heun|-|-|-|13.285|92.85|1×|10.98|
> |ETC w/Heun|0.027|1.879|0.158|13.273|28.82|2.59×|4.23|
> |UniPC|-|-|-|13.209|46.86|1×|5.56|
> |ETC w/UniPC|0.029|1.885|0.149|13.189|14.83|2.45×|2.27|
> |DPM++|-|-|-|13.276|46.86|1×|5.50|
> |ETC w/DPM++|0.027|1.880|0.153|13.270|14.83|2.43×|2.26|

---

> > ### Author Response · Authors · 2025-11-30
> >
> > >Q1: Contribution is incremental in nature and comparison to more complex approaches as for example second order denoising methods (DPM solver).
> >
> > A1: We thank the reviewer for the constructive comments. We realize that the concern may stem from our insufficient clarification in the manuscript regarding the distinction between our method and solver–based acceleration approaches. We therefore provide additional explanation here and will make this distinction clearer in the revised version to avoid potential confusion.
> >
> > First, our method does not fall under the solver-based acceleration category. Instead, our method belongs to the class of **step-wise model-output reuse** approaches. Solver-based acceleration methods (e.g., DDIM, DPM-Solver) improve sampling efficiency from the perspective of numerical integration by modifying ODE discretization, but they still require model inference per step. Their acceleration stems from higher-order or more stable integration schemes. In contrast, our method reduces the number of model inferences by approximating model outputs.
> >
> > Second, within the step-wise reuse framework, the central challenge lies in the fact that approximation errors in model outputs accumulate and amplify across denoising steps. **Appendix A.5** provides a theoretical derivation showing how such errors propagate over time, underscoring their non-negligible impact. Existing reuse-based methods do not offer sufficiently approximations of model outputs. For example, TeaCache estimates future trends solely from the output difference between two adjacent steps. When short-term fluctuations occur, this formulation amplifies those fluctuations, causing large approximation errors and degraded generation quality. Although SADA mitigates this issue via third-order finite differences to obtain a more stable trend estimation, but its approximation depth is limited to at most four steps before requiring a model inference, which restricts further acceleration.
> >
> > To address the challenge of maintaining accuracy under higher acceleration, our method is designed around two principles: (1) capturing global trend evolution across all historical model outputs, which suppresses short-term fluctuations and maintains trajectory consistency; and (2) estimating the model’s error-correction limit, enabling dynamic adjustment of approximation depth to maximize acceleration while avoiding noticeable quality degradation. These design choices target the core difficulties of the reuse paradigm, providing new solutions rather than incremental improvements.
> >
> > Finally, because our method operates at the level of model-output approximation, it is inherently orthogonal and fully compatible with solver-based approaches. To evaluate whether our approach provides consistent speedup across solvers without causing noticeable quality degradation, we conduct quantitative experiments using multiple solvers across different models. Specifically, for SDXL, we test our method with DDIM[1], iPNDM[2], and DPM-Solver++[3]. For FLUX, OpenSora, Wan, and TangoFlux, which predict $v$ rather than noise $\epsilon$, we select Heun[4], UniPC[5], Euler, and DPM++, with the *prediction_type* set to *flow_prediction*. Detailed solver settings can be found in Appendix A.8.3. Additionally, we did not re-tune the threshold values. As shown in Table 1, our method maintains consistent acceleration and generation quality across different solvers without re-tuning the threshold. This further highlights the generality and broad applicability of our approach. More **visual results are provided in Appendix A.8.3**.
> >
> > [1] Song J, Meng C, Ermon S. Denoising diffusion implicit models[J]. arXiv preprint arXiv:2010.02502, 2020.
> >
> > [2] Luping Liu, Yi Ren, Zhijie Lin, and Zhou Zhao. Pseudo numerical methods for diffusion models on manifolds. arXiv preprint arXiv:2202.09778, 2022.
> >
> > [3] Lu C, Zhou Y, Bao F, et al. Dpm-solver++: Fast solver for guided sampling of diffusion probabilistic models[J]. Machine Intelligence Research, 2025: 1-22.
> >
> > [4] Karras T, Aittala M, Aila T, et al. Elucidating the design space of diffusion-based generative models[J]. Advances in neural information processing systems, 2022, 35: 26565-26577.
> >
> > [5] Zhao W, Bai L, Rao Y, et al. Unipc: A unified predictor-corrector framework for fast sampling of diffusion models[J]. Advances in Neural Information Processing Systems, 2023, 36: 49842-49869.

---

> > > ### Author Response · Authors · 2025-11-30
> > >
> > > >Q2: Slightly outperforms evaluated approaches, but often providing higher speed-up in expense for the lower quality of generations.
> > >
> > > A2: We understand that the concern mainly arises from the minor differences between our method and the baselines in some metrics reported in Table 1. Here, we aim to clarify the nature of these differences and emphasize that our method achieves a balance between quality and speed without “trading substantial quality for acceleration.”
> > >
> > > In the SDXL experiments, our method shows slight decreases compared to SADA in LPIPS, SSIM, and PSNR, namely 9% (0.096 → 0.105), 0.6% (0.882 → 0.876), and 2% (28.881 → 28.017), while achieving an 18% speedup over SADA. In contrast, AdaptiveDiffusion, which achieves a comparable 10% speedup, exhibits much larger drops in LPIPS, SSIM, and PSNR: 79% (0.096 → 0.172), 7% (0.882 → 0.816), and 13% (28.881 → 25.062), respectively. Similarly, in the FLUX experiments, our method shows minor decreases of 9% (0.062 → 0.068) in LPIPS and 5% (29.342 → 29.176) in PSNR, while achieving a 22% speedup over SADA. In comparison, TeaCache-fast achieves a 16% speedup but suffers substantial quality degradation, with average drops of 353% (0.062 → 0.281) in LPIPS, 18% (0.923 → 0.775) in SSIM, and 35% (29.881 → 25.062) in PSNR.
> > >
> > > These results demonstrate that, unlike methods that trade quality for speed, such as AdaptiveDiffusion and TeaCache, our method achieves faster acceleration with only minor quality degradation. Across OpenSora, Wan, and TangoFlux, our method consistently provides both higher speed and better quality.
> > >
> > > Moreover, it is worth noting that **our method consistently outperforms other approaches in CLIP and VBench scores**. CLIP evaluates semantic alignment between text and generated images, while VBench assesses text-video consistency as well as visual quality in terms of color, structure, and temporal coherence—critical metrics for text-conditioned generation. The improvement in CLIP and VBench confirms that our method achieves a combined enhancement of speed and perceptual quality without sacrificing the fidelity of the generated content.
> > >
> > > >Q3: Motivation by enforcing the same trajectory between the original and accelerated method.
> > >
> > > A3: We thank the reviewer for highlighting the critical issue of trend consistency. First, the denoising process is a gradual evolution from $x_T$ to $x_0$. Without maintaining consistency in the denoising trajectory, the accumulated errors can amplify over the process of denoising (see **Appendix A.5** for a detailed proof), potentially altering the distribution of $x_0$ and leading to **semantic inconsistencies** or **structural distortions** in the generated content.
> > >
> > > To explore these issues, we increased the threshold in our method and applied four approximation steps in the first trend-approximation round. Detailed quantitative results and visualizations are provided in **Appendix A.9**. Under this setting, our method achieves up to 4–5× speedup. However, the generated results exhibit semantic inconsistencies and structural distortions. For example, as shown in **Figure 19 in the Appendix**, given the prompt “a person eating a burger,” the video generated without enforcing trend consistency includes only the semantic elements “person” and “burger,” but omits the “eating” action. Furthermore, complex structures such as fingers are distorted in both image and video generation. Therefore, **ensuring trend consistency is crucial for maintaining high-quality outputs** when applying acceleration.

---

> ### Author Response · Authors · 2025-11-30
>
> >Q4: No implicit mechanisms enforcing smoothness between different timesteps in any of the diffusion process formulations.
>
> A4: We thank the reviewer for raising this important point. It is true that different diffusion-process formulations do not inherently enforce smoothness in the evolution trajectory. However, we examine smoothness from both the forward and reverse processes.
>
> First, by the definition of the **forward process** in diffusion models, the variable at timestep $t$ is given by $x_t = \alpha(t)\cdot x_0+\beta(t)\cdot \epsilon$, where $\alpha(t)$ and $\beta(t)$ are noise scheduling coefficients. Taking expectation over $x_0\sim p(x_0)$,$\epsilon \sim \mathcal{N}(0,I)$, and $t\sim$ Uniform({$1,...,T$}), we get $E_{x_0,\epsilon,t}=E[E_{x_0}[\alpha(t)\cdot x_0]+E_\epsilon[\beta(t)\cdot \epsilon]]$.
>
> Since $\epsilon \sim \mathcal{N}(0,I)$, we have $E[\epsilon]=0$, and thus $E_{x_0,\epsilon,t} = E_t[\alpha(t)\cdot E_{x_0}[x_0]]=\alpha(t)\cdot E_{x_0}[x_0]$ .
>
> From the equation, we know that the trajectory {$x_t$} is continuous in expectation. Consequently, the empirical average $x_t$ also exhibits continuity by the law of large numbers[6], demonstrating that the **forward process evolves smoothly from $x_0$ to $x_t$**.
>
> For the **backward process**, the predictions in diffusion models can be categorized into two forms: $\epsilon$-prediction and $v$-prediction. The noise prediction model is trained to estimate the noise contained in $x_t$, which is proved to be equivalent to estimating the log-likelihood gradient by Ho et al.[7]. Taking expectation over $x_t\sim p(x_t)$ and $t \sim$Uniform$(1,...,T)$, we get  $E_{x_t,\epsilon_\theta,t}=A\cdot E_{x_t}[x_t] + B\cdot E_{\epsilon_\theta}[\epsilon_\theta(x_t,t)] (1)$, where $A=\sqrt{\bar{\alpha}_{t-1}}/ \sqrt{\bar{\alpha}_t}$, B=$\sqrt{1-\bar{\alpha}_t}(1-A)$.
>
> Suppose the training is sufficiently converged, then $\epsilon_\theta(x_t,t)\approx \epsilon \sim \mathcal{N}(0,I)$. Since $\epsilon_\theta \sim \mathcal{N}(0,I)$, we have $\mathbb{E}[\epsilon_\theta]=0$, and thus $E_{x_t,\epsilon_\theta,t}=A\cdot E_{x_t}[x_t]$$(2)$.
>
> Flow matching defines a vector field $v$ that guides the data from the initial distribution $x_0$ toward the target distribution $x_T$. Taking expectation over $x_t\sim p(x_t)$ and $t \sim$Uniform$(1,...,T)$, we get $E_{x_t,v_\theta,t}=E_{x_{t}}[x_t]-\frac{1}{T}E_{v_\theta}[v_\theta(x_t,t)]$$(3)$. Suppose the training is sufficiently converged, then $v_\theta(x_t,t)\approx v$,  and thus $E_{x_t,v_\theta,t}=\frac{T-t+1}{T-t}\cdot E_{x_t}[x_t]$$(4)$.
>
> **The detailed proof is provided in Appendix A.2.2**. From Equation (1) and Equation (2), a well-trained denoiser permits the sampling trajectory from  $x_t$ to $x_0$ continuous in expectation. Consequently, the empirical average $x_t$ also exhibits continuity by the law of large numbers, demonstrating that the backward process evolves smoothly from $x_t$ to $x_0$.
>
> However, as shown in Figure 12 (b) and (c) in the paper, the mean of the model’s output varies across timesteps during inference. This occurs because different timesteps correspond to different noise levels and contain different amounts of information, leading to variations in the predicted values. Consequently, as implied by Equations (3) and (4), under a fixed scheduler, the differences in the denoising trajectory are determined by the model $\theta$. Therefore, the diffusion process formulation cannot enforce smooth model outputs, and therefore cannot guarantee a smooth temporal evolution from $x_t$ to $x_0$.
>
> However, since the model outputs are Lipschitz continuous with respect to both $x_t$ and $t$ (**detailed proof provided in Appendix A.4**), short-term fluctuations may occur but will not deviate excessively from the overall evolutionary trend. In other words, the model outputs still exhibit structured and predictable variations across time steps. Based on this property, we design the trend-consistent prediction module, which captures the global characteristics of all historical trends to mitigate the effect of short-term fluctuations and yield more stable trend estimates. Since our method is designed at the model-output level, rather than at the level of $x_t$, the previous phrasing in the preliminaries referring to the “smooth temporal evolution of data features” may mislead readers. We have therefore revised the preliminaries to focus on the Lipschitz continuity of the model outputs and moved the discussion of smooth temporal evolution of data features to the appendix for better clarity and readability. We sincerely thank the reviewer for raising this important point.
>
> [6] William Feller et al. An introduction to probability theory and its applications, volume 963. Wiley New York, 1971.
>
> [7] Jonathan Ho, Ajay Jain, and Pieter Abbeel. Denoising diffusion probabilistic models. Advances in neural information processing systems, 33:6840–6851, 2020.

---

> ### Author Response · Authors · 2025-11-30
>
> >Q5: Cost of the recursive trend estimation, dynamic window and tolerance search, and the memory footprint of the proposed solution.
>
> A5: We thank the reviewer for raising this important point. Let the computational cost of computing the change between two trends be $C$, and the memory required to store a single trend be $M$. During inference, our method requires only **constant overhead**: at any timestep the computation needed to estimate the future trend is $C$, and the memory footprint is fixed at $2M$. To evaluate the computational overhead of our method, we measure both VRAM consumption and FLOPs. The quantitative results are shown below.
>
> ||FLOPs(M)↓|VRAM(MB)↓|
> |:------: | :------: |:------: |
> |SDXL+ETC|9.61|0.75|
> |FLUX+ETC|25.83|1.01|
> |OpenSora+ETC|23.65|18.34|
> |Wan+ETC|211.76|40.99|
> |TangoFlux+ETC|4.74|0.94
>
> Our method requires only a few of megabytes of additional VRAM during diffusion inference, and its computational cost is negligible compared with the overall diffusion model inference. This demonstrates that our approach is computation-efficient.
>
> The computational overhead of the threshold search is shown below. Our method requires only inference on a small dataset using a single GPU to determine the threshold. This cost is negligible compared with training-based approaches, which require large datasets, multi-node hardware, and many epochs of optimization. For example, SDXL-Lightning[8] is trained on 64× A100 80GB GPUs with a batch size of 512, whereas our threshold can be obtained on a single GPU in 1.6 minutes before deploying our acceleration. FastVideo[9] trains Wan acceleration on 8 nodes with 64× H200 GPUs using a global batch size of 64 for 3000 steps (~52 hours), while our threshold search completes in 151 minutes on one GPU. Therefore, in terms of hardware demand and time cost, our threshold search is significantly smaller than training-based methods; its offline overhead is minimal, highlighting the advantage of being training-free.
>
> |Model|Searching time (min)|
> |:------: | :------: |
> |SDX|1.61|
> |FLUX|4.92|
> |OpenSora|7.54|
> |Wan|151.74|
> |TangoFlux|0.93|
>
> [8] Shanchuan Lin, Anran Wang, and Xiao Yang. Sdxl-lightning: Progressive adversarial diffusion distillation, 2024.
>
> [9] Zhang P, Chen Y, Su R, et al. Fast video generation with sliding tile attention[J]. arXiv preprint arXiv:2502.04507, 2025.
>
> >Q6: Y axis in Fig 1a is confusing. X axis for Fig 3a is not explained before reference. Also “As shown in Figure 3a, we subtract a fixed value from the latent to represent the approximation error” - where is it shown.
>
> We thank the reviewer for carefully pointing out these issues. In Fig. 1a, the Y-axis represents the model output at a specific location of the latent (for example, if the latent has a shape of (4096,64), we plot the value at location [1024,32] across timesteps). The purpose of this figure is to visualize how closely our method and other acceleration approaches match the ground-truth model outputs at each denoising step. For Fig. 3a, the X-axis denotes the magnitude of noise added to the latent at a given timestep, which is used to simulate different levels of accumulated approximation error. This figure illustrates how the model output changes under varying degrees of latent perturbation compared with the error-free output. We also acknowledge that the sentence “As shown in Figure 3a, we subtract a fixed value from the latent to represent the approximation error” was misleading due to an incorrect ordering. The description of the operation should precede the reference to the figure. Otherwise, it gives the impression that the figure visualizes the operation. We have revised the manuscript to correct this ordering and ensure the explanation is clear. We thank the reviewer again for the helpful feedback.

---

### Author Response · Authors · 2025-12-03
**Rebuttal Summary**

Our paper proposes a training-free stepwise diffusion acceleration method that leverages all historical model outputs to estimate future trends, and employs a model-specific error threshold to control the accumulation of approximation errors. These two components are designed to ensure denoising trajectory consistent under high acceleration, effectively balancing image quality and inference speed. In response to reviewer concerns, we provide the following additional experiments and clarifications:

1. **Robustness across out-of-domain prompts, samplers, and resolutions**. We added extensive experiments covering multiple solvers, resolutions, and challenging prompts such as dense text. Results show that **our method generalize well** to these diverse settings, preserving both acceleration performance and image quality without deviations.

2. **Comparison with  intra-layer feature reuse and training-based acceleration methods**. Experiments indicate that out method achieves higher speed and better quality than  intra-layer feature reuse approaches, while requiring far lower deployment overhead than training-based few-step methods. Our method can also be **combined** with both categories with minimal quality loss, demonstrating **generality and composability**.

3. **Novelty relative to step-wise diffusion acceleration methods like TeaCache and SADA**. We replace TeaCache's trend-estimation and SADA's stability-control modules with our trajectory-consistent trend prediction and error-threshold mechanism and **improvements in either quality or speed**. These improvement validating our method innovations in both output trend approximation and acceleration error-controlled.

4. **Computational overhead of the proposed method**. We provide a quantitative analysis of the additional memory usage and computational cost. Across different models, the introduced overhead is marginal relative to the overall inference pipeline and does not impose significant constraints in deployment.

5. **Additional clarifications and theoretical supplements**. We added the proof of the continuity of model output changes, a workflow supplement for analyzing fluctuation points in threshold search methods, and the motivation for ensuring trajectory consistency.

All corresponding experiments, analyses, and clarifications have been incorporated into the revised paper, expanding it from **16 pages to 33 pages** and improving the readability and completeness.

---

### Meta-Review · Area_Chair_j2Uj · 2026-01-05

**Summary:**

Reviewer iKFU raised concerns on the incremental contribution, minor performance improvements, and asked for reporting of the efficiency and memory overhead. Reviewer XXmn raised concerns on lack of theoretical proof of error consistency assumption, raised questions such as out of domain prompt generalization and comparison to other solvers like DDIM. Reviewer sJqw raised similar concerns of the generalization of the learned threshold, and pointed out the proposed method is similar to TeaCache and SADA, and no comparisons with them are reported. Reviewer cWaK also raised several concerns on the lack of experimental comparisons and justification.

**Reviewer Concerns:**

Authors did spend a lot of effort addressing reviewers’ questions and addressed some of them with new experiments but reviewers did not get the chance to confirm upon them.

But AC feels the concern of the limited novelty and advantage of the method over other alternatives is still not clear, e.g. improvement over SADA is not obvious. Also, multiple concerns on lack of experiments from all reviewers, for which authors provided many new results, but this would be way significant effort/change over the original version.

Also generalization for out-of-domain prompts, samplers and resolutions are not fully convincing and need significant more experiments to justify.

**Reviewer Scores:**

Reviewers probably will keep their scores.

---

### Decision · Program_Chairs · 2026-01-26

Reject